# Optimization of the proliferation and persistency of CAR T cells derived from human induced pluripotent stem cells

Tatsuki Ueda[1,2], Sara Shiina[1,3], Shoichi Iriguchi[1,3], Seitaro Terakura [4],
Yohei Kawai[1], Ryotaro Kabai[5], Satoko Sakamoto[5], Akira Watanabe[5],
Kohei Ohara[1], Bo Wang[1,3], Huaigeng Xu[6], Atsutaka Minagawa[1], Akitsu Hotta [6],
Knut Woltjen [7], Yasushi Uemura[8], Yuzo Kodama[9], Hiroshi Seno[2],
Tetsuya Nakatsura[8], Koji Tamada[10] & Shin Kaneko [1,3]✉

The effectiveness of chimaeric antigen receptor (CAR) T-cell immunotherapies against solid tumours relies on the accumulation, proliferation and persistency of T cells at the tumour site. Here we show that the proliferation of CD8αβ cytotoxic CAR T cells in solid tumours can be enhanced by deriving and expanding them from a single human induced-pluripotent-stem-cell clone bearing a CAR selected for efficient differentiation. We also show that the proliferation and persistency of the effector cells in the tumours can be further enhanced by genetically knocking out diacylglycerol kinase, which inhibits antigen-receptor signalling, and by transducing the cells with genes encoding for membrane-bound interleukin-15 (IL-15) and its receptor subunit IL-15Rα. In multiple tumour-bearing animal models, the engineered hiPSC-derived CAR T cells led to therapeutic outcomes similar to those of primary CD8 T cells bearing the same CAR. The optimization of effector CAR T cells derived from pluripotent stem cells may aid the development of long-lasting antigen-specific T-cell immunotherapies for the treatment of solid tumours.

When T cells are stimulated by cognate antigen-presenting cells, several types of downstream cell-signalling pathways effectively and coordinately induce T-cell effector functions. Research on chimaeric antigen receptor (CAR)-expressing T (CAR-T) cells have revealed that the quality of these signals is crucial for generating effective responses[1–4]. T-cell signalling is primarily composed of three signal categories. The first is the CD3ζ-mediated signal, which broadly branches to downstream pathways to induce different types of transcription factors that activate the T cells. The second is the co-stimulatory signal, transmitted by a co-stimulatory molecule, such as CD28 and 4-1BB, on the T-cell surface to coordinately enhance the first signal. To enhance this second signal, a CAR construct includes co-stimulatory domain(s) with CD3ζ. The third

[1]Shin Kaneko Laboratory, Department of Cell Growth and Differentiation, Center for iPS Cell Research and Application (CiRA), Kyoto University, Kyoto, Japan. [2]Department of Gastroenterology and Hepatology, Kyoto University Graduate School of Medicine, Kyoto, Japan. [3]Takeda-CiRA Joint Program (T-CiRA), Fujisawa, Japan. [4]Department of Hematology and Oncology, Nagoya University Graduate School of Medicine, Nagoya, Japan. [5]Medical Innovation Center, Kyoto University Graduate School of Medicine, Kyoto, Japan. [6]Department of Clinical Application, Center for iPS Cell Research and Application (CiRA), Kyoto University, Kyoto, Japan. [7]Department of Life Science Frontiers, Center for iPS Cell Research and Application (CiRA), Kyoto University, Kyoto, Japan. [8]Division of Cancer Immunotherapy, Exploratory Oncology Research and Clinical Trial Center, National Cancer Center, Kashiwa, Japan. [9]Department of Gastroenterology, Kobe University Graduate School of Medicine, Kobe, Japan. [10]Department of Immunology, Yamaguchi University Graduate School of Medicine, Yamaguchi, Japan. ✉e-mail: kaneko.shin@cira.kyoto-u.ac.jp

signal is a cytokine signal transmitted by JAK/STAT. Different γ-chain cytokines, such as interleukin (IL)-2, IL-7, IL-15 and IL-21, have been used as the third signal. Effective modulation of these signals in T cells is expected to enhance the functions of killer and helper cells, prolong T-cell survival and avoid T-cell exhaustion to increase the therapeutic efficacy of T-cell therapy.

Emerging technologies to regenerate cytotoxic immune cells, such as T cells and natural killer (NK) cells, from pluripotent stem cells are expected to provide a universally accessible approach to cancer immunotherapy[5,6]. Regeneration of antigen-specific T cells via induced pluripotent stem cells (iPSCs) was first reported in 2013 (refs. [7,8]) and is a potential source of cytotoxic T lymphocytes (CTLs)[6]. In this strategy, T cells are reprogrammed into iPSCs (T-iPSCs), which are subsequently differentiated back into T cells (iPS-T cells) but with rejuvenated phenotypes[7]. (For the remainder of this study, all comments about iPSCs refer to T-iPSCs unless otherwise stated.) However, previous reports have shown that iPS-T cells can have unexpected phenotypes. Especially in the case of CAR modifications in iPSCs, cytotoxic T cells derived from second-generation CD19 CAR-transduced iPSCs were reported to have properties of γδ T cells, partially expressing CD8αα but not CD8αβ, according to the gene expression profiles and CAR-independent cytotoxicity[5]. Recent reports indicated improved differentiation protocols to synthesize CD8αβ-expressing T cells that showed effector functions more closely resembling primary T cells[6,9,10]. CAR transduction to such iPS-T cells was confirmed to work as effectively as primary T cells on a B-cell malignancy animal model when iPS-T cells were supported with an IL-15-mediated third signal[11]. However, unlike haematological malignancy, solid tumours are more refractory to cellular immunotherapies with respect to accessibility and sustained T-cell effector function at the local tumour site.

In this Article, to overcome the obstacles using CD8αβ iPS-T cells derived from CAR-modified iPSCs, we selected optimal CAR constructs without tonic signalling during the CD8αβ T-cell differentiation. Next, the CD3ζ-mediated signal pathway was enhanced by inhibiting the intracellular immunological checkpoint molecules, namely DGKα and DGKζ by clustered regularly interspaced short palindromic repeats (CRISPR)-Cas9, to allow the proliferation of iCAR-T cells in the tumour. Finally, membrane-bound IL-15/IL-15RA was transduced in iCAR-T cells to prolong their survival by enhancing the third signal. The therapeutic efficacy of modified iCAR-T cells was confirmed in tumour-bearing animal models, demonstrating the generation of highly effective iCAR-T cells through a combination of selecting the appropriate CAR constructs, optimizing the differentiation process and enhancing the CAR and cytokine signalling by genetic manipulation. These findings suggest that our iCAR-T cells have therapeutic effects against solid tumours; they are comparable to primary CAR-T cells but with longer persistency. Thus, these findings indicate the potential to enhance the cancer immunity of CAR-T-cell therapies using iPSC technology.

## Results

### Selection of the CAR construct impacts the differentiation of CAR-iPSCs into T lineage cells

During their ex vivo expansion, CAR tonically signals to induce the exhaustion phenotype[12]. Antigen-independent tonic signalling is reported to affect the lymphopoiesis of CAR-engineered haematopoietic stem cells (HSCs) and promote NK cell-like development[13,14]. In contrast, T cells derived from CD19 CAR-engineered iPSCs show phenotypic similarities to γδ T cells[5]. The effects of tonic signals, including T-cell differentiation, have been reported to depend on the CAR construct[12]. We, therefore, evaluated different CAR constructs with the same scFv for their impact on the T-cell differentiation of iPSCs. We transduced first- to third-generation CARs, consisting of CD19-targeting FMC63 scFv, *CD3ζ* and co-stimulatory signalling molecules (Fig. 1a) into haematopoietic progenitor cells obtained from TKT3v1-7, a healthy donor αβT-iPSC line[7], using a lentiviral vector. Haematopoietic progenitor

cells were used to induce CD4CD8 double-positive T cells (DP cells). The differentiation efficiency from CD4CD8 double-negative cells (DN cells) to DP cells was the same for 4-1BBz-based second-generation and third-generation CARs; however, it was substantially decreased if the iPSCs were modified by first-generation or 28z-based second-generation CAR compared with the differentiation efficiency of CAR-unmodified iPSCs (Fig. 1b). We confirmed that no molecules targeted by CAR, such as CD19, were expressed on iPSCs or differentiating cells during the culture (Extended Data Fig. 1a) and that differentiation to DP cells remained uninterrupted under the scFv-deleted construct (Fig. 1b). To confirm the effects of CAR expression on T-cell differentiation from iPSC-derived haematopoietic progenitor cells, we transduced a doxycycline-inducible CAR-harbouring lentiviral vector (inducible 1928z; Fig. 1a) into iPSCs. CAR induction at the early T-cell differentiation stage (days 14–35) interfered with the differentiation of DP cells (Extended Data Fig. 1b). Moreover, we confirmed that CAR induction of 1928z but not 1928bbz in differentiating cells caused the phosphorylation of CD3ζ, suggesting non-specific activation signalling (Fig. 1c). Recent investigations reported that 1928z transduction into HSCs impaired T-cell differentiation capability[13,14] and promoted NK-like cell development by suppressing the transcription factor BCL11B, which is indispensable for T-lineage development of lymphoid progenitors during early phases of ex vivo T-cell generation. It could be a possible reason also for T-cell differentiation from 1928z CAR-transduced iPSC[5].

CAR induction at the DP cell-enriched stage (days 35–42) decreased the number of CD4CD8 DP cells and increased the number of CD8-single positive (SP) cells (Extended Data Fig. 1c). In addition, we found that 1928z induction increased the expression of NK cell-related genes, such as *CD161*, *DNAM-1*, *CD56*, *NKG2D*, *NKG2A*, *NKp46* and *NKp44* (Extended Data Fig. 1d). As we confirmed that 1928bbz-transduced iPSCs efficiently differentiated into CD4CD8DP cells compared with 1928z-transduced iPSCs (Fig. 1d), we decided to use the third-generation construct for further study.

### GPC3 CAR-iPSCs generated two types of iCAR-T cells with distinct phenotypes

A previous study[5] has reported that regenerated 1928z CAR-T cells derived from CAR-engineered iPSCs exerted anti-tumour function comparable to that of γδ T cells transduced with the same CAR. Therefore, the generation of CD8αβ-positive T cells from CAR-iPSCs via CD4CD8 DP is expected to enhance the anti-tumour function. Recently, we and other groups reported the generation of adaptive-like CD8αβ-positive T cells from iPSCs[6,9,10]. However, how the lineage modification impacts cell function, especially in vivo, requires more study. Although CAR-T therapies are effective against haematological malignancies, they are less effective against solid tumours. To develop an iPSC-derived CAR-T therapy for solid tumours, we recently generated CAR-expressing iPSC-derived NK/innate lymphocyte cells (ILCs) that target glypican3 (GPC3)-expressing tumours[15], such as hepatocellular carcinoma and ovarian clear cell carcinoma. GPC3 is a cancer-specific cell membrane protein and a promising target for cancer immunotherapy. To understand how the T-cell lineage derived from CAR-iPSCs impacts anti-tumour function, we compared progeny cells from CD4CD8 DP with previously reported innate-like iPS-T cells.

A CAR-targeting GPC3 (Fig. 2a) was transduced into iPSCs (CAR-iPSCs) using a lentiviral vector, and the cells were cloned by limiting the dilution after selection by stably expressing humanized Kusabira Orange 1 (hKO1). CAR-iPSCs were induced to form immature T cells (immature iCAR-T), which correspond to a CD4CD8 DP population, via haematopoietic progenitor cells (Fig. 2b). After subsequent maturation by an anti-CD3 antibody with dexamethasone[6], immature iCAR-T cells were differentiated into iCAR-T cells post-maturation, a population that includes CD8αβ SP cells. The analysis of surface molecules related to the memory T-cell phenotype revealed that immature iCAR-T and

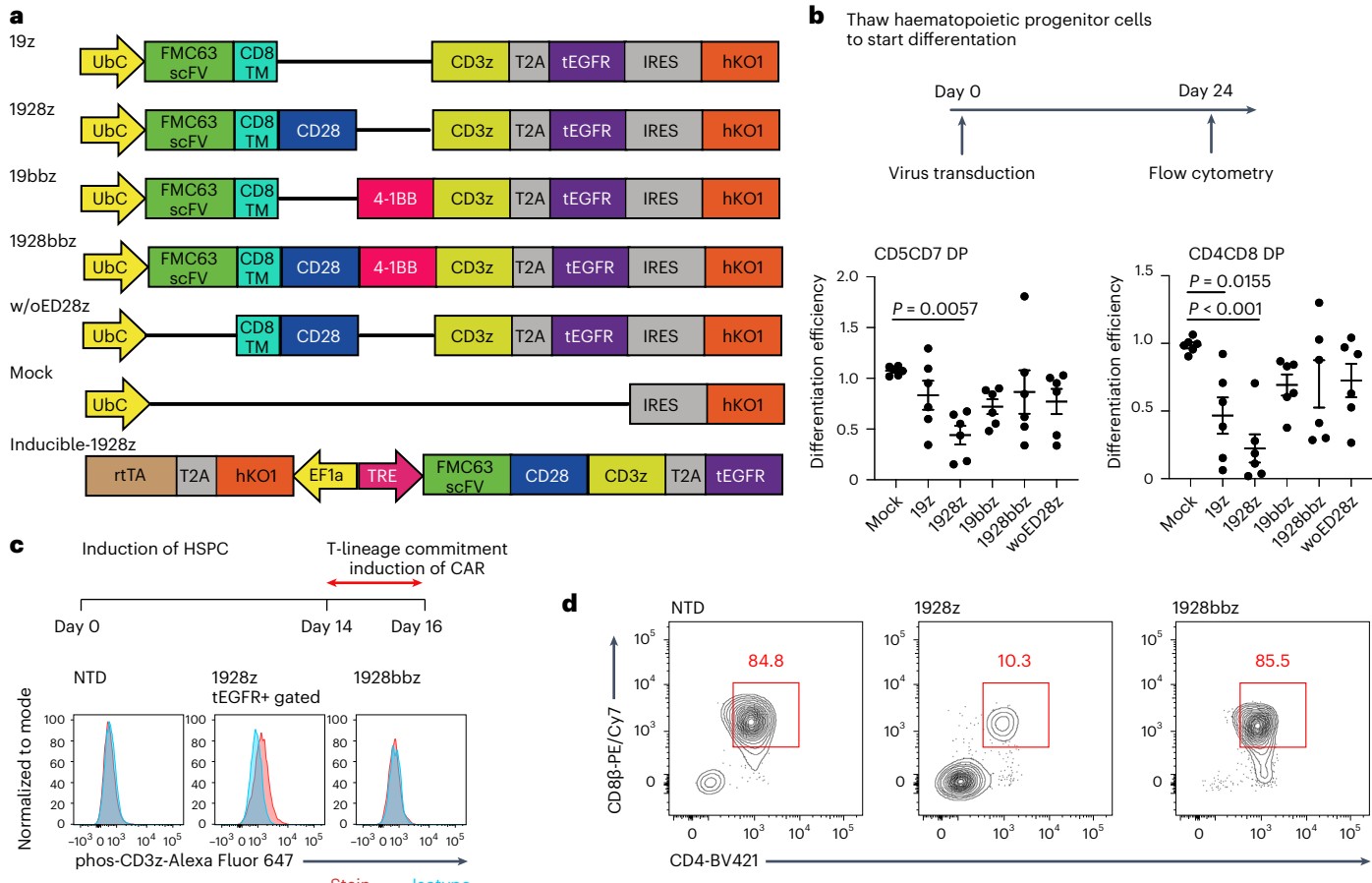

**Fig. 1 | The CAR construct impacts the differentiation of CAR-iPSCs into T-cell lineages. a**, Schematic presentation of the CAR constructs 19z, 1928z, 19bbz, 1928bbz and w/oED28z CAR (w/oED28z lacks the extracellular scFv domain of 1928z CAR). Each construct was inserted at the indicated internal promotor(s) of the lentiviral vector pCS. **b**, Comparison of surface antigen profiles of immature iCAR-T cells from various kinds of iCARs. Haematopoietic progenitor cells derived from iPSCs were transduced with CARs (see **a**) and differentiated into T-cell lineages. To compensate for well-to-well variation, we used hKO1-negative cells in the same well as internal controls. Differentiation efficiencies were calculated as follows: CD5CD7 DP = (percentage of CD5CD7 DP cells in hKO1-positive cells)/(percentage of CD5CD7 DP cells in hKO1-negative cells); CD4CD8 DP = (percentage of CD4CD8 DP cells in hKO1-positive cells)/(percentage of CD4CD8 DP cells in hKO1-negative cells). Each dot represents biological replicates ($n = 6$). One-way ANOVA with Dunnett's multiple comparisons test. **c**, Haematopoietic progenitor cells derived from inducible 1928z or 1928bbz CAR-transduced T-iPSCs were divided into two groups and subsequently differentiated on FcDLL4 for 2 days in the presence (2 μg ml⁻¹) or absence of doxycycline. Representative FCM data representing the phosphorylation of CD3ζ in the two groups are shown. **d**, Representative FCM data comparing surface antigen expression of CD4 and CD8β on immature iCAR-T cells from 1928z-transduced and 1928bbz-transduced iPSCs. NTD, not transduced.

iCAR-T post-maturation showed heterogeneous expression profiles regarding the surface antigens CD5 and CD8β that are expressed in peripheral CD8 T cells. Moreover, the cells showed heterogeneous expression patterns for CD62L, CCR7, CD27 and CD28, which are primarily expressed in naïve and central memory T cells (Fig. 2c,d).

To clarify the functional properties of CAR-iPSC-derived CD5CD8β DP cells through in vitro and in vivo assays, we expanded the cells and subsequently confirmed the conserved expression of CD8αβ (iCAR-T$_{CTL}$; Extended Data Fig. 2a). Furthermore, primary CD8 T cells modified with the same CAR (pCAR-T$_{CTL}$) and previously reported iPS-T cells with ILC-like function (iCAR-T$_{ILC}$) were prepared. iCAR-T$_{ILC}$, which were directly induced from DN cells using an agonistic stimulation protocol[7,16,17], expressed low levels of CD5, high levels of CD56 and CD161, and did not express CD8β, a phenotype that is consistent with that of ILCs[18] (Extended Data Fig. 2a,b), although the pre-rearranged T-cell receptor (TCR)-derived CD3 expression was preserved (Extended Data Fig. 2a). To avoid any biases from clone-specific phenotypes, we used the same clone, TKT3V1-7, to generate both iCAR-T$_{CTL}$ and iCAR-T$_{ILC}$.

Although the three types of cells commonly expressed certain T/NK cell lineage markers, TCRαβ/CD3 complex and transduced CAR, differences existed in the expression of cell lineage-related markers (Extended Data Fig. 2b). To further characterize the three types of CAR-expressing cells, we conducted a transcriptional analysis of 259 human T-cell-related genes (Supplementary Table 1) at the single-cell level. Among differentially expressed genes (DEGs) between iCAR-T$_{CTL}$ and iCAR-T$_{ILC}$, the expression of naïve/memory-related genes, such as *SELL*, *CCR7*, *TCF7*, *IL7R* and *CD27*, was high in iCAR-T$_{CTL}$, suggesting that CD5CD8β DP iCAR-T$_{CTL}$ maintained a suitable phenotype for therapeutic efficacy in vivo compared with iCAR-T$_{ILC}$ even after proliferation (Extended Data Fig. 2c). Gene Ontology (GO) analysis of those genes revealed a high enrichment of genes related to T-cell differentiation (fold enrichment 47.34, false discovery rate (FDR) $2.24 \times 10^{-11}$) and T-cell activation (fold enrichment 34.04, FDR = $1.06 \times 10^{-11}$). On the other hand, the top 30 DEGs for pCAR-T$_{CTL}$ compared with iCAR-T$_{CTL}$ included *IL2*, *IFNG*, *TNF* and *IL7R*, indicating possible enrichment of terms for cytokine-mediated signalling pathways (fold enrichment 21.12, FDR $8.18 \times 10^{-23}$) and lymphocyte activation (fold enrichment

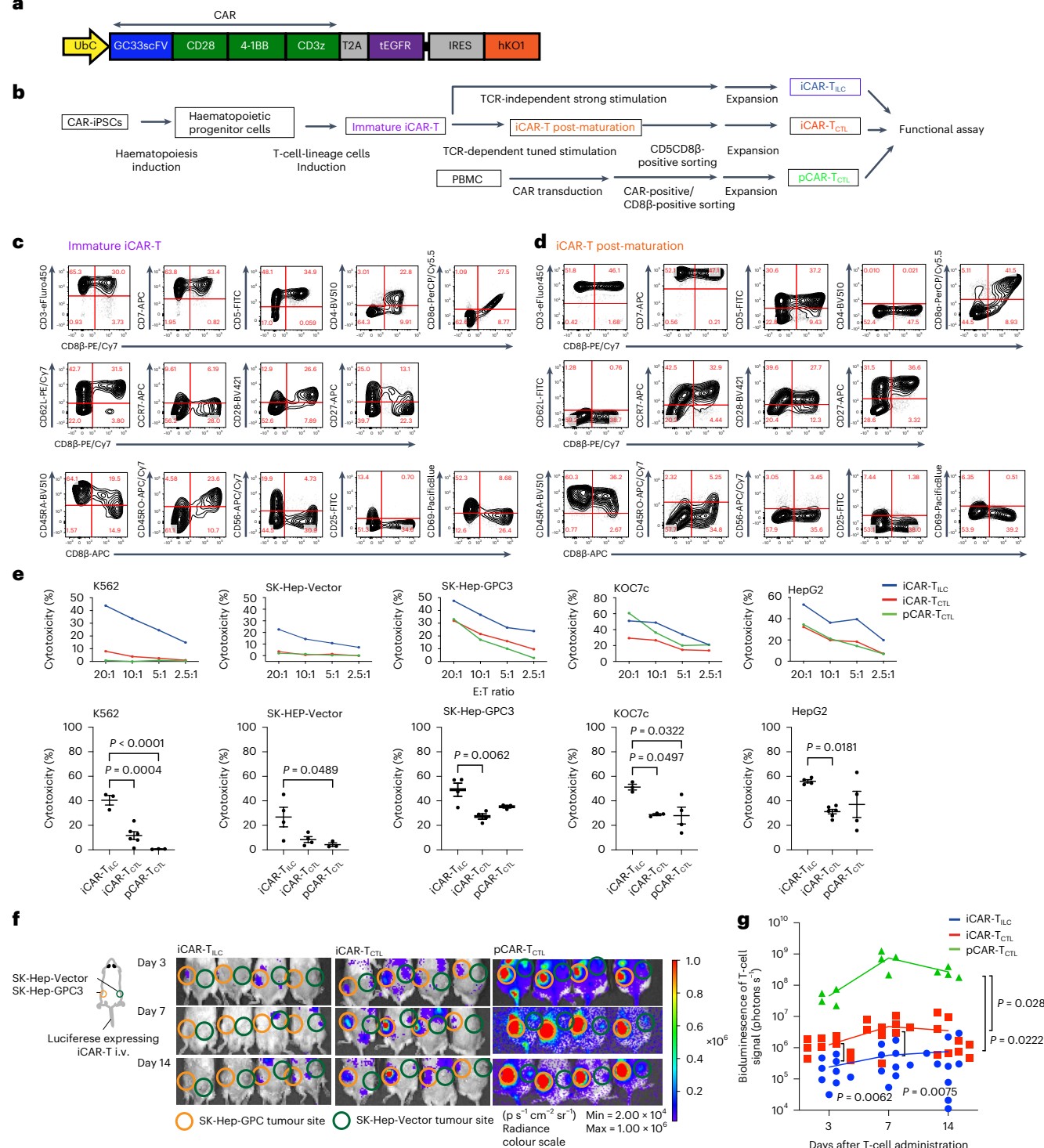

**Fig. 2 | iCAR-T$_{CTL}$ suppressed tumour progression and accumulated to the tumour site *in vivo* better than iCAR-T$_{ILC}$ but did not reach pCAR-T$_{CTL}$.**
**a**, Schematic presentation of the GC28bbz CAR construct conjugated with *tEGFR* genes by aT2A. The construct was inserted at an internal promotor, UbC, of the lentiviral vector pCS-UbC-RfA-IRES-hKO1. **b**, Schematic illustration of the differentiation of iCAR to iCAR-T post-maturation. **c**,**d**, Surface antigen profiles of immature iCAR-T (**c**) and iCAR-T post-maturation (**d**) from CD3-positive αβT-iPSCs (clone TKT3V1-7) lentivirally transduced with GC28bbz CAR. The results show surface protein expression patterns on day 21 on OP9-DL1. Each number in the panels indicates the percentage of cells in the corresponding red rectangles. **e**, $^{51}$Cr release assay of iCAR-T$_{CTL}$, iCAR-T$_{ILC}$ and pCD8CART against GPC3-positive (SK-Hep-GPC3, Koc7c and HepG2) or negative (SK-Hep-Vector and

K562) cancer cell lines. $n = 3–6$, mean ± s.e.m. statistical tests. The bottom panel shows the cytotoxicity at E:T ratio of 20:1. One-way ANOVA with Tukey's multiple comparisons test. **f**, Five million SK-Hep-GPC3 and SK-Hep-Vector cancer cell lines were inoculated at the left or right flank of an NSG mouse, respectively. Next, $1 × 10^7$ luciferase-transduced iCAR-T$_{CTL}$, iCAR-T$_{ILC}$ or pCAR-T$_{CTL}$ were injected intravenously (i.v.) from the tail vein 14 days after the tumour inoculation. In vivo imaging surrounding the tumour inoculation sites. Orange and green circles indicate measurements obtained from the sites inoculated with SK-HEP-GPC3 and SK-Vector, respectively. **g**, Total flux (photons s$^{-1}$) of the injected iCAR-T or pCAR-T cells in the SK-Hep-GPC3 tumour was quantified at the indicated timepoints. $n = 9$ (iCAR-T$_{ILC}$, iCAR-TCTL), $n = 4$ (pCAR-T$_{CTL}$) mean ± s.e.m. Two-way ANOVA with Tukey's multiple comparisons test.

23.71, FDR $3.75 \times 10^{-13}$). These results suggested that iCAR-T$_{CTL}$ could be functionally closer to pCAR-T$_{CTL}$ than iCAR-T$_{ILC}$ (Extended Data Fig. 2d). However, it could be insufficient in multiple aspects of cancer immunity in comparison with pCAR-T$_{CTL}$.

### iCAR-T$_{CTL}$ suppressed tumour progression and accumulated to the tumour site in vivo better than iCAR-T$_{ILC}$ but did not reach pCAR-T$_{CTL}$

To elucidate the functions of the three cell types, we first evaluated the target-mediated CAR-dependent cytokine production of iCAR-T$_{CTL}$ and iCAR-T$_{ILC}$ by co-culturing the cells with SK-Hep-1 (a liver cancer cell line) transduced with GPC3 (SK-Hep-GPC3) as a stimulator. Although both cell types produced IFN-γ and TNF upon stimulation, the levels were lower than those of expanded pCAR-T$_{CTL}$ (Extended Data Fig. 2e). We did not observe any difference between the impact of 4-1BBz-based second-generation and third-generation CAR on T-cell differentiation. Thus, we compared cytokine production between second-generation BBz iCAR-T$_{CTL}$ and third-generation 28BBz iCAR-T$_{CTL}$, and found that 28BBz iCAR-T$_{CTL}$ produced IFN-γ and TNF significantly better than BBz iCAR-T$_{CTL}$ following SK-Hep-GPC3 stimulation (Extended Data Fig. 3). Thus, we selected third-generation 28BBz iCAR-T$_{CTL}$ for further experiments. Next, to examine CAR-dependent and CAR-independent cytotoxicity, which characterizes the target-antigen specificity, iCAR-T$_{CTL}$, iCAR-T$_{ILC}$ and pCAR-T$_{CTL}$ were co-cultured with GPC3-absent SK-Hep-1 (SK-Hep-Vector), K562 (leukaemia cell line), GPC3-present SK-Hep-GPC3, KOC7c (ovarian cancer cell line) and HepG2 (hepatocellular carcinoma cell line) (Fig. 2e). iCAR-T$_{CTL}$ and pCAR-T$_{CTL}$ showed similar levels of CAR target-specific cytotoxicity in all GPC3-positive cell lines. In contrast, iCAR-T$_{ILC}$ showed CAR-dependent and CAR-independent cytotoxicity, as estimated by NK cell-activating receptor-mediated signalling[16,17], resulting in stronger cytotoxicity against GPC3-expressing cells than iCAR-T$_{CTL}$. iCAR-T$_{CTL}$ showed similar tumour-suppressive function as iCAR-T$_{ILC}$ in a peritoneal dissemination model of ovarian cancer, KOC7c (Extended Data Fig. 4a–d).

In the cancer immunity cycle[19], an effective T-cell therapy depends on the trafficking of T cells to the tumour site, expansion of T cells accompanied by their recognition of tumour cells, and the duration of the effector function. Accordingly, we investigated these functions in iCAR-T$_{CTL}$ and iCAR-T$_{ILC}$. The cellular kinetics of cells were evaluated using NSG mice carrying either SK-Hep-Vector or SK-Hep-GPC3, transplanted subcutaneously at each flank on day −14. Afterward, iCAR-T$_{CTL}$, iCAR-T$_{ILC}$ or pCAR-T$_{CTL}$ engineered to express luciferase were injected into the tail vein of tumour-bearing mice on day 0. Bioluminescence imaging of mice revealed luminescence intensity gradually and selectively that significantly increased at the SK-Hep-GPC3-transplanted site from days 3 to 7 in iCAR-T$_{CTL}$-injected mice and considerably in pCAR-T$_{CTL}$-injected mice (Fig. 2f,g and Extended Data Fig. 5a). Almost none of iCAR-T$_{CTL}$, iCAR-T$_{ILC}$ or pCAR-T$_{CTL}$ accumulated at the SK-Hep transplanted site. In addition, we confirmed the accumulation of iCAR-T$_{CTL}$ by pathological analysis and flow cytometry (FCM) analysis. SK-Hep-GPC3 tumour sections from mice infused with iCAR-T$_{CTL}$ showed an enhanced infiltration of CD3-positive T cells compared with iCAR-T$_{ILC}$ infusion (Extended Data Fig. 5b). Infiltrating iCAR-T$_{CTL}$ was positive for the active cell cycle molecule Ki-67; moreover, some were positive for the cytotoxic molecule Granzyme B as demonstrated by immunohistopathology (Extended Data Fig. 5c). Next, regarding the iCAR-T$_{CTL}$ group, the characteristics of iCAR-T$_{CTL}$ observed in the tumour were evaluated by comparing them with iCAR-T$_{CTL}$ found in the spleen. Global transcriptional profiles of iCAR-T$_{CTL}$ in the tumour and spleen revealed that the two groups have distinguishable transcriptional profiles (Extended Data Fig. 5d). In total, 1,492 genes were upregulated in isolated iCAR-T$_{CTL}$ from the tumour, and further analysis demonstrated that iCAR-T$_{CTL}$ in the tumour showed enrichment of gene sets related to proliferation, such as mitotic cell cycle, cell division, M phase and mitotic cell cycle phase transition, and related to effector function,

such as cytokine signalling in the immune system and interferon signalling (Extended Data Fig. 5e,f).

### Signal 1 enhancement by DGK deletion improved the accumulation, persistency and effector function of iCAR-T$_{CTL}$

Although iCAR-T$_{CTL}$ showed preferable kinetics in a systemic injection model, its therapeutic efficacy in a local injection model and accumulation at the tumour site in the systemic injection model was inferior to those of primary CAR-T. This finding led us to consider whether the tumour microenvironment negatively affected the functions of iCAR-T$_{CTL}$. As several signals (first signal: TCR signal, second signal: co-stimulatory signal, third signal: cytokine signal) are necessary to efficiently activate T cells, we hypothesized that a combinatory enhancement of these signals overcomes the insufficient function of iCAR-T$_{CTL}$. To enhance antigen receptor-mediated first signal, we modified PD-1 signalling. PD-1-deleted iPSC was established and differentiated to iCAR-T$_{CTL}$ to assess if PD-1 deletion was effective in keeping the differentiated iCAR-T$_{CTL}$ activated (Extended Data Fig. 6a). PD-1 deletion significantly but slightly improved cytotoxicity and proliferation and did not improve ERK phosphorylation and cytokine production of iCAR-T$_{CTL}$ (Extended Data Fig. 6b–g). The tumour-suppressive capability was not enhanced by blocking the combination of iCAR-T$_{CTL}$ and PD-1 by antibody (Extended Data Fig. 6h). As a different approach, CAR overexpression slightly improved cytotoxicity; however, it showed less proliferation and cytokine producibility with increasing expression of exhaustion markers (Extended Data Fig. 7a–g).

Next, we focused on controlling the diacylglycerol (DAG) metabolism. Following TCR- or CAR-mediated phosphorylation of CD3ζ, DAG recruits and activates Ras guanyl nucleotide-releasing protein 1 (RasGRP1) to activate the MEK/ERK pathway. The two major DAG kinase isoforms, DGKα and DGKζ, found in T cells are known to attenuate the MEK/ERK signalling by degrading DAG[20]. The inhibition of DGK enhances the RAS/ERK pathway-mediated AP-1 and NF-κB activation[21]. In addition, disruption of DGK enhanced T-cell effector function[22,23]. Therefore, we disrupted both DGKα and DGKζ by CRISPR–Cas9 at the iPSC stage (DGK-dKO-iCAR) (Extended Data Fig. 8a), and subsequently differentiated iPSCs into iCAR-T$_{CTL}$. Although we observed decreased efficiency of T-cell differentiation along with disruption of both DGKs (Supplementary Fig. 1a), which is compatible with the previous observation in DGK-dKO mice[24], we successfully obtained DGK-dKO-iCAR-T$_{CTL}$ that were confirmed to have no DGKα and DGKζ proteins (Extended Data Fig. 8b). Next, we evaluated their phenotype by FCM and performed gene expression analysis to compare with iCAR-T$_{CTL}$ (Extended Data Fig. 8c and Supplementary Fig. 2a,b). DGK disruption did not considerably affect naïve/memory phenotype except slight upregulation of *CCR7*. It increased the expression of metabolic fitness genes, activation genes and immune regulatory genes, and decreased NK cell-related activating receptor genes and exhaustion genes such as *HAVCR2* (TIM3) and *PDCD1* (PD-1).

Next, Phosflow assay was performed to evaluate the phosphorylation downstream of DAG in the tested T cells. Although ERK1/2 phosphorylation in iCAR-T$_{CTL}$ was less than that of pCAR-T$_{CTL}$, DGK-dKO-iCAR-T$_{CTL}$ showed enhanced ERK1/2 phosphorylation that was similar to pCAR-T$_{CTL}$ when the cells were stimulated by the target antigen (Fig. 3a). The level of recovered ERK phosphorylation showed that the proliferation capability of DGK-dKO-iCAR-T$_{CTL}$ was significantly improved compared with DGK-unmodified iCAR-T$_{CTL}$ in response to SK-Hep-GPC3 (Fig. 3b). In addition, improved IFNγ and TNF production from DGK-dKO-iCAR-T$_{CTL}$ was observed in response to SK-Hep-GPC3 (Supplementary Fig. 2c). To determine if DGK disruption improved the effector function and therapeutic persistency of iCAR-T$_{CTL}$ at the tumour site, we treated an intraperitoneal dissemination model of an ovarian cancer cell line, KOC7c, in NSG mice by intraperitoneally injecting DGK-dKO-iCAR-T$_{CTL}$. As expected, DGK-dKO-iCAR-T$_{CTL}$ significantly suppressed tumour growth compared with iCAR-T$_{CTL}$

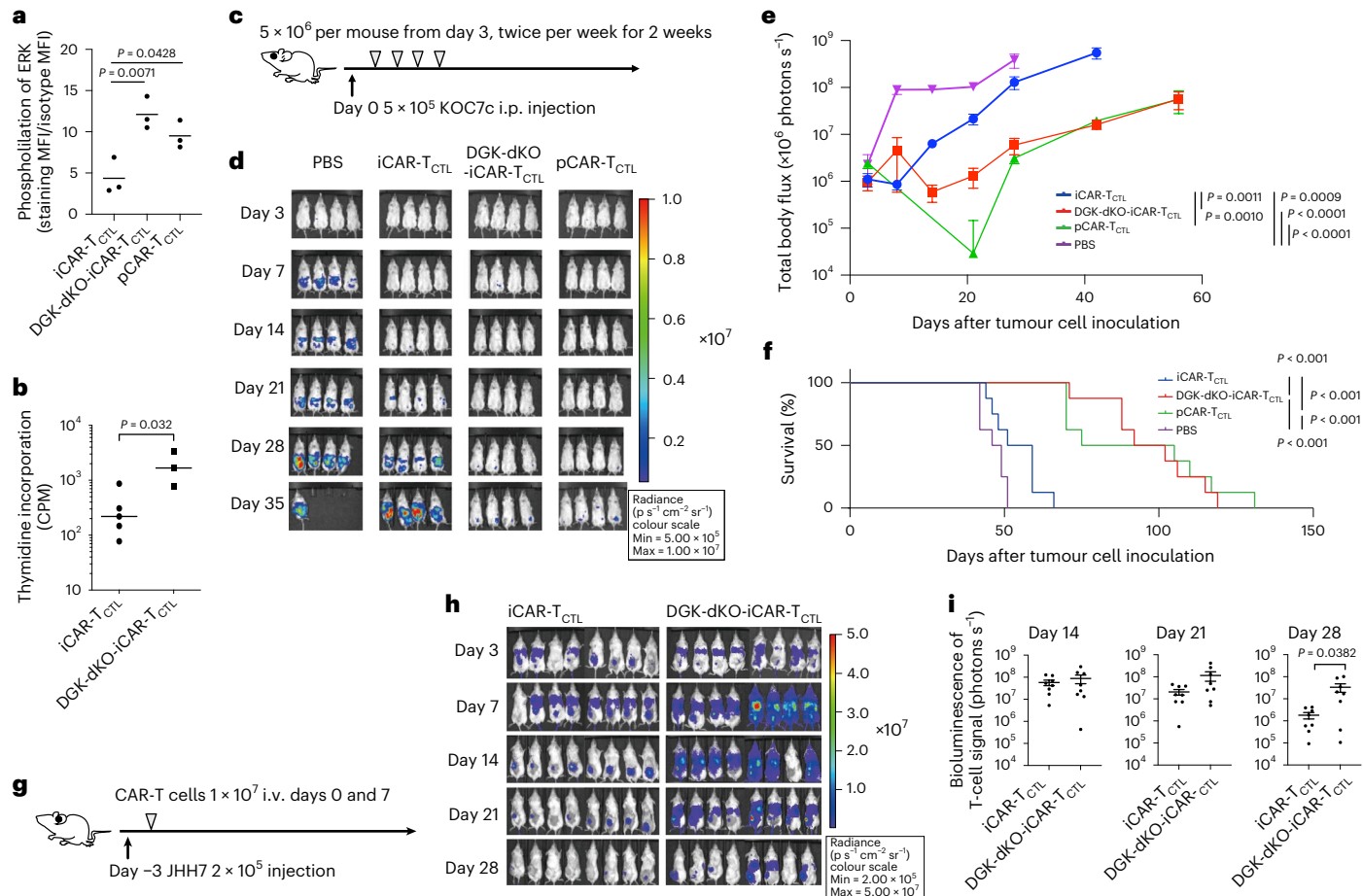

**Fig. 3 | Signal enhancement improved the accumulation, persistency and effector function of iCAR-T$_{CTL}$. a**, Detection of phosphorylated ERK (pERK) in iCAR-T$_{CTL}$, DGK-dKO-iCAR-T$_{CTL}$ and pCAR-T$_{CTL}$ 60 min after co-culturing with irradiated SK-Hep-GPC3. One-way ANOVA with Tukey's multiple comparisons test **b**, Target cell-mediated proliferation of iCAR-T$_{CTL}$ and DGK-dKO-iCAR-T$_{CTL}$ was determined by co-culturing with irradiated SK-Hep-GPC3 and using a standard $^3$H-thymidine incorporation assay at 72 h ($n = 3$, mean ± s.e.m.). Two-sided Student's $t$-test. **c**, Therapeutic efficacy of iCAR-T$_{CTL}$ and DGK-dKO-iCAR-T$_{CTL}$ in a peritoneal ovarian cancer model, and treatment schedule of the ovarian cancer peritoneal dissemination xenograft model: NSG mice were injected intraperitoneally (i.p.) with $5 × 10^5$ KOC7c cells expressing luciferase, and from the third day after the ovarian cancer inoculation, $5 × 10^6$ iCAR-T$_{CTL}$, DGK-dKO-iCAR-T$_{CTL}$ or pCAR-T$_{CTL}$ were injected intraperitoneally twice a week for 2 weeks ($n = 8$ for each group). **d**, In vivo bioluminescence imaging of luciferase-labelled

KOC7c in NSG mice treated with iCAR-T$_{CTL}$, DGK-dKO-iCAR-T$_{CTL}$ or pCAR-T$_{CTL}$. **e,f**, Change in the total body flux as the total tumour volume (mean ± s.e.m.) (**e**) and survival (**f**) were evaluated at the indicated timepoints after the injection. One-way ANOVA with Tukey's multiple comparisons test and log-rank test with Bonferroni multiple comparisons test. **g**, Therapeutic efficacy of iCAR-T$_{CTL}$ and DGK-dKO-iCAR-T$_{CTL}$ with a subcutaneous liver cancer model; treatment schedule of the liver cancer subcutaneous xenograft model. NSG mice were injected intraperitoneally with $2 × 10^5$ JHH7 cells 3 days before treatment. In total, $1 × 10^7$ iCAR-T$_{CTL}$ and DGK-dKO-iCAR-T$_{CTL}$ were injected intravenously on days 0 and 7. **h**, In vivo bioluminescence imaging of injected T cells in NSG mice treated with iCAR-T$_{CTL}$, DGK-dKO-iCAR-T$_{CTL}$. **i**, Total flux (photons s$^{-1}$) of the injected iCAR-T cells in the JHH7 tumour was quantified at the indicated timepoints ($n = 8$, mean ± s.e.m.). Two-sided Student's $t$-test.

and prolonged mouse survival to a level comparable to the pCAR-T$_{CTL}$-treated group (Fig. 3c–f).

Next, to evaluate the effect on survival and proliferation of DGK-dKO-iCAR-T$_{CTL}$ in vivo, we injected DGK-dKO-iCAR-T$_{CTL}$ into the tail vein of JHH7-xenografted NSG mice and performed in vivo imaging of the subcutaneous tumour. We found that DGK-dKO-iCAR-T$_{CTL}$ accumulated more and persisted longer at the subcutaneous tumour site than iCAR-T$_{CTL}$ (Fig. 3g–i). FCM analysis of the resected tumour supported the improved presence of DGK-dKO-iCAR-T$_{CTL}$ in the tumour (Extended Data Fig. 8d); however, the therapeutic impact was limited (Extended Data Fig. 8e).

**Signal 3 enhancement by mbIL15 transduction improved the accumulation, persistency and effector function of iCAR-T$_{CTL}$**

We next focused on the augmentation of the third signal. As IL-15 increased the proliferation of iCAR-T$_{CTL}$ in response to anti-CD3 antibody

and target antigen-expressing cell line, the most effective among three kinds of signal 3 cytokines, IL-15, IL-7 and IL-21 (Supplementary Fig. 3a), we focused on enhancing the IL-15 signal pathway to improve the persistency in vivo and maintain the memory phenotype. Membrane-bound IL-15/IL-15Rα (mbIL15) gene has been previously reported to improve the persistence and anti-tumour effect of genetically modified primary T cells in a xenograft animal model[25]. We transduced the mbIL15 gene (mbIL15tg) with a retroviral vector (pMX-mbIL15, kindly provided by Dr Nakayama; Fig. 4a) into iCAR-T$_{CTL}$ and pCAR-T$_{CTL}$ (Supplementary Fig. 3b) and checked the impact of mbIL15 on their phenotype by FCM (Supplementary Fig. 3c,d) and gene expression profile (Supplementary Fig. 2a,b). The mbIL15-transduced iCAR-T$_{CTL}$ significantly increased the expression of memory-related markers such as CCR7 and CD62L showing no elevation of exhaustion-related maker expression (Supplementary Fig. 3c,d). With respect to the gene profile, mbIL15 overexpression increased the expression of certain early memory-related marker

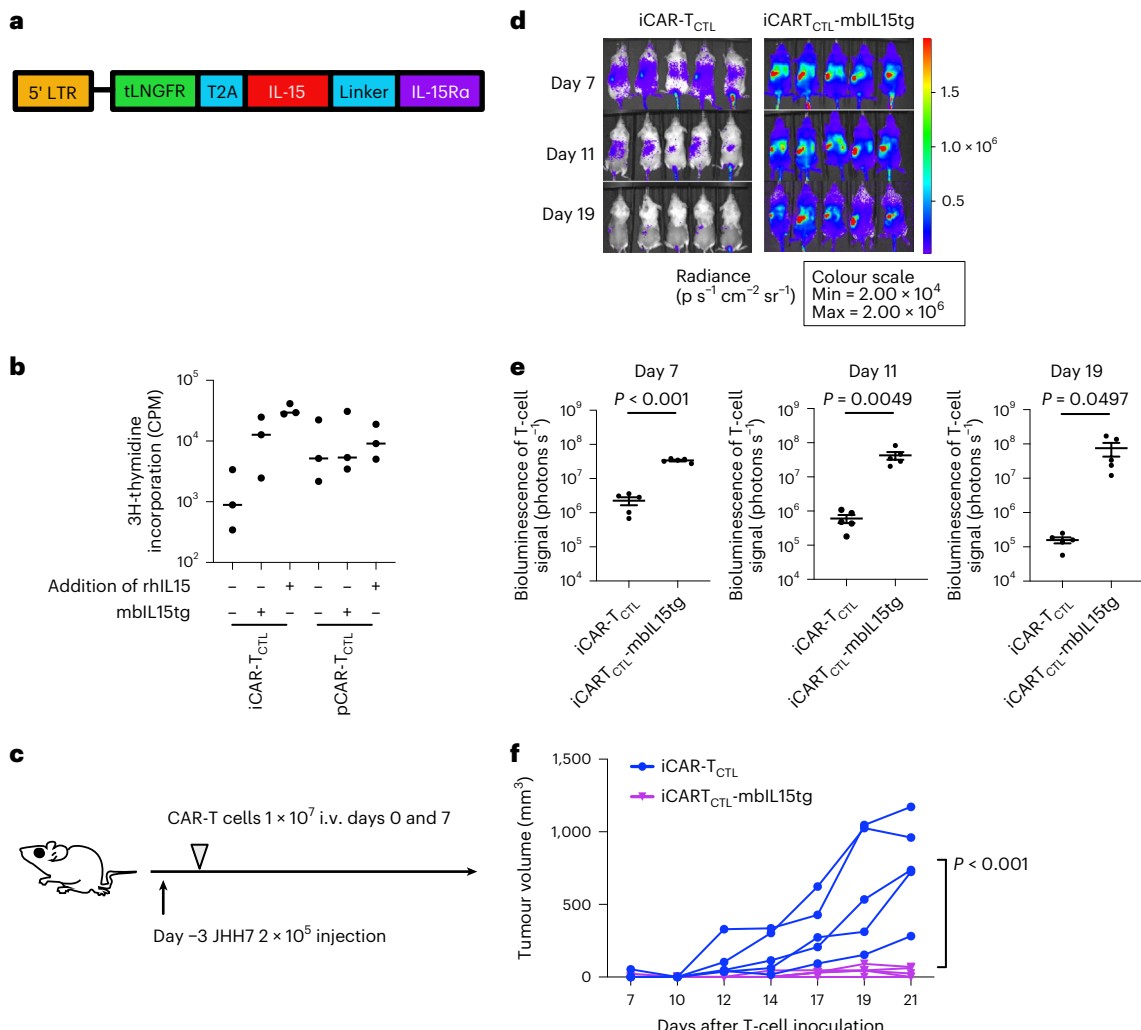

**Fig. 4 | Signal enhancement improved the accumulation, persistency, and effector function of iCAR-T$_{CTL}$. a**, Schematic presentation of the mbIL-15 constructs used in this study. Each construct was inserted under the retroviral long terminal repeat (LTR) promoter. tLNGFR, truncated form of low-affinity nerve growth factor receptor and marker of transduced mbIL15. **b**, Proliferation of iCAR-T$_{CTL}$ and pCAR-T$_{CTL}$ in the presence or absence of additional IL-15 or transgene modification of mbIL-15 ($n = 3$, mean ± s.e.m.). Proliferation was determined 72 h after co-culturing with irradiated SK-Hep-GPC3 using a standard [3H]-thymidine incorporation assay. **c**, Therapeutic efficacy of iCAR-T$_{CTL}$ and iCAR-T$_{CTL}$ mbIL-15 with a subcutaneous liver cancer model. The treatment

schedule of the liver cancer subcutaneous xenograft model. NSG mice were injected intraperitoneally (i.p.) with $2 \times 10^5$ JHH7 cells 3 days before treatment. Three days after the liver cancer inoculation, $1 \times 10^7$ iCAR-T$_{CTL}$ or iCAR-T$_{CTL}$-mbIL15tg were administered intravenously. **d**, In vivo bioluminescence imaging of the injected T cells in NSG mice. **e**, Total flux (photons s$^{-1}$) of the injected iCAR-T cells in the JHH7 tumour was quantified at the indicated timepoints ($n = 5$ of each group). Two-sided Student's $t$-test. **f**, Volume of inoculated JHH7 at the indicated timepoints in individual mice treated with the indicated test cells. Mean tumour size ± s.e.m. ($n = 5$ of each group). Two-way ANOVA.

genes such as *LEF1* and *SELL*, increased the expression of AKT/mTOR signal-related genes and decreased the expression of exhaustion-related markers in iCAR-T$_{CTL}$ (Supplementary Fig. 2b).

We next compared the proliferative potential upon co-culture with SK-Hep-GPC3 in the presence or absence of rhIL-15. We found that the transduction of mbIL15 enhanced the target-dependent expansion of iCAR-T$_{CTL}$ in vitro (Fig. 4b). In addition, the transduction of mbIL15 prolonged the persistence and enhanced the anti-tumour effect of both iCAR-T$_{CTL}$ and pCAR-T$_{CTL}$ in vivo (Fig. 4c–f and Supplementary Fig. 4a–d).

**Combinatorial enhancement of signals 1 and 3 on iCAR-T$_{CTL}$ improved anti-tumour function of cells in tumour-bearing animal model**

To assess the function of combinatorial signal-enhanced CAR-T cells, we transduced mbIL15 into DGK-dKO-iCAR-T$_{CTL}$. We examined the

impact of each signal enhancement on the cell phenotype by comparing the gene expression profiles of iCAR-T$_{CTL}$, iCAR-T$_{CTL}$-mbIL15tg, DGK-dKO-iCAR-T$_{CTL}$ and DGK-dKO-iCAR-T$_{CTL}$-mbIL15tg. Principal component analysis and hierarchical clustering analysis revealed that iCAR-T$_{CTL}$-mbIL15tg and DGK-dKO-iCAR-T$_{CTL}$ formed a distinct population from iCAR-T$_{CTL}$ (Supplementary Fig. 2a). The transduction of the *mbIL15* gene resulted in an enrichment of DNA conformation change, DNA replication, chromosome organization, DNA metabolic process, DNA-dependent replication and cell cycle, whereas DGK-dKO induced enrichment of inflammatory response, regulation of response to external stimulus, locomotion, cell migration, regulation of cell proliferation and cell motility (Supplementary Table 2). DGK-dKO increased the activation and co-stimulation-related genes, whereas mbIL15tg increased the expression of naïveness-related genes and decreased that of exhaustion-related genes. In combination with both manipulations, DGK-dKO iCAR-T$_{CTL}$-mbIL15tg showed additional expression of genes

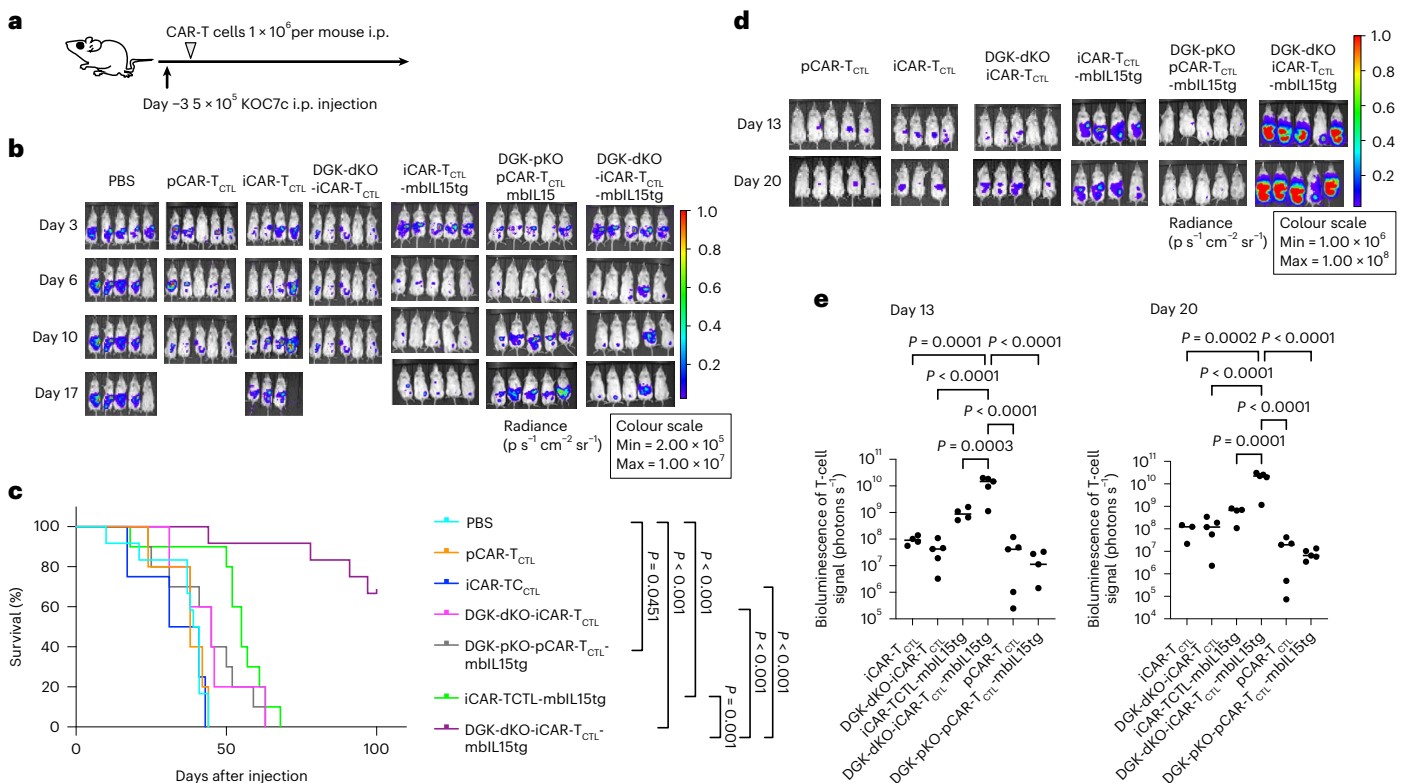

**Fig. 5 | Combinatorial signal enhanced iCAR-T$_{CTL}$ anti-tumour function.**
**a**, Treatment schedule of the ovarian cancer peritoneal dissemination xenograft model. NSG mice were injected intraperitoneally with $5 \times 10^5$ KOC7c cells expressing luciferase. From the third day after the ovarian cancer inoculation, $1 \times 10^6$ iCAR-T$_{CTL}$, DGK-dKO-iCAR-T$_{CTL}$, iCAR-T$_{CTL}$-mbIL15tg, DGK-dKO-iCAR-T$_{CTL}$-mbIL15tg, pCAR-T$_{CTL}$ or DGK-pKO-pCAR-T$_{CTL}$-mbIL15tg were injected intraperitoneally ($n = 5$ for each group). **b**, In vivo bioluminescence imaging of *Renilla* luciferase-labelled KOC7c. **c**, Survival of treated mice at the indicated timepoints after the injection. log-rank test with Bonferroni multiple comparisons test. **d**, In vivo bioluminescence imaging of firefly luciferase-labelled injected CAR-T cells in NSG mice. **e**, Total flux (photons s$^{-1}$) of injected CAR-T cells at the indicated timepoints. One-way ANOVA with Tukey's multiple comparisons test.

related to cell survival and persistence such as *TP53*, *MYC* and *ICOS* to DGK-dKO iCART$_{CTL}$ (Supplementary Fig. 2b). In vitro, these types of cells had similar cytotoxicities (Supplementary Fig. 2d); however, the antigen-dependent cytokine production capacity was enhanced by DGK deletion (Supplementary Fig. 2c). To evaluate if the combination of IL-15 signalling and DGK-dKO enhanced the anti-tumour effect at the tumour site in vivo, we administered $1 \times 10^6$ test cells, which is equivalent to 1/20 the cells administered in Extended Data Fig. 4, into the peritoneal cavity of a KOC7c pre-inoculated peritoneal dissemination mouse model (Fig. 5a). Although iCAR-T$_{CTL}$ lost their tumour suppressive capability with this small number of cells, the cohort of mice treated with the mbIL-15 gene-modified DGK-dKO-iCAR-T$_{CTL}$ showed significantly prolonged survival (Fig. 5b,c) and significantly better cell persistence than other cohorts (Fig. 5d,e).

In addition, we intravenously injected $1 \times 10^6$ test cells, which is equivalent to 1/20 of the cells administered in Fig. 4, into a JHH7 pre-inoculated subcutaneous solid tumour model (Fig. 6a). Similarly to the results obtained from the peritoneal injection model, DGK-dKO-iCAR-T$_{CTL}$-mbIL15tg showed significantly better tumour control and survival than other cohorts (Fig. 6b–d), whereas the same combinational signal enhancement of pCAR-T$_{CLT}$ could not be proved to be effective, which could be attributed to the limited efficiency of genome editing with a current protocol[23] we applied (Supplementary Fig. 2e). FCM analysis of the peripheral blood at 28 days after the injection confirmed that only DGK-dKO-iCAR-T$_{CTL}$-mbIL15tg was detected among all cohorts (Supplementary Fig. 2f), indicating that the IL-15-mediated improved persistency of

DGK-dKO-iCAR-T$_{CTL}$ contributed to the therapeutic efficacy in the solid tumour model.

Next, we evaluated the therapeutic impact of these genetic modifications on iCAR-T$_{ILC}$ in comparison with iCAR-T$_{CTL}$ to understand if these modifications generated better iCAR-T$_{ILC}$ than iCAR-T$_{CTL}$ (Extended Data Fig. 9 and Supplementary Figs. 4–6). In comparison with DGK-dKO iCAR-T$_{CTL}$, we did not find any advantage of DGK-dKO iCAR-T$_{ILC}$ about ERK phosphorylation and proliferation in co-culture with SK-Hep-GPC3 (Supplementary Fig. 5). In addition, we did not observe any significant results with respect to peritoneally disseminated tumour control when test cells were directory injected into the peritoneal cavity of immunodeficient mice (Extended Data Fig. 9). iCAR-T$_{ILC}$ exhibited stronger but non-specific cytotoxicity to tumour cell lines than iCAR-T$_{CTL}$ (Supplementary Fig. 6) after transduction of the *mbIL-15* gene; however, we still found inferiority of tumour accumulation of iCAR-T$_{ILC}$-mbIL15tg in comparison with iCAR-T$_{CTL}$-mbIL15tg when they were intravenously injected into JHH7-bearing mice (Supplementary Fig. 7). On the basis of these results, we conclude that iCAR-T$_{CTL}$-based modified cells would be useful for solid tumour immunotherapy.

To know if this combinatory modification strategy could be applied to other CARs, we transduced second-generation 19bbz-CAR into above-characterized GPC3 iCAR-T$_{CTL}$ and DGK dKO GPC3 iCAR-T$_{CTL}$-mbIL15, and found signal enhancement and proliferation advantage of the combination of IL-15 expression and DGK disruption in vitro (Extended Data Fig. 10a–e) as well as enhanced T-cell survival and tumour suppressive capability in vivo (Extended Data Fig. 10f–i). These results suggest that enhancing the combinational signals of

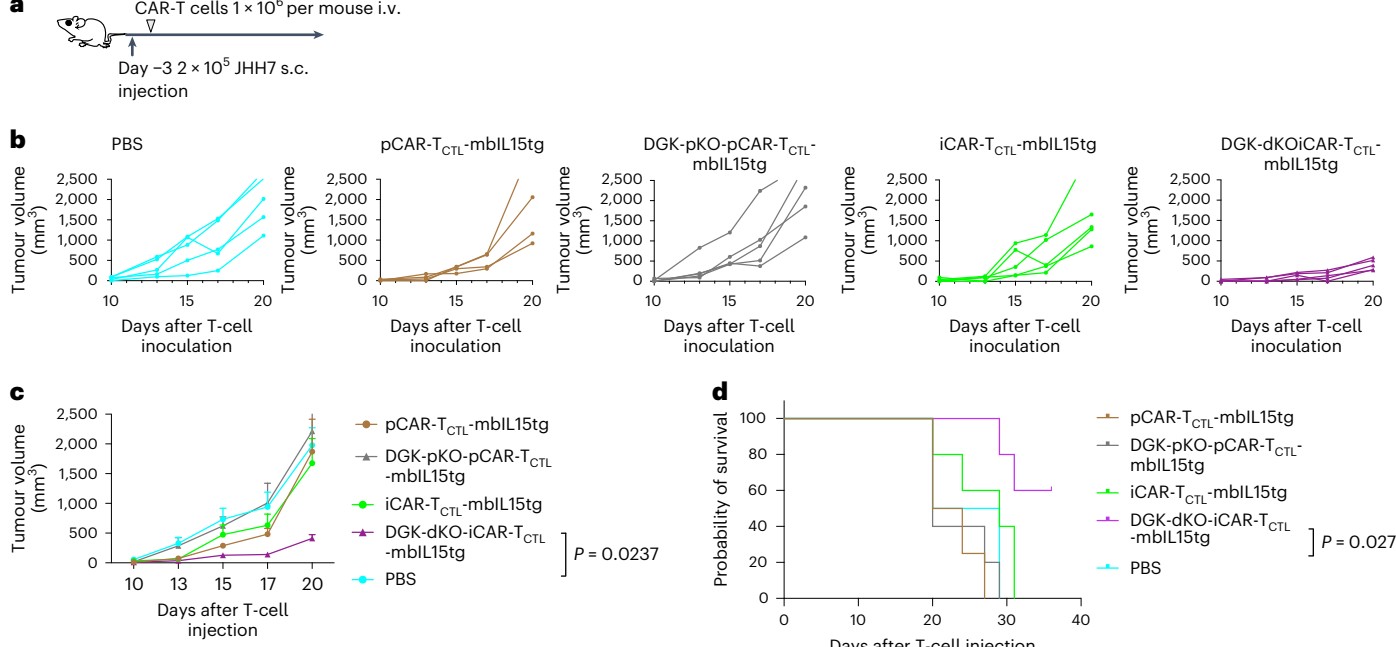

**Fig. 6 | Combinatorial signal enhanced iCAR-T$_{CTL}$ anti-tumour function.**
**a**, Treatment schedule of the liver cancer subcutaneous (s.c.) xenograft model. NSG mice were injected intraperitoneally with $2 \times 10^5$ JHH7 cells 3 days before treatment. In total, $1 \times 10^6$ iCAR-T$_{CTL}$-mbIL15tg, DGK-dKO-iCAR-T$_{CTL}$-mbIL15tg or pCAR-T$_{CTL}$-mbIL15tg were injected intravenously on day 0 ($n = 5$ for each group).

**b** Volume of inoculated JHH7 at indicated timepoints in individual mice treated with indicated test cells. **c**, Volume of inoculated JHH7 at indicated timepoints. Mean tumour size ± s.e.m. ($n = 5$ of each group), by two-way ANOVA. **d**, Survival of treated mice at indicated timepoints after injection. log-rank test with Benjamini–Hochberg multiple comparisons test.

iCAR-T$_{CTL}$ could form the basis for different CAR-modified regenerative T-cell immunotherapies.

## Discussion

We have recapitulated the differentiation process from iPSCs to acquire potent CD8αβ cytotoxic T cells in vitro to decipher the characteristics of CAR-modified iPS-T cells (iCAR-T cells) by engineering the lymphopoiesis pathway using co-stimulatory signalling component selection and enhanced iCAR-T cell effector function against solid tumours. To achieve this, we knocked out immunological checkpoint molecules and transduced cytokine/cytokine receptor chimaeric genes. These enhanced features led to better therapeutic effects in tumour-bearing animal models.

A tonic signal is attributed to the aggregation of CAR molecules that cause CAR-T exhaustion[12] and are reported to inhibit the expression of master transcription factor BCL11B by inhibiting the Notch signalling that affects the lymphopoiesis of CAR-modified haematopoietic stem and progenitor cells[14,22]. We investigated how the engineering of iPSCs impacts their differentiation propensity and found that the 28z construct decreased the differentiation efficiency of iPSCs to CD4CD8 DP cells through tonic signalling. In addition, we found that replacement with 4-1BBz or additional 4-1BB signalling to 28z attenuated the tonic signal during differentiation. This finding is consistent with that reported in the literature on primary CAR-Ts that an additional 4-1BB signal to 28z-based CAR restricted the downstream Zap70 phosphorylation at a basal level as well as after antigenic stimulation, thus preserving the therapeutic function by different affinity CARs[26]. Although the detailed mechanism of how additional 4-1BB signalling rescues T-cell differentiation from iPSC needs to be elucidated, we believe a mechanism compatible with that reported previously should be present. Certain genetic modifications on iPSCs for functional enhancement may inhibit differentiation. We observed decreased efficiency of T-cell differentiation from DGK dKO iPSC, possibly by enhanced duration and magnitude of TCR signalling, resulting in negative selection to a part of differentiating DP cells (Supplementary Fig. 1a), which is compatible with the previous observation in DGK-dKO mice[24]. Although the low differentiation efficiency decreased the collected T-cell yield from iPSCs, we could sufficiently expand the differentiated T cells through TCR stimulation[10]. We believe this is one of the advantages of using iPSC as the unlimited suppliable starting material. The IL-15-mediated signal is known to drive common lymphoid progenitor cells to NK cell progenitor in combination with AKT/mTOR signal activation[27]. In the presence of IL-15 during T-lineage cell differentiation from iPSCs, we observed NK-lineage-biased differentiation to CD4/CD8β double-negative NK progenitor cells[17]. This is compatible with previous observations in IL15tg mice[28]. Therefore, in this study, we modified matured iCAR-T cells by mbIL15 during TCR stimulation-mediated expansion after T-cell differentiation. Good temporal controls of these engineering steps could contribute to the large-scale production of therapeutic T cells as we indicated the concept of temporal control by using of DGKα-/-DGKζf/f iPSCs, which partially rescued the efficiency of T-cell differentiation (Supplementary Fig. 1a–d). In addition, this strategy could be applied to the temporal control of mbIL15tg, whereas other kinds of temporal control such as endogenous promoters (CD4, *TRAC* and so on) and synNotch would be available for further studies[29,30]. iPSCs can serve as the cell source for an infinite number of genetically engineered cells. This can be an advantage over peripheral blood T cells in terms of certainty, safety, and applicability for the mass production of cell therapies.

iCAR-T$_{CTL}$ retains the basic properties of CAR-T cells such as homing, proliferation and cytotoxicity at the tumour site. However, as exemplified by the results of ERK phosphorylation, signalling in the differentiated cell is generally attenuated than iCAR-T cells, although the underlying causes remain unclear. We consider this could be a result of the overall quality in each step of ex vivo manipulation to induce

haematopoietic progenitor cells, progenitor T cells, CD4CD8DP T cells, matured CD8ab T cells and so on. Therefore, each differentiation step needs to be improved physiologically, similar to the differentiating cells in our body. A recently reported method for iPS-T-cell differentiation using organoid culture could improve the quality of iCAR-T$_{CTL}$[31]. In contrast, we here report methods to improve the deficient functions by genetic manipulations. Thus, the optimization of signals 1–3 has an independent impact on iPSC-derived T cells in terms of their induction and function. Specifically, signal 1 impacts the proliferation and effector functions, signal 2 impacts the activation and tonic signalling and signal 3 impacts cell survival and persistency. A combination of DGK deletion and mbIL15 transduction is one of the examples of such modifications. Because of great advances achieved in CRISPR–Cas9-based human genome editing, several new clinical applications are under development, especially in the field of T-cell immunotherapy. For example, certain clinical trials using primary T cells with *TCR*- and/or *HLA*-knockout are known to diminish the alloreactivity of T cells[32,33], whereas *PD-1*- or *CTLA-4* knockout can overcome immunosuppression of the tumour microenvironment[34,35]. As iPSCs can be manipulated as a single-cell clone, it is possible to achieve 100% genome editing accuracy of the target without off-target effects. By targeting DGK, we demonstrated that gene editing can enhance its function by regulating the expression of intracellular molecules, which cannot be manipulated by antibody administration such as anti-PD1 and anti-CTLA4 antibodies. PD1 inhibition, the same checkpoint molecule, unexpectedly exerted no impact. However, it is known that PD1 expression does not increase in iPS-T cells without frequent stimulation[10]. Therefore, it is inferred that the response was not as strong as that of pCAR-T. Conversely, a combination of DGK deletion and mbIL15 transduction exerted a limited effect on T cells (Figs. 5 and 6); however, it was effective in boosting the function of less-responsive iCAR-T cells. In addition to the approach used in this experiment, several target genes exist that can improve each signal; it is important to optimize the combination of manipulations of these genes, especially for iPSC-derived T cells because they tend to have less responsiveness as mentioned above. It is possible that a more intense approach could be more useful than that for primary T cells.

Although cytotoxic ILCs (NK cells) are similar to CTLs in their ability to eliminate target cells, they differ substantially in the manner they sense target cells and in their kinetics[36]. It is reported that iPSC-derived T cells with properties similar to ILC/NK appear during T-cell differentiation from iPSCs. A direct comparison of iCAR-T$_{ILC}$ and iCAR-T$_{CTL}$ differentiated from identical CAR-iPSCs revealed their different properties that were compatible with those of their physiological counterparts, namely NK cells and CTLs. mbIL-15, which was found to be useful in iCAR-T$_{CTL}$ in this study, has been reported to be useful in cord-blood-derived NK cells, and in improving the persistency of CD19CAR iPS-T cells. In this study, the DGK deletion for iCAR-T$_{ILC}$ resulted in improved effector function comparable to DGK-dKO-iCAR-T$_{CTL}$, and mbIL-15 improved the persistency in vivo. However, the lack of improvement in the subcutaneous tumour accumulation suggested that these two modifications are insufficient for iCAR-T$_{ILC}$ and may indicate differences in properties from iCAR-T$_{CTL}$. However, iCAR-T$_{ILC}$ has an advantage in cytotoxicity, and the expression of mbIL15 from the iPSC stage may be advantageous for NK cell differentiation, which would be reflected in future clinical development[15].

Finally, we enhanced the therapeutic effect of iCAR-T therapy so that its effectiveness matched that of healthy-volunteer-derived CAR-modified CD8 T cells by modifying multiple T-cell signalling pathways. However, this approach could also increase the risk of unwanted immune reactions or tumourigenesis after administration of the cell therapy[37,38]. Suicide genes, such as *tEGFR*[39,40], *iCasp9* (ref. [41]) and *HSV-TK*[42], or antigen-specific induction systems such as syn-Notch receptors[30] could reduce these risks. Focusing on the risk of graft-versus-host disease caused by allogeneic TCR, deleting endogenous TCR genes in iPSCs by knocking in a pleiomorphic TCR such as gdTCR or iNKT-TCR, or knocking in CAR in the *TRAC* locus[29] reduce the risk of graft-versus-host disease. In addition, the risk of cell rejection from the host could be reduced by HLA matching and HLA editing the iPSCs[43]. Another challenge associated with iCAR-T-cell therapies is the generation of CD4 iCAR-T cells. CD4 CAR-T cells have important therapeutic effects[44]. Although the mechanism is not completely understood, the induction of T cells expressing CD4 from iPSCs has been reported in three-dimensional culture[45]. These technologies demonstrate the potential of iPSCs for the production of effective and safe allogeneic cancer immunotherapies.

In summary, several genetic modifications have been attempted to enhance the therapeutic effect of CAR-T therapies, and the artificial design of T cells is being pursued as a cancer immunotherapy strategy against solid tumours. iPS-T cells are feasible for combinatorial signalling that can enhance iCAR-T therapies for solid tumours.

## Methods

### Mice
Mice used in this study were 6- to 12-week-old female NOD-SCID IL2Rγc$^{null}$ (NSG) mice purchased from Oriental Bio. The mice were housed under controlled conditions, humidity and light/dark cycle in a specific-pathogen-free facility. All animal experiments were performed in accordance with the Ethical Review Body at Kyoto University.

### Cell lines
HepG2 and JHH7 cell lines were purchased from the JCRB Cell Bank. KOC7c, SK-Hep-1, SK-Hep-1 transduced with GPC3, and K562 were provided by the National Cancer Institute, Japan. The mycoplasma status of the cells was routinely checked. JHH7, HepG2, KOC7c and SK-Hep-1 were maintained in Dulbecco's modified Eagle medium supplemented with 10% foetal bovine serum. K562 cells were maintained in Roswell Park Memorial Institute 1640 medium supplemented with 10% foetal bovine serum. The cells were cultured under 5% CO$_2$ at 37 °C.

TKT3v1-7, an iPSC line established from the T cells of a healthy donor, was maintained on mouse embryonic fibroblasts in a human iPSC medium (Dulbecco's modified Eagle medium/F12 FAM supplemented with 20% knockout serum replacer, 2 mM L-glutamine, 1% non-essential amino acids, 10 mM 2-mercaptoethanol and 5 ng ml$^{-1}$ basic fibroblast growth factor (bFGF)) or iMatrix-511 (Matrixome)-coated plates in StemFit AK03 medium (Ajinomoto) at 5% CO$_2$ and 37 °C.

### Generation of CAR-engineered iPSCs
The composition of GC33-CD28-41BB-T2A EGFR was designed by the authors and produced by GenScript. The constructs were subcloned into the lentivirus vector CS-UbC-RfA-IRES-hKO1, which contains the hKO1 fluorescent protein gene with an internal ribosomal entry site (IRES) and is under the control of the human ubiquitin C (UbC) promoter. For the inducible CAR construct, we used the CD28 transmembrane domain[46]. The recombinant lentiviral vector was produced using the method described previously[6]. TKT3v1-7 was transduced with a lentiviral vector by spin infection in 24-well plates. hKO1-positive iPSCs were FACS sorted, cloned by limiting dilution and expanded.

### Generation of iPSCs expressing GFPluc2 and iPSCs with DGK disrupted by CRISPR–Cas9
We designed specific single-guide RNA (sgRNA) for exon 7 of DGKa and exon 2 of DGKz. To construct custom sgRNA expression vectors, two oligos containing the sgRNA target site and a universal reverse primer were PCR amplified and cloned into the BamHI-EcoRI site of the pHL-H1-ccdB-mEFα-RiH vector (addgene #60601), which used the H1 promoter to drive sgRNA. We transferred nuclease-expressing plasmids (5 μg of pHL-EF1α-SphcCas9-iP (addgene #60599) and 5 μg of guide RNA-expressing vector to disrupt DGK or 1 μg of pXAT2 (addgene #80494) and 3 μg of pAAVS1-P-CAG-GFPluc2 (addgene #80493) to

introduce GFPluc2) (ref. [47]) into $1 \times 10^6$ cells by electroporation using a NEPA 21 electroporator (poring pulse: pulse voltage, 175 V; pulse width, 2.5 ms; pulse number, 2; Negagene). To disrupt DGK, iPSC clones having non-sense mutations in both *DGKα* and *DGKζ* were selected. To introduce the luciferase gene, GFPluc2-positive iPSCs were purified by puromycin selection and fluorescence cell sorting. To establish DGKz flox iPSCs, we designed specific sgRNA for intron 6 and intron 11 of DGKz and loxP-ssODN, which has a loxP sequence sandwiched between the homology arms for each site[48]. Ten micrograms of recombinant *Saccharomyces pyogenes* Cas9 (Thermo Fisher Scientific) and 5 μg of chemically synthesized sgRNAs were incubated for 20 min before the electroporation to generate Cas9-gRNA ribonucleoprotein (RNP) complexes. A total of $5 \times 10^5$ cells were resuspended in the MaxCyte electroporation buffer and mixed with Cas9-gRNA RNP complexes and 5–10 μg of loxP-ssODN. Electroporation was performed with a MaxCyte STX.

### T-cell differentiation from CAR-iPSCs

To differentiate iPSCs into T cells, we used a previously reported protocol[6] with slight modifications. First, CAR-iPSCs were differentiated into haematopoietic precursors through SAC formation on feeder cells or through feeder-free embryoid body formation. For SAC formation, clumps of iPSCs were transferred onto C3H10T1/2 feeder cells and cultured in the embryoid body medium (EB medium; StemPro-34, Invitrogen, 2 mM L-glutamine, 400 μM monothioglycerol, 50 μg ml⁻¹ ascorbic acid-2-phosphate and insulin–transferrin–selenium supplements) containing rhVEGF (R&D). On day 7, rhSCF (Peprotech) and rhFlt-3L (Peprotech) were added to the culture. For the formation of feeder-free embryoid body, undifferentiated T-iPSC colonies were treated with TrypLE Select (Gibco) for 8 min and transferred to low-attachment plates to allow for the formation of embryoid bodies in Stemfit AK03N containing 10 μM Y-27632 overnight. These embryoid bodies were collected, transferred to StemPro-34 (Thermo Fisher Scientific) supplemented with 10 ng ml⁻¹ penicillin–streptomycin (Sigma), 2 mM GlutaMAX (Thermo Fisher Scientific), 50 μg ml⁻¹ ascorbic acid-2-phosphate (Sigma), $4 \times 10^{-4}$ M monothioglycerol (Nacalai) and 1× insulin–transferrin–selenium solution (ITS-G, Thermo Fisher Scientific) (referred to as EB basal medium), 50 ng ml⁻¹ recombinant human (rh) BMP-4 (R&D Systems), 50 ng ml⁻¹ rhVEGF (R&D Systems) and 50 ng ml⁻¹ bFGF (FujiFilm Wako). After 24 h, 6 μM SB431542 (FujiFilm Wako) was added. On day 4, the embryoid bodies were collected, transferred to EB basal medium supplemented with a cocktail of haematopoietic cytokines (50 ng ml⁻¹ rhVEGF, 50 ng ml⁻¹ rhbFGF and 50 ng ml⁻¹ rhSCF (R&D Systems)). On day 7, the embryoid bodies were collected, transferred to EB basal medium supplemented with 50 ng ml⁻¹ rhVEGF, 50 ng ml⁻¹ rhbFGF, 50 ng ml⁻¹ rhSCF, 30 ng ml⁻¹ rhTPO (PeproTech) and 10 ng ml⁻¹ FLT3L (PeproTech). These differentiated cells on days 8–14 of culture were transferred onto Delta-like 1-expressing OP9 (OP9-DL1) feeder cells or FcDLL4-coated plates and cultured in the presence of a cocktail of T-cell-lineage cytokines (10 ng ml⁻¹ hFlt3L and 5 ng ml⁻¹ IL-7). After 21–28 days of culture, haematopoietic cells were differentiated into CD3⁺ T-cell lineages. T-cell lineages were collected and stimulated with 500 ng ml⁻¹ OKT3 (Miltenyi Biotech) in a-MEM (Sigma) supplemented with 15% foetal bovine serum (Thermo Scientific), 2 mM L-glutamine (Thermo), 100 U ml⁻¹ penicillin (Thermo Scientific) and 100 ng ml⁻¹ streptomycin (Thermo Scientific) in the presence of 10 ng ml⁻¹ rhIL-7 and 10 nM dexamethasone (Dexart, Fuji Pharma). To generate iCAR-T, the generated CD5CD8β DP T cells were sorted and expanded by mixing them with irradiated peripheral mononuclear cells (PBMCs) and co-culturing in the α-MEM medium in the presence of 10 ng ml⁻¹ IL-7, 5 ng ml⁻¹ IL-15 (Peprotech) and 2 μg ml⁻¹ phytohemagglutinin (PHA) (Sigma). For functional analysis, we purified CD5CD8β DP T cells before expanding and maintaining them as the major population. To generate iCAR-T$_{ILC}$, T-cell lineages were collected and directly expanded with PHA.

### Generation of mbIL-15-transduced CAR-T cells

RD18 cells that can produce a retroviral vector encoding mbIL15-T2A-NGFR were a kind gift from Dr Kazuhide Nakayama of Takeda Pharmaceuticals. CAR-T cells were stimulated with Dynabeads Human T Activator anti-CD3/28 (Thermo Scientific). Three days after the stimulation, the cells were transduced with a retroviral vector encoding mbIL15-T2A-NGFR. NFGR-positive cells were purified by FACS.

### Generation of DGK-deleted primary T cells

Primary T cells were stimulated with Dynabeads Human T Activator anti-CD3/28 (Thermo Scientific). Four days after the stimulation, electroporation was performed using an Amaxa P3 Primary Cell Kit and 4D-Nucleofecter (Lonza). Ten micrograms of recombinant *S. pyogenes* Cas9 (Thermo Fisher Scientific) and 1.25 μg of chemically synthesized sgRNAs for DGKa and DGKz were incubated for 20 min before electroporation to generate Cas9-gRNA RNP complexes. A total of $5 \times 10^5$ cells were resuspended, and P3 buffer was added to the pre-incubated Cas9-gRNA RNP complexes. Cells were nucleofected using the program EO-115. One week after the electroporation, the cells were collected and the genomic DNA was isolated. The frequency of indels was calculated by Tracking of Indels by Decomposition (TIDE) analysis.

### FCM analysis

Stained cell samples were analysed using an LSR or FACS AriaII Flow Cytometer (BD Biosciences), and the data were processed using FlowJo (Tree Star). All human cells were first gated on FSC/SSC according to cell size and granularity, using stained human PBMCs as a positive control and reference for cell size, granularity and staining intensity. Unstained samples were used to set up negative gates, and stained human PBMCs were used to set up positive gates. Dead cell populations were excluded using propidium iodide staining (Supplementary Fig. 8). For T-cell phenotyping, the following antibodies were used: CD3-eFluor 450 (clone UCHT1, eBioscience), CD3-APC-Cy7 (clone UCHT1, BioLegend), CD3-APC (clone UCHT1, BioLegend), CD4-Brilliant Violet 421 (clone OKT4, BioLegend), CD4-Brilliant Violet 510 (clone OKT4, BioLegend), CD5-FITC (clone UCHT2, eBioscience), CD5-PE-Cy7 (clone UCHT2, eBioscience), CD7-APC (clone CD7-6B7, BioLegend), CD8a-PerCP-Cy5.5 (clone SK1, BioLegend), CD8b-PE-Cy7 (clone SIDI8BEE, eBioscience), CD8b-APC (clone 2ST8.5H7, BD), CD19-PE (clone HIB19, BD), CD25-FITC (clone BC96, BioLegend), CD27-APC (clone O323, BioLegend), CD28-Brilliant Violet 421 (clone CD28.2, BioLegend), CD45-Brilliant Violet 510 (clone HI30, BioLegend), CD45RA-Brilliant Violet 510 (clone HI100, BioLegend), CD45RO-APC-Cy7 (clone UCHL1, BioLegend), CD56-APC-Cy7 (clone HCD56, BioLegend), CD62L-PE-Cy7 (clone DREG-56, BioLegend), CD69-PacificBlue (clone FN50, BioLegend), CD94 (NKG2A)-Brilliant Violet 421 (clone HP-3D9, BD Biosciences), CD159a (KLRC)-PacificBlue (clone S19004C, BioLegend), CD161 (KLRB)-PE-Cy7 (clone HP-3G10, BioLegend), CD197 (CCR7)-APC (clone G043H7, BioLegend), CD226 (DNAM)-APC (clone 11A8, BioLegend), CD247 (pY142)-Alexa Fluor 647 (clone K25-407.69, BD), CD314 (NKG2D)-PE-Cy7 (clone 1D11, BioLegend), CD335 (NKp46)-FITC (clone 900, BioLegend), CD336 (NKp44)-APC (clone P44-8, BioLegend), CD337 (NKp30)-APC (clone P30-15, BioLegend), EGFR-Brilliant Violet 421 (clone AY13, BioLegend), ERK1/2 (pT202/pY204)-Alexa Fluor 647 (clone 20 A, BD Biosciences), mouse IgG2a k-Alexa Fluor 647 (clone MOPC-173, BD Biosciences), TCRab-FITC (clone WT31, eBioscience) and TIGIT-PerCP-eFluor 710 (clone MBSA43, eBioscience).

### Western blot analysis

iCAR-T cells derived from wild-type iPSCs, DGKα KO iPSCs, DGKζ KO iPSCs and DGK-dKO iPSCs were used to detect DGKα or DGKζ by capillary electrophoresis using the Protein Simple Wes System. Anti-DGKα (Abcam, AB88672-50) and anti-DGKζ (Abcam, AB88672-50) primary antibodies were incubated with the sample, followed

by incubation with the secondary antibody. ACTB was used as the loading control. Data were processed using the Compass software (Protein Simple).

## Cytokine release, 51Cr release assays and non-RI cytotoxic assay

T-cell cytokine secretion was conducted using the Human Th1/Th2/Th17 Kit (BD Bioscience). 51Cr release assays were performed to evaluate the cytolytic ability of effector cells. Target tumour cells were loaded with 1.85 MBq 51Cr for 1 h, and afterward, 5,000 tumour cells were co-incubated with effector cells for 5 h at effector-to-target (E:T) ratios ranging from 2.5:1 to 20:1. Supernatants were collected, and 51Cr release was quantified using a beta counter. The percentage lysis was calculated as lysis (%) = (experimental lysis − spontaneous lysis)/(maximal lysis − spontaneous lysis) × 100%, where maximal lysis was induced by incubation in 2% Triton X-100 solution. For the non-RI cytotoxic assay, target cells were labelled with N-SPC non-radioactive cellular cytotoxicity assay kit (Techno Suzuta) according to the manufacturer's instructions. The target cells were pulsed with the BM-HT reagent at 37 °C and washed thrice, and $5 \times 10^3$ cells were seeded into a well of a 96-well plate. Effector cells were loaded into the wells at E:T ratios ranging from 2.5:1 to 20:1 and co-cultured for 4 h. Next, 20 µl of the co-culture supernatant was mixed with 100 µl of Eu solution, and time-resolved fluorescence was measured through the EnVision 2105 multimode plate reader (PerkinElmer). Percentage lysis was calculated as lysis (%) = (experimental lysis − spontaneous lysis)/(maximal lysis − spontaneous lysis) × 100%, where maximal lysis was induced by incubation in a detergent solution provided by the manufacturer.

## ³H-thymidine uptake proliferation assay and cell trace reagent proliferation assay

The proliferation of T cells was measured using the ³H-thymidine uptake assay. In total, $1 \times 10^5$ effector cells were co-cultured with irradiated $1 \times 10^4$ target cells for 3 days. Next, ³H was pulsed, and 16 h later, ³H-thymidine was measured by a beta counter. For cell trace reagent proliferation assay, effector cells are stained with 1 µM CellTrace Violet before stimulating them with target cells. Three days after stimulation, the decline in fluorescence was evaluated using FCM.

## Bioluminescence imaging

Bioluminescence was detected using the Xenogen IVIS Imaging System (Xenogen), as previously described[6,16]. We performed imaging for 10–15 min after the intraperitoneal injection of VivoGlo Luciferin (3 mg per mouse, Promega).

## Analysis of tumour-infiltrating CAR-T cell model

On day 14, $5 \times 10^6$ SK-Hep-1 cells transduced with GPC3 were injected subcutaneously at the left flank of NSG mice. Luciferase-knock-in CAR-T cells were intravenously injected on day 0. Mice were killed on day 7, and subcutaneous tumours were collected. Each tumour was divided into two fractions. One fraction was extensively minced with a razor and digested gently with a MACS Dissociator. Next, the fraction was filtered through 70 mm nylon mesh cell strainers, and red blood cells were lysed. Single-cell suspensions were stained with fluorochrome-conjugated antibodies as indicated. The other fraction was prepared for histopathological analysis.

## Subcutaneous tumour animal model 1

On day 14, $5 \times 10^6$ SK-Hep-1 cells transduced with GPC3 were injected subcutaneously at the left flank of NSG mice, and $5 \times 10^6$ SK-Hep-1 cells without transduction were injected subcutaneously at the right flank. Luciferase-knock-in CAR-T cells were intravenously injected on day 0. Luciferase activity from T cells was measured by in vivo bioluminescence imaging.

## Subcutaneous tumour animal model 2

A total of $2 \times 10^5$ JHH7 cells were injected subcutaneously 3 days before the treatment, and $1 \times 10^7$ or $1 \times 10^6$ CAR-T cells were injected intravenously on days 0 and 7 or day 0 only. Tumour dimensions were measured with calipers, and tumour volumes were calculated using the formula $V = LW^2/2$, where $L$ is length (longest dimension) and $W$ is the width (shortest dimension).

## Peritoneal dissemination tumour animal model

A total of $5 \times 10^5$ KOC7c cells were injected intraperitoneally on day 0, and $5 \times 10^6$ CAR-T cells were intraperitoneally injected twice per week for 2 weeks from day 3 or $10^6$ CAR-T cells were intraperitoneally injected on day 3. Luciferase activity from T cells or tumour burden was measured by in vivo bioluminescence imaging (IVIS 100 Imaging System, Caliper).

## Nalm6 leukaemia model

A total of $5 \times 10^5$ Nalm6 cells were injected intravenously on day 0, and $1 \times 107$ CD19BBz-iCAR-$T_{CTL}$ and CD19BBz-DGK-dKO-iCAR-$T_{CTL}$-mbIL15tg were injected intravenously on days 4. Luciferase activity from T cells was measured by in vivo bioluminescence imaging (IVIS 100 Imaging System, Caliper). Mouse blood cells were collected on day 23. Nalm6 (hCD45$^+$ CD19$^+$) cells were analysed by FCM.

## RNA sequencing of iCART$_{CTL}$ in tumours or spleens of xenograft models

Libraries for RNA sequencing were prepared using the SMART-seq v3 Ultra Low Input RNA Kit for Sequencing according to the manufacturer's instructions. Briefly, 10–20 cells were lysed, followed by reverse transcription and amplification. The libraries were sequenced for 96 cycles and 8 bp dual-index using the HiSeq SR Cluster Kit v2 (Illumina) and HiSeq Rapid SBS Kit v2 on a HiSeq2500 operated using the HiSeq Control Software 2.2.58. All sequence reads were extracted in the FASTQ format using the BCL2FASTQ Conversion Software 1.8.4 in the CASAVA 1.8.2 pipeline. The sequence reads were mapped to the hg19 reference genome, downloaded on 10 December 2012, using TopHat v2.0.8b, and quantified using RPKMforGenes. Hierarchical clustering was conducted using the Euclidian distance and the ward method of the hclust function in R3.2.1.

## Whole transcriptome sequencing using the Ion AmpliSeq Transcriptome Human Gene Expression Kit

Total RNA was extracted from 100,000 cells using the NucleoSpin RNA Kit (Macherey-Nagel, 740902.250) according to the manufacturer's instructions. A total of 1 ng of RNA was reverse transcribed into complementary DNA using the SuperScript VILO cDNA Synthesis Kit (Life Technologies, 11754050). Next, the cDNA was amplified using the Ion AmpliSeq Transcriptome Human Gene Expression Core Panel. Amplified libraries were evaluated for quality using an Agilent 2100 Bioanalyzer (Agilent Technologies) and quantified by qPCR using the Ion Library Quantitation Kit (Life Technologies, 4468802). The templated libraries were loaded onto an Ion 540 Chip (Life Technologies, A27765) using the Ion Chef System and subsequently sequenced on an Ion 5SXL.

## Read alignment and differential gene expression analysis

The AmpliSeq sequencing data were analysed using the ampliSeqRNA plugin available in Ion Torrent sequencing platforms. This plugin utilizes the Torrent Mapping Alignment Program, which aligns raw Ion Torrent sequencing reads against a custom reference sequence containing the targeted transcripts included in the AmpliSeq gene expression kit. Differential gene expression analysis was performed using R after quality control, which included counts per million conversion, log transformation and the filtering of lowly expressed genes. Normalization was performed using the TMM normalization method. DEG analysis between two groups was performed using the voom

function provided with the limma package v3.42.0. DEGs were determined using a $P$ value less than 0.05 and a fold change greater than |2.0|. The Benjamini–Hochberg procedure was used to adjust multiple comparisons.

Enrichment analysis was performed using clusterProfiler v3.14.3 and ReactomePA v1.30.0 with upregulated and downregulated genes, respectively. Fisher's method was used to combine $P$ values from each test, and the combined values were adjusted for multiple comparisons using the Benjamini–Hochberg procedure. A $P$ value less than 0.05 was considered significant for GO terms of the Reactome pathway.

## Statistics

All data are presented as mean ± standard error of the mean (s.e.m.) unless otherwise noted. All statistics were performed using Prism (GraphPad software). Analysis of variance (ANOVA) and Tukey's multiple comparison tests were used to compare multiple groups, and two-sided Student's $t$-test was used to compare two groups in parametrical data. Values of $P < 0.05$ were considered significant.

## Reporting summary

Further information on research design is available in the Nature Portfolio Reporting Summary linked to this article.

## Data availability

The primary data supporting the results in this study are available within the paper and its Supplementary Information. The raw and analysed datasets generated during the study are too large to be publicly shared, yet they are available for research purposes from the corresponding author on reasonable request. Source data are provided with this paper.

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

## Acknowledgements
We thank S. Yamanaka (Kyoto Univ.) for critical advice, H. Nakauchi (Univ. of Tokyo) for providing the iPSC lines, H. Miyoshi (Keio Univ.) for providing the lentiviral vectors, Y. Mishima, H. Yano (Kyoto Univ.), S. Kitayama, A. Tanaka, M. Yoshida (Kyoto Univ.), A. Kumagai, S. Kamibayashi, E. Imai, H. Takakubo, T. Ishii, K. Makino and K. Noda (Kyoto Univ.) for technical assistance, and P. Karagiannis (Kyoto Univ.) for editing the manuscript. Part of this work was made possible by support from K. Nakayama (Takeda-CiRA Joint Program (T-CiRA)), who provided the retrovirus vector carrying the *mbIL-15* gene, Y. Kassai (Takeda Pharmaceutical Company), X. Song (Orizuru Therapeutics, Inc.), S. Asano and A. Mitsui (Axcelead Drug Discovery Partners Inc.), who conducted the gene expression analysis, and all staff of the Kaneko project and T-CiRA. The entire study was conducted in accordance with the Declaration of Helsinki and permitted by the institutional ethics board and Animal Care and Use Committee of Kyoto University. This work was supported in part by Japan Agency for Medical Research and Development (grant numbers JP15cm0106115, JP18ck0106204 and JP22bm0104001) and the National Cancer Center Research Fund (grant number 28-A-8).

## Author contributions
T.U. and S.K. designed the study, interpreted the data and wrote the manuscript; T.U., S.I., K.O., B.W. and S. Shiina performed the experiments; T.U., S.I. and S.K. analysed the data; R.K. and S. Sakamoto performed the genetic experiments and analysed the data; A.W. interpreted and supervised the data of genetic experiments; Y.K., A.M., S.T., Y.U., A.H., K.W., Y.K., H.S., T.N., H.X. and K.T. provided critical materials, protocols and advice for conducting experiments; S.K. supervised the study.

## Competing interests
S.K. is a founder, shareholder and director of Thyas Co., Ltd., and received research funding from Takeda Pharmaceutical Co., Ltd., Kirin Co., Ltd., Astellas Co., Ltd., Terumo Co., Ltd., Mitsui-soko Co., Ltd., KOTAI Biotechnologies, and Thyas Co., Ltd. K.T. is chief executive officer of Noile-Immune Biotech Inc. and receives research funding from it. The remaining authors declare no competing financial interests.

## Additional information
**Extended data** is available for this paper at https://doi.org/10.1038/s41551-022-00969-0.

**Correspondence and requests for materials** should be addressed to Shin Kaneko.

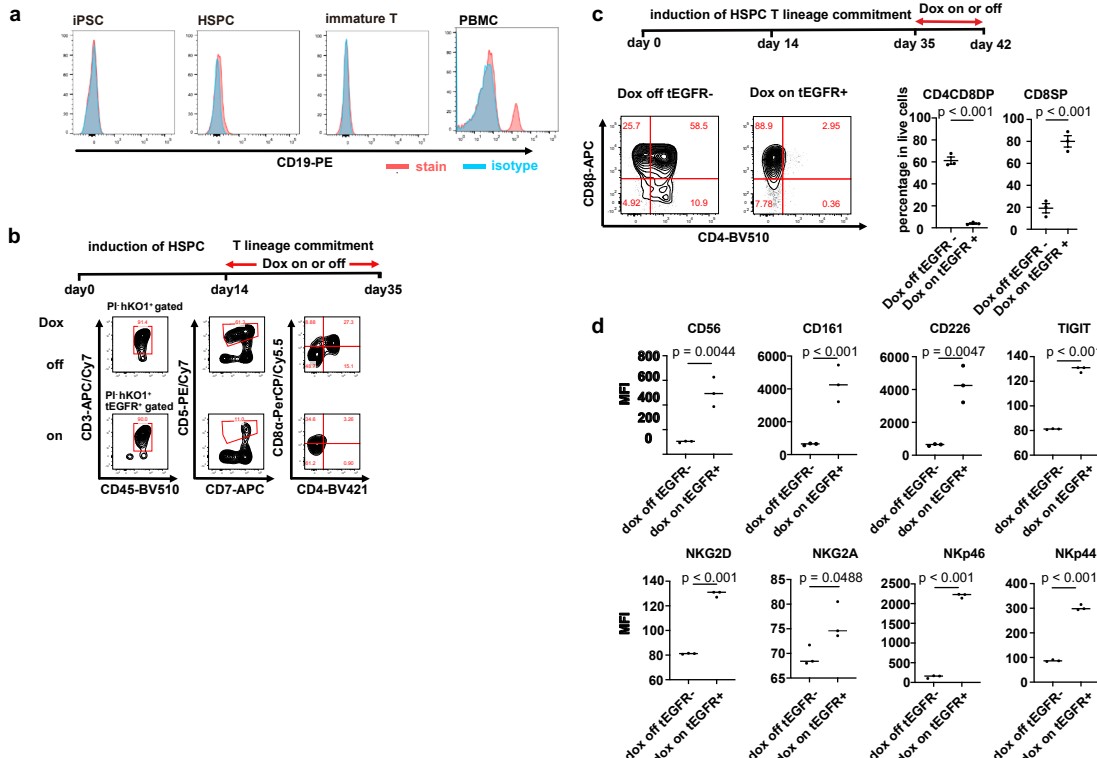

**Extended Data Fig. 1 | The CAR construct impacts the differentiation of CAR-iPSCs into T-cell lineages. a.** CD19 expression on iPSCs, hematopoietic progenitor cells, and T-cell lineages in the differentiation process. The CD19 expression on PBMCs is indicated as a reference (right). **b.** Hematopoietic progenitor cells derived from inducible 1928z CAR-transduced T-iPSCs were divided into two groups and subsequently differentiated into immature iCAR-T in the presence (2 µg/mL) or absence of doxycycline (Dox). Surface antigen expression of CD3, CD45, CD5, CD7, CD4, CD8a from the two groups is shown. HSPC, hematopoietic stem progenitor cells. **c,d.** Inducible 1928z CAR-transduced T-iPSCs were differentiated into immature iCAR-T cells without Dox and cultured for 1 more week in the presence or absence of Dox. The expression profile of CD4 and CD8β (**c**) and o NK-cell-related surface antigens (**d**) are indicated. $N = 3$, mean ± SEM, statistical tests. Two-sided Student's t test. MFI, mean fluorescence intensity, SEM, standard error of the mean.

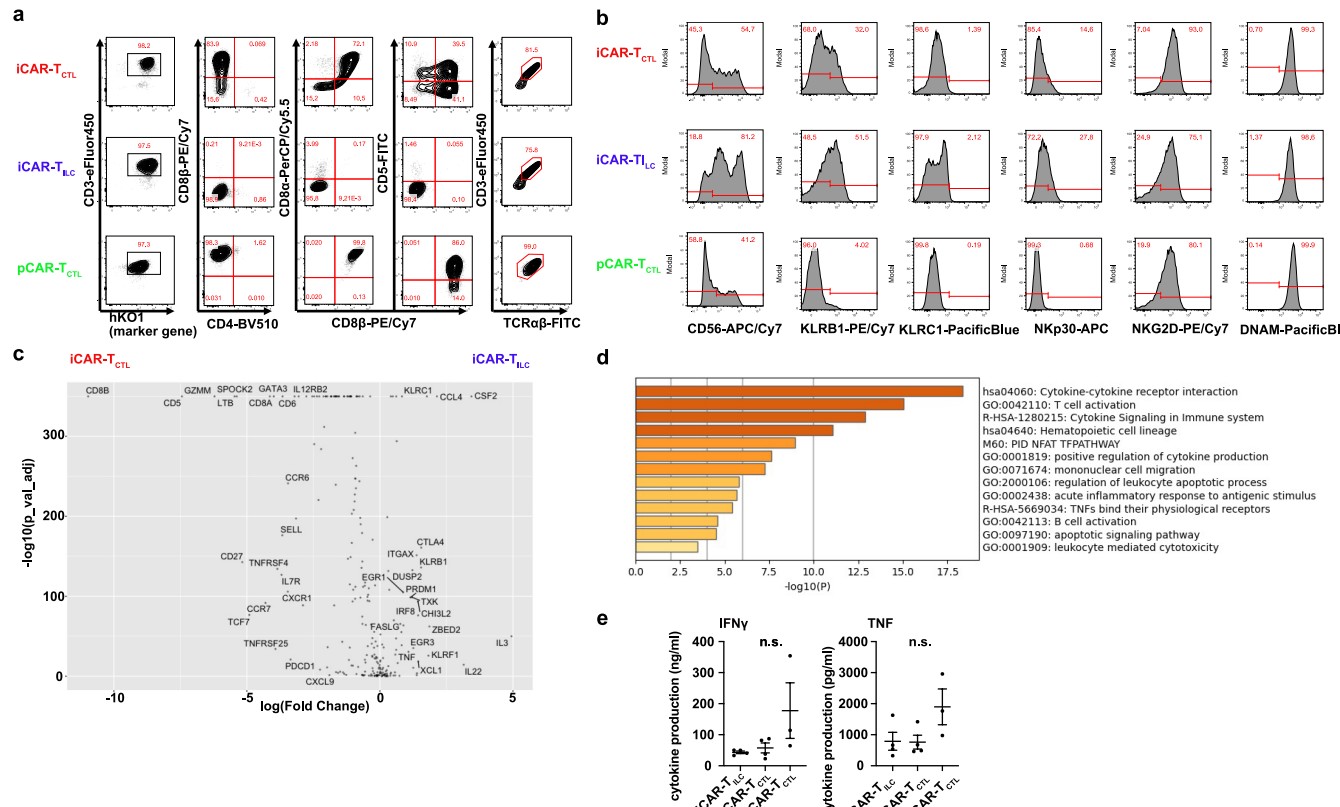

**Extended Data Fig. 2 | GPC3 CAR-iPSCs generated two types of iCAR-T cells with distinct phenotypes. a.** Representative flow cytometry plots showing T cell lineage-related surface antigen profiles of iCAR-T$_{CTL}$, iCAR-T$_{IL}$, and pCAR-T$_{CTL}$. **b.** Flow cytometry plots showing the expression of surface markers associated with the NK cell lineage on iCAR-T$_{CTL}$, iCAR-T$_{ILC}$, and pCAR-T$_{CTL}$. **c.** Volcano plot of DEGs between iCAR-T$_{CTL}$ and iCAR-T$_{ILC}$. **d.** The list of upregulated genes in iCAR-T$_{CTL}$.

Pathway and process enrichment analysis was performed using online tools in Metascape (http://metascape.org/)[49]. **e.** Cytokine production of iCAR-T$_{CTL}$, iCAR-T$_{ILC}$, and pCAR-T$_{CTL}$ 48 h after co-culturing with irradiated SK-Hep-GPC3 ($n = 3$, mean ± SEM). ns, not significant. One-way ANOVA with Tukey's multiple comparisons test.

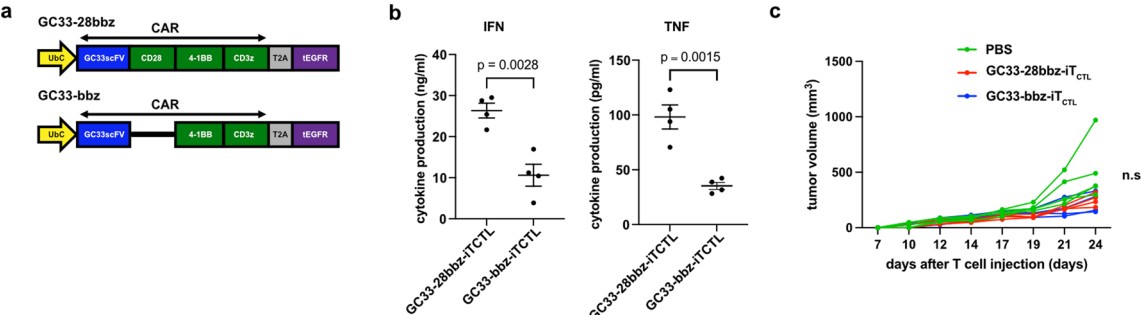

**Extended Data Fig. 3 | GC33-bbz-CAR-transduced-iPS-T cells did not show a remarkable difference from GC33-28bbz-CAR-transduced-iPS-T cells in tumor suppressive function. a**. Schematic presentation of CAR constructs GC33-28bbz and GC33-bbz CAR **b**. Cytokine production of GC33-28bbz and GC33-bbz CAR-transduced-iPS-T cells 48 h after co-culturing with irradiated SK-Hep-GPC3 ($n$ = 4, mean ± SEM). Two-sided Student's t test. **c**. NSG mice were injected subcutaneously with $2 \times 10^5$ JHH7 cells 3 days before the treatment. In total, $1 \times 10^7$ GC33-28bbz-iT$_{CTL}$ or GC33-bbz-iT$_{CTL}$ were injected intravenously on days 0 and 7 ($n$ = 4 for each group). Tumor volume of the inoculated JHH7 at the indicated time points in individual mice treated with indicated test cells. n.s., not significant; SEM, standard error of the mean.

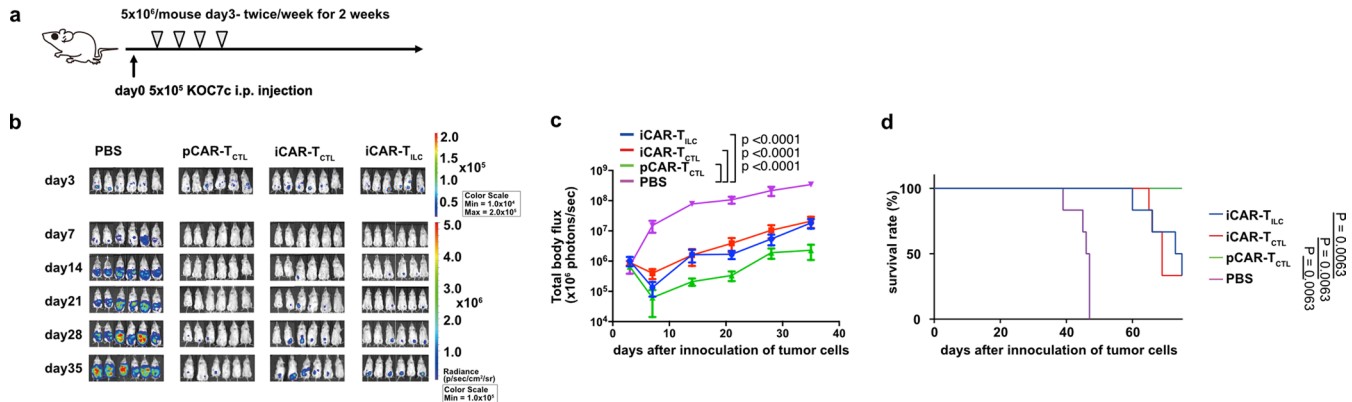

**Extended Data Fig. 4 | iCAR-T$_{CTL}$ suppressed tumor progression but did not reach pCAR-T$_{CTL}$. a–d.** Treatment schedule of the ovarian cancer peritoneal dissemination xenograft model (**a**) NSG mice were injected intraperitoneally with $5 \times 10^5$ KOC7c cells expressing luciferase. From the third day after the ovarian cancer inoculation, $5 \times 10^6$ iCAR-T$_{CTL}$, iCAR-T$_{ILC}$, or pCAR-T$_{CTL}$ were injected intraperitoneally twice a week for 2 weeks ($n = 6$ for each group). *In vivo* bioluminescence imaging of luciferase-labeled KOC7c in NSG mice treated with iCAR-T$_{ILC}$, iCAR-T$_{CTL}$, or pCAR-T$_{CTL}$ (**b**). Change in the total body flux as the total tumor volume (lower left panel, mean ± SEM) (**c**) and survival (lower right panel) were evaluated at the indicated time points after the injection (**d**). One-way ANOVA with Tukey's multiple comparisons test and log-rank test with Bonferroni multiple comparisons test.

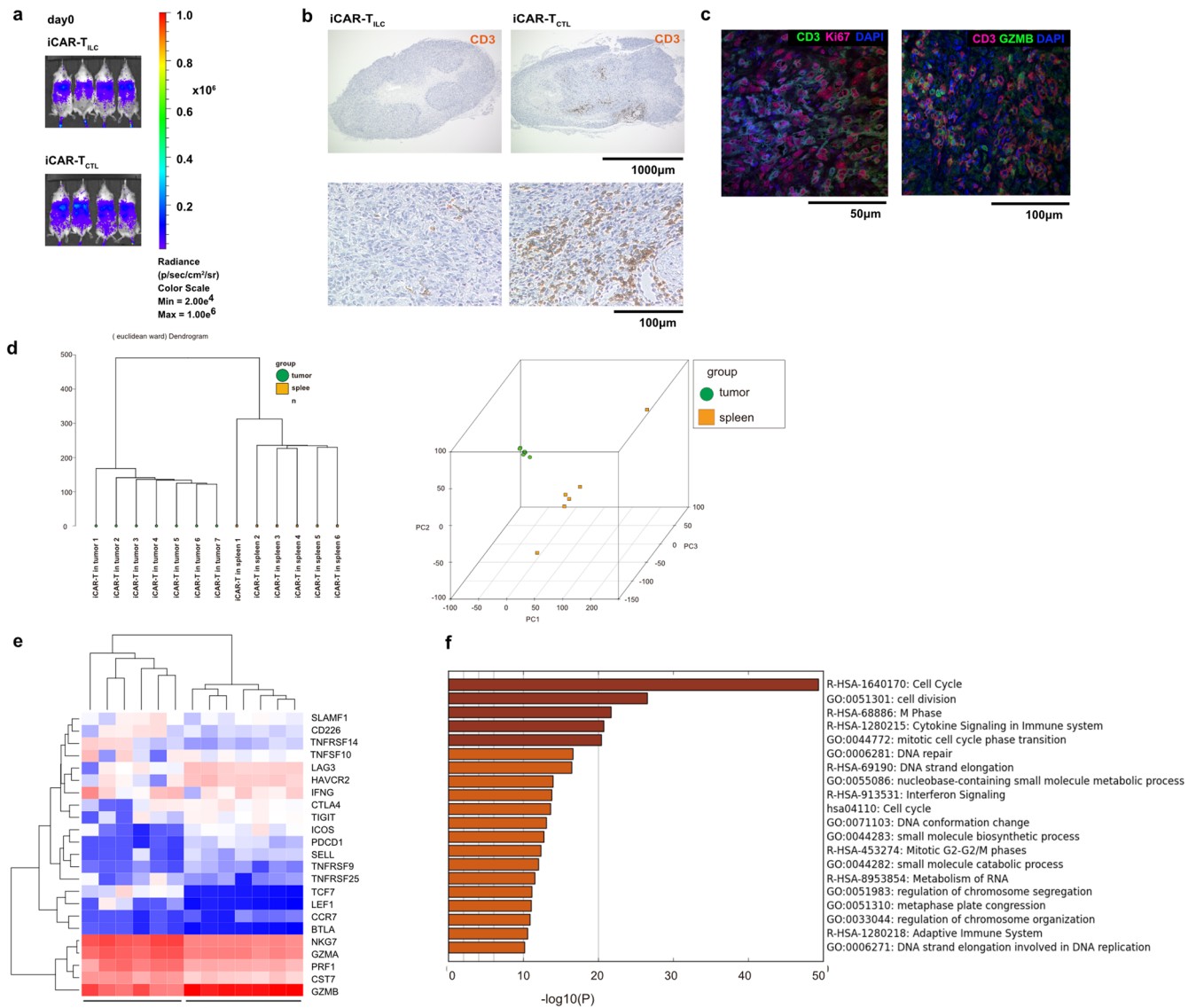

**Extended Data Fig. 5 | Adaptive type regenerated T cells derived from GPC3 CAR-iPSCs have the superior capability to accumulate at the tumor site in vivo than innate type regenerated T cells. a.** *In vivo* imaging of mice into which iCAR-T$_{CTL}$ and iCAR-T$_{ILC}$ are injected intravenously from the tail vein on day 0. **b.** Representative immunohistochemical staining of dissected SK-Hep-GPC3 tumors treated with the indicated iCAR-T cells. The migration of injected cells was detected by an anti-human CD3 antibody (clone EP41) (*n* = 3). **c.** Representative immunofluorescence staining of dissected tumors treated with iCAR-T$_{CTL}$ with anti-human CD3 antibody (green) and anti-human Ki-67 antibody (red) (left panel) or anti-human CD3 antibody (red) and anti-human Granzyme B antibody (green) (right panel). Cell nuclei were stained with DAPI (blue) (*n* = 3).

**d–f.** RNA expression profiles of iCAR-T$_{CTL}$ were isolated from the tumor or spleen after the first week of administration. Dendrogram of iCAR-T$_{CTL}$ isolated from the tumor and spleen (*n* = 6–7) (**d**). Principal component analysis (PCA) of iCAR-T$_{CTL}$ isolated from the tumor and spleen. To build the list of upregulated genes in tumor-infiltrated iCAR-T$_{CTL}$, pathway and process enrichment analysis was performed using online tools in Metascape (http://metascape.org/)[49]. Heatmap of gene expression in iCAR-T$_{CTL}$ in the tumor or spleen by RNA expression profiling of memory-, cytotoxicity-, co-stimulatory- and activation/exhaustion-related genes (**e**). The top 20 clusters with their representative enriched terms across DEGs between iCAR-T$_{CTL}$ isolated from the tumor and spleen (**f**). Each bar represents a − log10-transformed adjusted *p*-value (Benjamini-Hochberg).

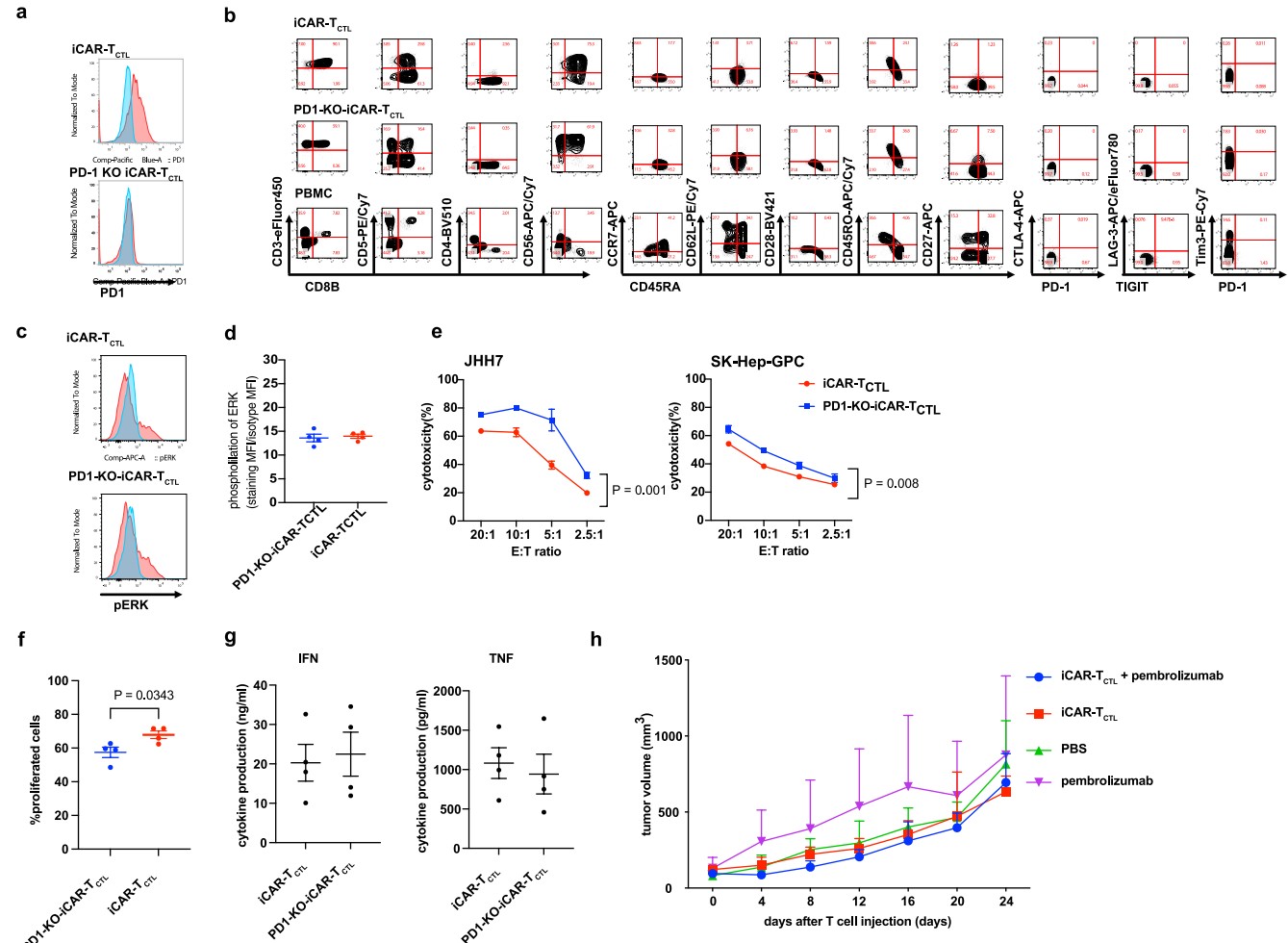

**Extended Data Fig. 6 | PD-1 deletion or blockade did not show distinctive enhancement of iCAR-T$_{CTL}$ therapy. a**. PD-1 expression of iCAR-T$_{CTL}$ and PD-1-KO-iCAR-T$_{CTL}$ after repeated stimulation. iCAR-T$_{CTL}$ and PD-1-KO-iCAR-T$_{CTL}$ were stimulated thrice with SK-Hep-GPC3 in a week and the expression of PD-1 was evaluated with flow cytometry. **b**. Surface antigen profiles of iCAR-T$_{CTL}$ and PD-1-KO-iCAR-T$_{CTL}$. iCAR-T$_{CTL}$ and PD-1-KO-iCAR-T$_{CTL}$ were evaluated using surface antigen expression related to T/NK linage, naïve/memory, and exhaustion phenotype. **c-d**. Detection of phosphorylated ERK (pERK) in iCAR-T$_{CTL}$ and PD-1-KO-iCAR-T$_{CTL}$ 60 min after co-culturing with irradiated SK-Hep-GPC3. **e**. *In vitro* cytotoxic assays of iCAR-T$_{CTL}$ and PD-1-KO-iCAR-T$_{CTL}$ against JHH7 and SK-Hep-

GPC3. $n = 4$, mean ± SD. Two-way ANOVA **f**. proliferation assay of iCAR-T$_{CTL}$ and PD-1-KO-iCAR-T$_{CTL}$. Two-sided Student's t test. **g**. Cytokine production of iCAR-T$_{CTL}$ and PD-1-KO-iCAR-T$_{CTL}$ 48 h after co-culturing with irradiated SK-Hep-GPC3 ($n = 4$). Two-sided Student's t test. **h**. NSG mice were injected subcutaneously with $2 × 10^5$ JHH7 cells 3 days before the treatment. In total, $1 × 10^7$ iCAR-T$_{CTL}$ were injected intravenously on days 0 and 7. For pembrolizumab group and combination therapy group of iCAR-T$_{CTL}$ and pembrolizumab, pembrolizumab was injected 200 μg/mice intraperitoneally every other day[50]. Tumor volume of the inoculated JHH7 at the indicated timepoint. Mean tumor size ± SEM ($n = 3$ of each group). SEM, standard error of the mean; SD, standard deviation.

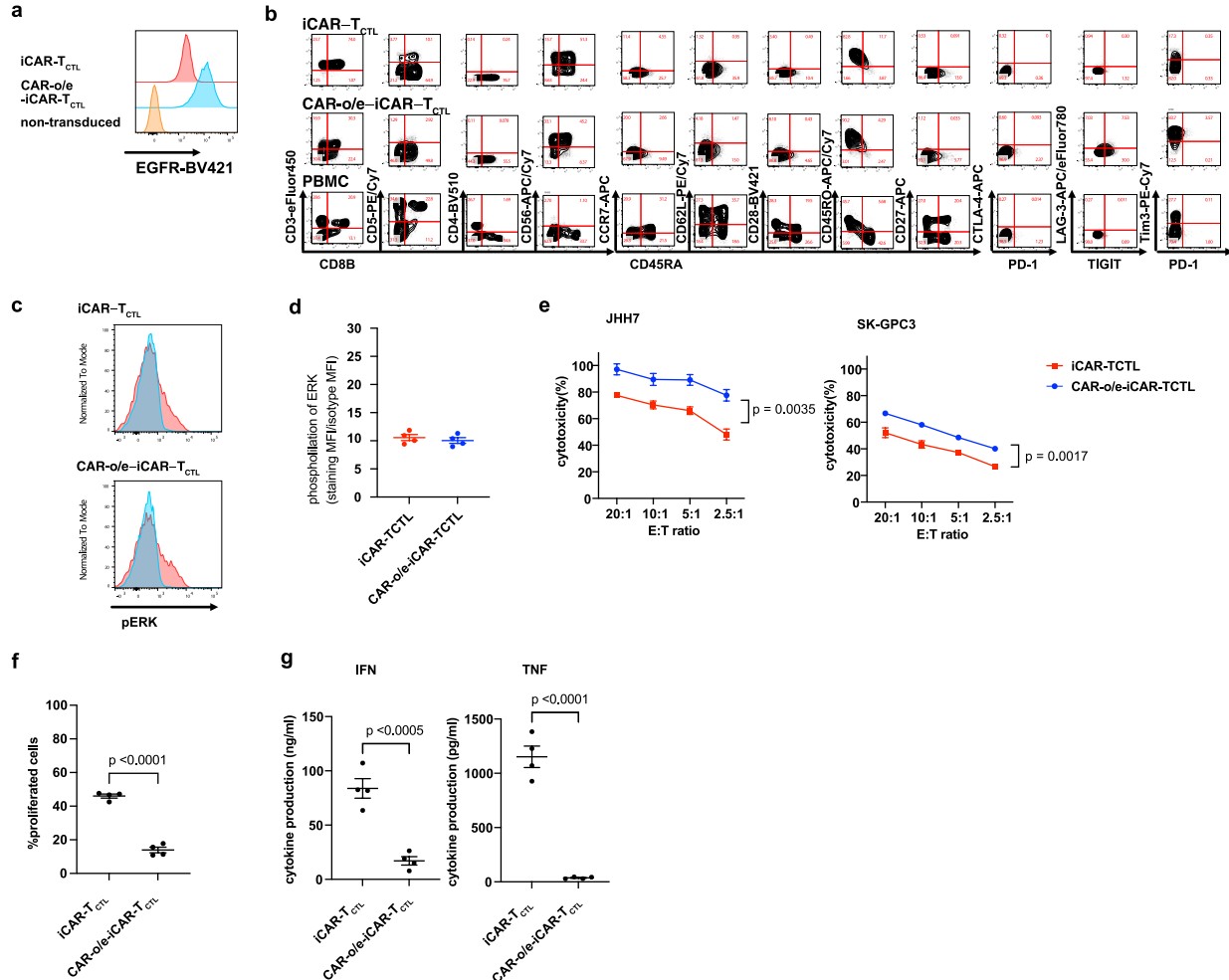

**Extended Data Fig. 7 | CAR overexpression did not show distinctive enhancement of iCAR-T$_{CTL}$ therapy. a**. Transgene marker expression of iCAR-T$_{CTL}$ and CAR-o/e-iCAR-T$_{CTL}$ **b**. Surface antigen profiles of iCAR-T$_{CTL}$ and CAR-o/e-iCAR-T$_{CTL}$. iCAR-T$_{CTL}$ and CAR-o/e-iCAR-T$_{CTL}$ were evaluated using surface antigen expression related to T/NK linage, naïve/memory, and exhaustion phenotype. **c-d**. Detection of phosphorylated ERK (pERK) in iCAR-T$_{CTL}$ and CAR-o/e-iCAR-

T$_{CTL}$ 60 min after co-culturing with irradiated SK-Hep-GPC3 **e**. *In vitro* cytotoxic assays of iCAR-T$_{CTL}$ and CAR-o/e-iCAR-T$_{CTL}$ against JHH7 and SK-Hep-GPC3. $n = 4$, mean ± SD. **f**. proliferation assay of iCAR-T$_{CTL}$ and CAR-o/e-iCAR-T$_{CTL}$. **g**. Cytokine production of iCAR-T$_{CTL}$ and CAR-o/e-iCAR-T$_{CTL}$ cells 48 h after co-culturing with irradiated SK-Hep-GPC3 ($n = 4$, mean ± SEM). Student's *t*-test.

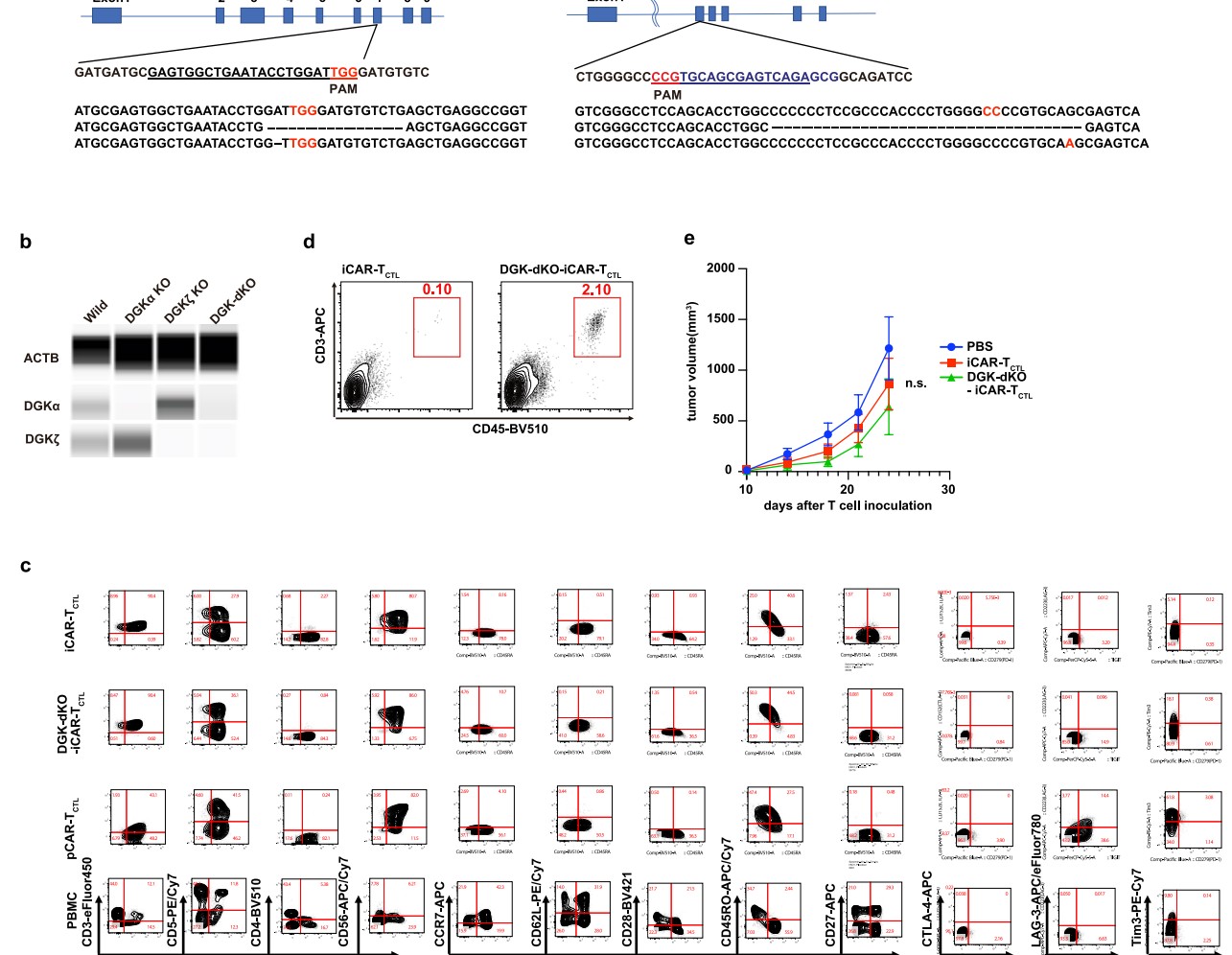

Extended Data Fig. 8 | Signal 1 enhancement improved the accumulation, persistency, and effector function of iCAR-T$_{CTL}$. a. The gRNA sequences of intact *DGKα* (exon 7) and *DGKζ* (exon 2) genes (upper and middle panels). Sequences of mutated *DGKα* and *DGKζ* genes in both alleles in DGK-dKO-CAR-iPSCs (lower panels). b. Western blotting for DGKα and DGKζ in iCAR-T$_{CTL}$ and DGK-dKO-iCAR-T$_{CTL}$ by a capillary electrophoresis-based Protein Simple Wes System. Data were processed using the Compass software (Protein Simple, San Jose, CA, USA). c. Ratio of CD5CD8β DP cells to live differentiated cells derived from wild-type iCAR, DGKα-KO iCAR, DGKζ-KO iCAR, DGK-dKO iCAR, and DGKα$^{-}$

$^{/-}$DGKζ$^{f/f}$ CAR-iPSCs. c. Surface antigen profiles of iCAR-T$_{CTL}$, DGK-dKO-iCAR-T$_{CTL}$, and pCAR-T$_{CTL}$. iCAR-T$_{CTL}$, DGK-dKO-iCAR-T$_{CTL}$, and pCAR-T$_{CTL}$ were evaluated using surface antigen expression related to T/NK linage, naïve/memory, and exhaustion phenotype. d. Detection of tumor-infiltrating T cells by FCM in JHH7-bearing xenograft mice treated with iCAR-T$_{CTL}$ or DGK-dKO-iCAR-T$_{CTL}$. e. Tumor volume of inoculated JHH7 at indicated time points in individual mice treated with indicated test cells. Mean tumor size ± SEM (n = 8 of each group). Two-way ANOVA. n.s., not significant; SEM, standard error of the mean.

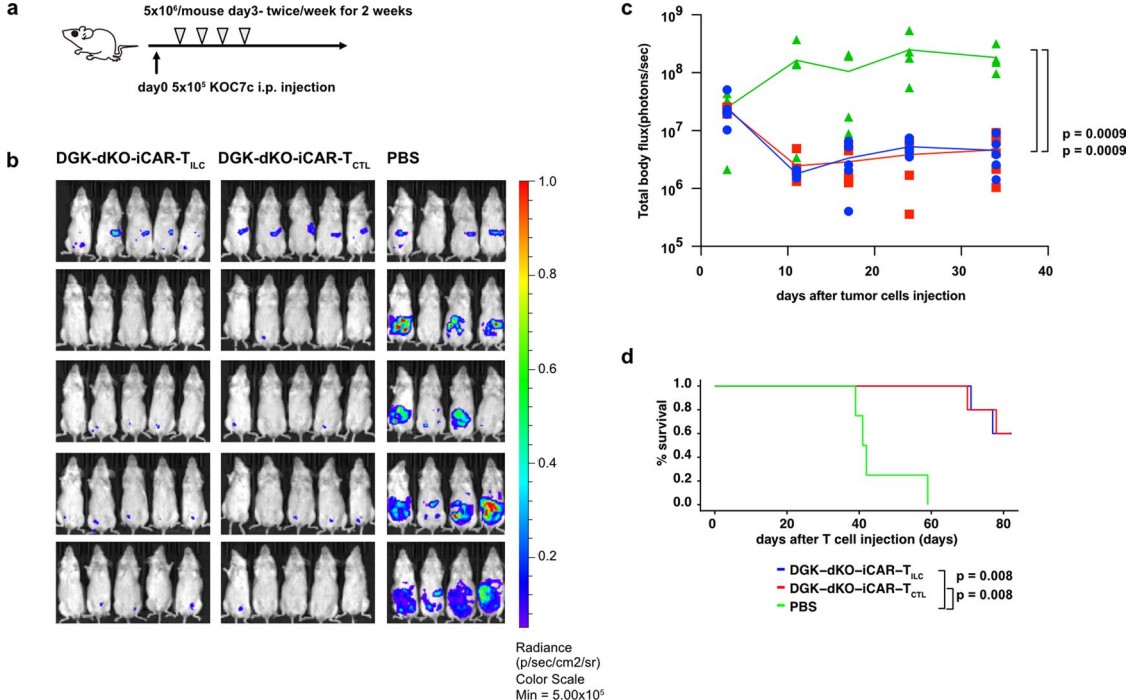

**Extended Data Fig. 9 | DGK-dKO-iCAR-T$_{ILC}$ have similar tumor suppressive function DGK-dKO-iCAR-T$_{CTL}$ in KOC7c pre-inoculated peritoneal dissemination mouse model. a-d**. NSG mice were injected intraperitoneally with $5 \times 10^5$ KOC7c cells expressing luciferase. From the third day after the ovarian cancer inoculation, $5 \times 10^6$ DGKdKO-iCAR-T$_{CTL}$ and DGKdKO-iCAR-TILC were injected intraperitoneally twice a week for 2 weeks ($n = 5$ for each group). *In vivo* bioluminescence imaging of luciferase-labeled KOC7c in NSG mice treated with DGKdKO-iCAR-T$_{CTL}$ and DGKdKO-iCAR-TILC (**b**). Change in the total body flux as total tumor volume (mean ± SEM) (**c**) and survival (**d**) were evaluated at the indicated time points after the injection. Log-rank test with Bonferroni multiple comparisons test. SEM, standard error of the mean.

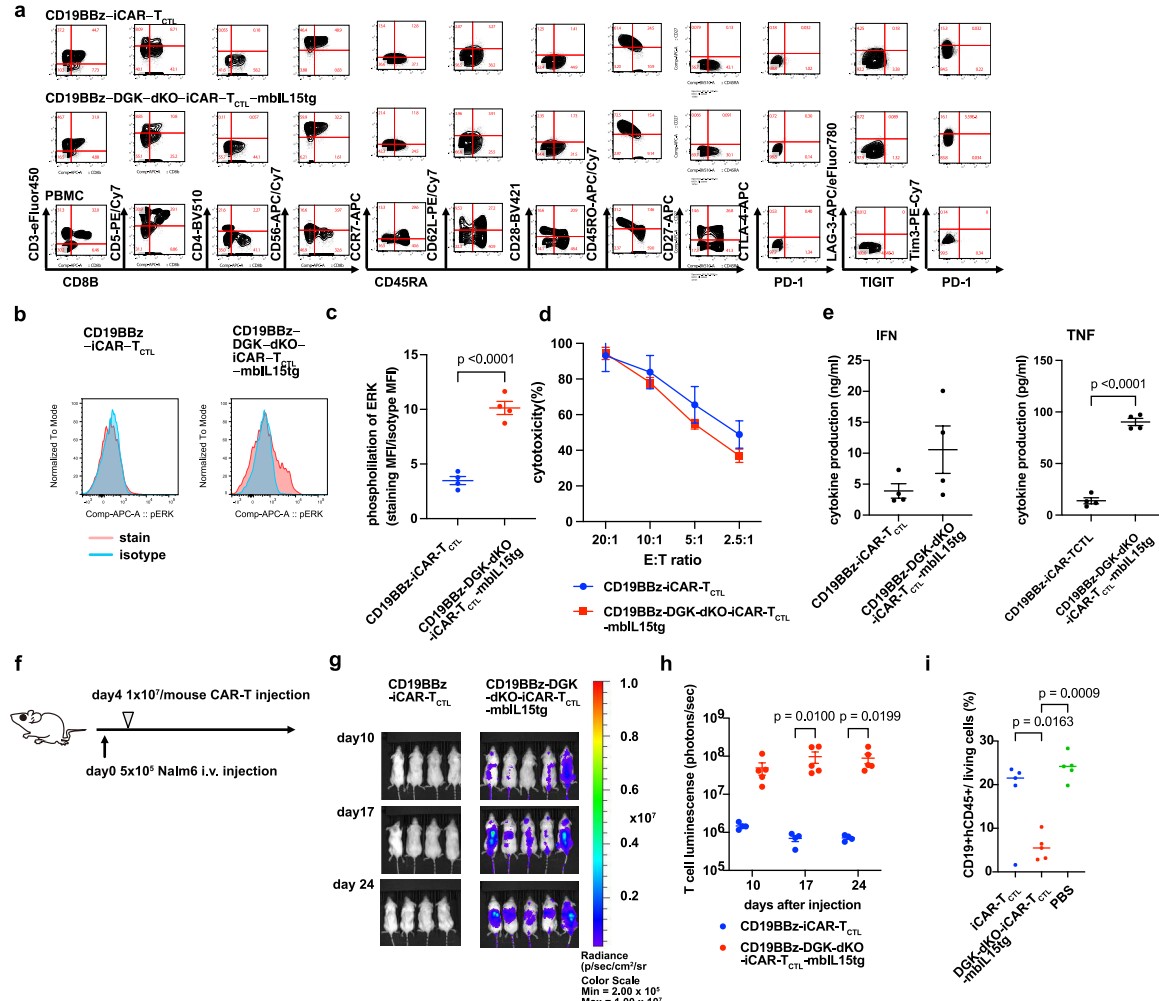

**Extended Data Fig. 10 | Combinatorial signal enhanced CD19CAR-iCAR-T$_{CTL}$ anti-tumor function. a** Surface antigen profiles of CD19BBz-iCAR-T$_{CTL}$ and CD19BBz-DGK-dKO-iCAR-T$_{CTL}$-mbIL15tg. CD19BBz-iCAR-T$_{CTL}$ and CD19BBz-DGK-dKO-iCAR-T$_{CTL}$-mbIL15tg were evaluated using surface antigen expression related to T/NK linage, naïve/memory, and exhaustion phenotype. **b-c**. Detection of phosphorylated ERK (pERK) in CD19BBz-iCAR-T$_{CTL}$ and CD19BBz-DGK-dKO-iCAR-T$_{CTL}$-mbIL15tg 60 minutes after co-culturing with irradiated Nalm6. Two-sided Student's t test. **d**. *In vitro* cytotoxic assays of CD19BBz-iCAR-T$_{CTL}$ and CD19BBz-DGK-dKO-iCAR-T$_{CTL}$-mbIL15tg against Nalm6. $n = 4$, mean ± SEM. **e**. Cytokine production of CD19BBz-iCAR-T$_{CTL}$ and CD19BBz-DGK-dKO-iCAR-T$_{CTL}$-mbIL15tg 48 h after co-culturing with irradiated Nalm6 ($n = 4$, mean ± SEM). Two-sided

Student's t test. **f–i**. Therapeutic efficacy of CD19CAR-iCAR-T$_{CTL}$ with in the Nalm6 systemic tumor model. Treatment schedule (**f**). NSG mice were injected intravenously with $5 \times 10^5$ Nalm6 cells 4 days before the treatment. In total, $1 \times 10^7$ CD19BBz-iCAR-T$_{CTL}$ and CD19BBz-DGK-dKO-iCAR-T$_{CTL}$-mbIL15tg were injected intravenously on days 4. *In vivo* bioluminescence imaging of the injected T cells in NSG mice treated with CD19BBz-iCAR-T$_{CTL}$, CD19BBz-DGK-dKO-iCAR-T$_{CTL}$-mbIL15tg (**g**). Total flux (photons/s) of the injected CAR-T cells was quantified at the indicated time points ($n = 4$ or 5, mean ± SEM). Two-way ANOVA Turkey's multiple comparison (**h**) mice blood cells were collected on day 23. Nalm6 (hCD45 + CD19 + ) cells were analyzed by flow cytometry. One-way ANOVA with Tukey's multiple comparisons test (**i**).

# Reporting Summary

## Statistics

For all statistical analyses, confirm that the following items are present in the figure legend, table legend, main text, or Methods section.

| n/a | Confirmed | |
|---|---|---|
| ☐ | ☒ | The exact sample size (*n*) for each experimental group/condition, given as a discrete number and unit of measurement |
| ☐ | ☒ | A statement on whether measurements were taken from distinct samples or whether the same sample was measured repeatedly |
| ☐ | ☒ | The statistical test(s) used AND whether they are one- or two-sided *Only common tests should be described solely by name; describe more complex techniques in the Methods section.* |
| ☐ | ☒ | A description of all covariates tested |
| ☐ | ☒ | A description of any assumptions or corrections, such as tests of normality and adjustment for multiple comparisons |
| ☐ | ☒ | A full description of the statistical parameters including central tendency (e.g. means) or other basic estimates (e.g. regression coefficient) AND variation (e.g. standard deviation) or associated estimates of uncertainty (e.g. confidence intervals) |
| ☐ | ☒ | For null hypothesis testing, the test statistic (e.g. *F*, *t*, *r*) with confidence intervals, effect sizes, degrees of freedom and *P* value noted *Give P values as exact values whenever suitable.* |
| ☒ | ☐ | For Bayesian analysis, information on the choice of priors and Markov chain Monte Carlo settings |
| ☒ | ☐ | For hierarchical and complex designs, identification of the appropriate level for tests and full reporting of outcomes |
| ☒ | ☐ | Estimates of effect sizes (e.g. Cohen's *d*, Pearson's *r*), indicating how they were calculated |

*Our web collection on statistics for biologists contains articles on many of the points above.*

## Software and code

Policy information about availability of computer code

| Data collection | FACS Diva 8.0.1 (Flow cytometry), FCAP array v. 3.0.19 (CBA) |
|---|---|
| Data analysis | Software used to analyse the data include FlowJo version 10, Prism, R version 4.2.0 (Codes can be made available on request.), and Living Image version 4.5.5. |

For manuscripts utilizing custom algorithms or software that are central to the research but not yet described in published literature, software must be made available to editors and reviewers. We strongly encourage code deposition in a community repository (e.g. GitHub). See the Nature Portfolio guidelines for submitting code & software for further information.

## Data

Policy information about availability of data

All manuscripts must include a data availability statement. This statement should provide the following information, where applicable:
- Accession codes, unique identifiers, or web links for publicly available datasets
- A description of any restrictions on data availability
- For clinical datasets or third party data, please ensure that the statement adheres to our policy

The primary data supporting the results in this study are available within the paper and its Supplementary Information. Source data for tumour burden is provided with this paper. The raw and analysed datasets generated during the study are too large to be publicly shared, yet they are available for research purposes from the corresponding author on reasonable request.

# Human research participants

Policy information about <u>studies involving human research participants and Sex and Gender in Research.</u>

| | |
|---|---|
| Reporting on sex and gender | PBMCs and T cells from healthy donors were used. Sex and gender were not available. |
| Population characteristics | — |
| Recruitment | Healthy donors were recruited in accordance with the protocol approved by the Kyoto University School of Medicine Ethical Committee. |
| Ethics oversight | The use of the purchased PBMCs was approved by the Kyoto University School of Medicine Ethical Committee |

Note that full information on the approval of the study protocol must also be provided in the manuscript.

# Field-specific reporting

Please select the one below that is the best fit for your research. If you are not sure, read the appropriate sections before making your selection.

☒ Life sciences ☐ Behavioural & social sciences ☐ Ecological, evolutionary & environmental sciences

For a reference copy of the document with all sections, see nature.com/documents/nr-reporting-summary-flat.pdf

# Life sciences study design

All studies must disclose on these points even when the disclosure is negative.

| | |
|---|---|
| Sample size | No statistical methods were used to predetermine the experimental sample size. Sample sizes were determined on the basis of relevant literature (such as Themeli et.al, Nat. Biotech. 2013). |
| Data exclusions | No data were excluded. |
| Replication | For all figures, two or three independent experiments were performed and all attempts at replicating the observations were successful. Similar results were obtained across two laboratories. |
| Randomization | All samples were number-coded until the measurement was completed. For the in vivo experiments, mice were randomly assigned to each group. |
| Blinding | Blinding was not performed. Fully blinded experiments were not possible owing to personnel availability during the experiments. |

# Reporting for specific materials, systems and methods

We require information from authors about some types of materials, experimental systems and methods used in many studies. Here, indicate whether each material, system or method listed is relevant to your study. If you are not sure if a list item applies to your research, read the appropriate section before selecting a response.

## Materials & experimental systems

| n/a | Involved in the study |
|---|---|
| ☐ | ☒ Antibodies |
| ☐ | ☒ Eukaryotic cell lines |
| ☒ | ☐ Palaeontology and archaeology |
| ☐ | ☒ Animals and other organisms |
| ☒ | ☐ Clinical data |
| ☒ | ☐ Dual use research of concern |

## Methods

| n/a | Involved in the study |
|---|---|
| ☒ | ☐ ChIP-seq |
| ☐ | ☒ Flow cytometry |
| ☒ | ☐ MRI-based neuroimaging |

# Antibodies

| | |
|---|---|
| Antibodies used | For T cell phenotyping, the following antibodies were used:<br>CD3-eFluor 450(clone UCHT1,eBioscience), CD3-APC-Cy7(clone UCHT1,BioLegend), CD3-APC(clone UCHT1,BioLegend), CD4-Brilliant Violet 421(clone OKT4,BioLegend), CD4-Brilliant Violet 510(clone OKT4,BioLegend), CD5-FITC(clone UCHT2,eBioscience), CD5-PE-Cy7(clone UCHT2,eBioscience), CD7-APC(clone CD7-6B7,BioLegend), CD8a-PerCP-Cy5.5(clone SK1,BioLegend), CD8b-PE-Cy7(clone |

SIDI8BEE,eBioscience), CD8b-APC(clone 2ST8.5H7,BD), CD19-PE(clone HIB19,BD), CD25-FITC(clone BC96,BioLegend), CD27-APC(clone O323,BioLegend), CD28-Brilliant Violet 421(clone CD28.2,BioLegend), CD45-Brilliant Violet 510(clone HI30,BioLegend), CD45RA-Brilliant Violet 510(clone HI100,BioLegend), CD45RO-APC-Cy7(clone UCHL1,BioLegend), CD56-APC-Cy7(clone HCD56,BioLegend), CD62L-PE-Cy7(clone DREG-56,BioLegend), CD69-PacificBlue(clone FN50,BioLegend), CD94(NKG2A)-Brilliant Violet 421(clone HP-3D9,BD Biosciences), CD159a(KLRC)-PacificBlue(clone S19004C,BioLegend ), CD161(KLRB)-PE-Cy7(clone HP-3G10,BioLegend), CD197(CCR7)-APC(clone G043H7,BioLegend), CD223(LAG-3)-APC-eFluor780(clone 3DS223H,eBioscience), CD226(DNAM)-APC(clone 11A8,BioLegend), CD247(pY142)-Alexa Fluor 647(clone K25-407.69,BD), CD152(CTLA-4)-APC(clone L3D10,BioLegend), CD271(NGFR)-APC-Fire750(clone ME20.4,BioLegend), CD279(PD-1)-Brilliant Violet 421(clone 29F.1A12,BioLegend), CD314(NKG2D)-PE-Cy7(clone 1D11,BioLegend), CD335(NKp46)-FITC(clone 900,BioLegend), CD336(NKp44)-APC(clone P44-8,BioLegend), CD337(NKp30)-APC(clone P30-15,BioLegend), EGFR-Brilliant Violet 421(clone AY13,BioLegend), ERK1/2 (pT202/pY204) -Alexa Fluor 647(clone 20A,BD Biosciences), Mouse IgG2a k-Alexa Fluor 647(clone MOPC-173,BD Biosciences), TCRab-FITC(clone WT31,eBioscience), TIGIT-PerCP-eFluor 710(clone MBSA43,eBioscience), Tim3-PE-Cy7(clone F38-2E2,BioLegend)

| | |
|---|---|
| Validation | Antibodies were validated using positive and negative cells using human PBMCs or isotype controls.<br>Validation reports were also provided by the antibody manufacturers(BioLegend, BD biosciences and eBiosciences). Compensation controls were used for every experiment.<br>https://www.thermofisher.com/antibody/product/CD3-Antibody-clone-UCHT1-Monoclonal/48-0038-42<br>https://www.biolegend.com/en-us/search-results/apc-cyanine7-anti-human-cd3-antibody-3929<br>https://www.biolegend.com/en-us/products/apc-cyanine7-anti-human-cd3-antibody-861?GroupID=BLG5900<br>https://www.biolegend.com/en-us/search-results/brilliant-violet-421-anti-human-cd4-antibody-7775<br>https://www.biolegend.com/ja-jp/products/brilliant-violet-510-anti-human-cd4-antibody-8010<br>https://www.thermofisher.com/antibody/product/CD5-Antibody-clone-UCHT2-Monoclonal/11-0059-42<br>https://www.thermofisher.com/antibody/product/CD5-Antibody-clone-UCHT2-Monoclonal/25-0059-42<br>https://www.biolegend.com/en-us/products/apc-anti-human-cd7-antibody-6088<br>https://www.biolegend.com/en-us/products/percp-cyanine5-5-anti-human-cd8-antibody-6389?GroupID=BLG10167<br>https://www.thermofisher.com/antibody/product/CD8b-Antibody-clone-SIDI8BEE-Monoclonal/25-5273-42<br>https://www.bdbiosciences.com/en-us/products/reagents/flow-cytometry-reagents/clinical-discovery-research/single-color-antibodies-ruo-gmp/apc-mouse-anti-human-cd8.641058<br>https://www.bdbiosciences.com/en-ca/products/reagents/flow-cytometry-reagents/research-reagents/single-color-antibodies-ruo/pe-mouse-anti-human-cd19.555413<br>https://www.biolegend.com/ja-jp/products/fitc-anti-human-cd25-antibody-615<br>https://www.biolegend.com/en-us/products/apc-anti-human-cd27-antibody-808?GroupID=BLG7922<br>https://www.biolegend.com/en-us/products/brilliant-violet-421-anti-human-cd28-antibody-8170?GroupID=BLG10175<br>https://www.biolegend.com/ja-jp/products/brilliant-violet-510-anti-human-cd45-antibody-8006<br>https://www.biolegend.com/en-us/products/brilliant-violet-510-anti-human-cd45ra-antibody-8007?GroupID=GROUP658<br>https://www.biolegend.com/en-us/products/apc-cyanine7-anti-human-cd45ro-antibody-7372?GroupID=GROUP658<br>https://www.biolegend.com/en-us/products/apc-cyanine7-anti-human-cd56-ncam-antibody-7115?GroupID=BLG15664<br>https://www.biolegend.com/en-us/products/pe-cyanine7-anti-human-cd62l-antibody-3944<br>https://www.biolegend.com/en-us/products/pacific-blue-anti-human-cd69-antibody-3360?GroupID=BLG10251<br>https://www.bdbiosciences.com/en-us/products/reagents/flow-cytometry-reagents/research-reagents/single-color-antibodies-ruo/bv421-mouse-anti-human-cd94.743948<br>https://www.biolegend.com/en-us/search-results/pacific-blue-anti-human-cd159a-nkg2a-antibody-20202<br>https://www.biolegend.com/en-us/products/apc-cyanine7-anti-human-cd161-antibody-9972?GroupID=BLG9768<br>https://www.biolegend.com/en-us/products/apc-anti-human-cd197-ccr7-antibody-7536?GroupID=BLG9613<br>https://www.thermofisher.com/antibody/product/47-2239-42.html?ef_id=Cj0KCQjw2_OWBhDqARIsAAUNTTGJ0a5OykcKSWEwYFzjrXTkkjoIvc5IPGRGNJh5smE1GEqMQrzBGWcaAufkEALw_wcB:G:s&s_kwcid=AL!3652!3!278870232429!!!g!!&cid=bid_pca_frg_r01_co_cp1359_pjt0000_bid00000_0se_gaw_dy_pur_con&gclid=Cj0KCQjw2_OWBhDqARIsAAUNTTGJ0a5OykcKSWEwYFzjrXTkkjoIvc5IPGRGNJh5smE1GEqMQrzBGWcaAufkEALw_wcB<br>https://www.biolegend.com/ja-jp/products/apc-anti-human-cd226-dnam-1-antibody-8465<br>https://www.bdbiosciences.com/en-nz/products/reagents/flow-cytometry-reagents/research-reagents/single-color-antibodies-ruo/alexa-fluor-647-mouse-anti-cd247-py142.558489<br>https://www.biolegend.com/ja-jp/products/apc-anti-human-cd152-ctla-4-antibody-6999<br>https://www.biolegend.com/ja-jp/products/apc-fire-750-anti-human-cd271-ngfr-antibody-16306<br>https://www.biolegend.com/ja-jp/products/brilliant-violet-421-anti-mouse-cd279-pd-1-antibody-7330<br>https://www.biolegend.com/en-us/products/pe-cyanine7-anti-human-cd314-nkg2d-antibody-6499?GroupID=BLG8540<br>https://www.biolegend.com/en-us/products/fitc-anti-human-cd335-nkp46-antibody-8464?GroupID=BLG8494<br>https://www.biolegend.com/en-us/search-results/apc-anti-human-cd336-nkp44-antibody-3850<br>https://www.biolegend.com/en-us/products/apc-anti-human-cd337-nkp30-antibody-3856?GroupID=BLG5091<br>https://www.biolegend.com/en-us/products/brilliant-violet-421-anti-human-egfr-antibody-8621<br>https://www.bdbiosciences.com/en-de/products/reagents/flow-cytometry-reagents/research-reagents/single-color-antibodies-ruo/alexa-fluor-647-mouse-igg2a-isotype-control.558053<br>https://www.bdbiosciences.com/en-us/products/reagents/flow-cytometry-reagents/research-reagents/single-color-antibodies-ruo/alexa-fluor-647-mouse-anti-erk1-2-pt202-py204.561992<br>https://www.thermofisher.com/antibody/product/TCR-alpha-beta-Antibody-clone-WT31-Monoclonal/11-9955-42<br>https://www.thermofisher.com/antibody/product/TIGIT-Antibody-clone-MBSA43-Monoclonal/46-9500-42<br>https://www.biolegend.com/ja-jp/products/pe-cyanine7-anti-human-cd366-tim-3-antibody-8303 |

# Eukaryotic cell lines

Policy information about cell lines and Sex and Gender in Research

| | |
|---|---|
| Cell line source(s) | TKT3V1-7 is an iPS cell line established by Nishimura et. al (Cell Stem Cell, 2013).<br>HepG2(JCRB1054) ,JHH7(JCRB1031) and Nalm6(CRL-3273) were purchased from JCRB.<br>SK-Hep1-GPC3, SK-Hep1-Vector, K562, RD-18 and KOC7c were provided by co-authors. |

| | |
|---|---|
| Authentication | The expression of glypican3 was checked by flow-cytometric analysis. |
| Mycoplasma contamination | All cell lines were confirmed negative for mycoplasma contamination. |
| Commonly misidentified lines (See ICLAC register) | None of the cell lines used in the study are listed in the ICLAC Database of Cross-contaminated or Misidentified Cell Lines. |

## Animals and other research organisms

Policy information about studies involving animals; ARRIVE guidelines recommended for reporting animal research, and Sex and Gender in Research

| | |
|---|---|
| Laboratory animals | 6–12 week-old female NOD-SCID IL2Rγcnull (NSG) mice were purchased from Oriental Bio (Yokohama, Japan). Mice were exposed to 12h:12 h light–dark cycles with free access to water and food. The ambient temperature was restricted to 20–26 degrees Celsius and room humidity was 40–70%. |
| Wild animals | The study did not involve wild animals. |
| Reporting on sex | Female mice were used. |
| Field-collected samples | The study did not involve samples collected from the field. |
| Ethics oversight | All animal experiments were performed in accordance with the Kyoto University School of Medicine Ethical Committee. |

Note that full information on the approval of the study protocol must also be provided in the manuscript.

## Flow Cytometry

### Plots

Confirm that:

☒ The axis labels state the marker and fluorochrome used (e.g. CD4-FITC).

☒ The axis scales are clearly visible. Include numbers along axes only for bottom left plot of group (a 'group' is an analysis of identical markers).

☒ All plots are contour plots with outliers or pseudocolor plots.

☒ A numerical value for number of cells or percentage (with statistics) is provided.

### Methodology

| | |
|---|---|
| Sample preparation | All stains were performed with < 1x10^6 cells per 100 μL staining buffer (PBS + 2% FBS) with 1:100 dilution of each antibody, 20 min on ice in dark. |
| Instrument | Stained cell samples were analyzed using LSR or FACS AriaII flow cytometer (BD Biosciences). |
| Software | The data were processed using FlowJo (Tree Star). |
| Cell population abundance | Sorted samples were confirmed for purity post-sort via flow cytometry. Sorted populations were confirmed to be of >95% purity. |
| Gating strategy | All human cells were first gated on FSC/SSC according to cell size and granularity, using stained human peripheral mononuclear cells (PBMCs) as a positive control and reference for cell size, granularity and staining intensity. Unstained samples were used to set up negative gates, and stained human PBMCs were used to set up positive gates. Dead-cell populations were excluded using PI staining. |

☐ Tick this box to confirm that a figure exemplifying the gating strategy is provided in the Supplementary Information.

