## [Peer Review File · Nature Biomedical Engineering]

Optimization of the proliferation and persistency of CAR T cells derived from human induced pluripotent stem cells

Corresponding author: Shin Kaneko

Editorial note

This document includes relevant written communications between the manuscript's corresponding author and the editor and reviewers of the manuscript during peer review. It includes decision letters relaying any editorial points and peer-review reports, and the authors' replies to these (under 'Rebuttal' headings). The editorial decisions are signed by the manuscript's handling editor, yet the editorial team and ultimately the journal's Chief Editor share responsibility for all decisions.

Any relevant documents attached to the decision letters are referred to as **Appendix #**, and can be found appended to this document. Any information deemed confidential has been redacted or removed. Earlier versions of the manuscript are not published, yet the originally submitted version may be available as a preprint. Because of editorial edits and changes during peer review, the published title of the paper and the title mentioned in below correspondence may differ.

Correspondence

Tue 25 Jan 2022

Decision on Article nBME-21-2928

Dear Prof Kaneko,

Thank you again for submitting to *Nature Biomedical Engineering* your manuscript, "Engineering differentiation and signaling pathways of CAR-T cells from human iPSCs enhances anti-solid tumor activity". The manuscript has been seen by three experts, whose reports you will find at the end of this message.

You will see that the reviewers appreciate the work. However, they express concerns about the degree of support for the claims, and provide useful suggestions for improvement. We hope that with significant further work you can address the criticisms and convince the reviewers of the merits of the study. In particular, we would expect that a revised version of the manuscript provides:

- * Enhanced (and statistically powered) comparisons across cell products, as suggested (directly or indirectly) by all reviewers.
- * Thorough characterization of the cell populations, as per the relevant comments of all reviewers.
- * Exhaustive reporting of the methodology and statistical analyses, so as to facilitate the replicability of the work and the adoption of the strategy for the generation of iPSC-derived CAR T cells.

Also, the overall write-up, and in particular the Discussion section, should be improved for clarity and flow. In addition, I suggest that the layout of the figures be improved for improved visualisation and interpretation at normal magnification. Articles can have up to 8 main figures; hence, in particular, Fig. 4 could be split into two and its layout improved.

When you are ready to resubmit your manuscript, please upload the revised files, a point-by-point rebuttal tothe comments from all reviewers, the reporting summary, and a cover letter that explains the main improvements included in the revision and responds to any points highlighted in this decision.

Please follow the following recommendations:

- * Clearly highlight any amendments to the text and figures to help the reviewers and editors find and understand the changes (yet keep in mind that excessive marking can hinder readability).
- * If you and your co-authors disagree with a criticism, provide the arguments to the reviewer (optionally, indicate the relevant points in the cover letter).
- * If a criticism or suggestion is not addressed, please indicate so in the rebuttal to the reviewer comments and explain the reason(s).
- * Consider including responses to any criticisms raised by more than one reviewer at the beginning of the rebuttal, in a section addressed to all reviewers.
- * The rebuttal should include the reviewer comments in point-by-point format (please note that we provide all reviewers will the reports as they appear at the end of this message).
- * Provide the rebuttal to the reviewer comments and the cover letter as separate files.

We hope that you will be able to resubmit the manuscript within 15 weeks from the receipt of this message. If this is the case, you will be protected against potential scooping. Otherwise, we will be happy to consider a revised manuscript as long as the significance of the work is not compromised by work published elsewhere or accepted for publication at *Nature Biomedical Engineering*.

We hope that you will find the referee reports helpful when revising the work. Please do not hesitate to contact me should you have any questions.

Best wishes,

Pep

Pep Pàmies
Chief Editor, Nature Biomedical Engineering

Reviewer #1 (Report for the authors (Required)):

Ueda, Kaneko, and colleagues report their recent results where they optimized manufacturing of CAR-expressing T-cells from iPSC progenitors. They focus on the generation of GPC3 CAR-armed iT cells that undergo full T-cell differentiation via the DP stage producing functional CD8ab-expressing single-positive T-cells following positive selection with anti-CD3 and dexamethasone. The activity of GPC3 CAR iT cells in vivo is further augmented by disrupting the expression of DGK, a key enzyme limiting the levels of DAG downstream of CD3zeta signaling, and by ectopic expression of the IL-15/IL-15Ra complex on the surface of iT-cells. The resulting GPC3 CAR T-cells armed with the said modification produce robust anti-tumor activity in mouse xenograft models of human hepatoma and ovarian carcinoma. Overall, the study is performed at a high technical level with appropriate controls and models, addresses a key limitation of current cell therapies of cancer, and would be of interest to the audience interested in engineered T-cells. Addressing the following points and questions would further augment the impact of this study.

1. Disruption of DGK isoforms has been previously reported to augment anti-tumor function of T-cells, including CAR T-cells (Riese et al., 2013 and Jung et al., 2018 – both of these studies should be mentioned in the paper). Similarly, the benefit of IL-15/IL-15Ra expression on CAR T-cells is well established in the field

(Hurton et al., 2016 and other studies). Why did the authors select these particular modifications over similar strategies that boost Signal 1 (modifying CAR expression level, amplifying zeta chains, disrupting checkpoint inhibitors) and Signal 3 (expression of IL-7 or IL-21 etc)? Does the DGK disruption and the IL-15 boost especially benefit iPSC-derived iT-cells, more so than conventional CAR T-cells that have similar modifications?

2. DGK limits the duration and magnitude of TCR signaling in T-cells. Does DGK disruption in iPSC affect the subsequent positive and negative selection of iT cells by modulating their TCR signaling intensity? Are DGK-deficient T-cells more prone to negative selection and does the DGK removal affect the naïve/memory/activation/exhaustion phenotype of the resulting iT cells? Finally – does DGK disruption augment cytotoxicity and survival of CAR iT cells or only modulates their proliferation?

3. Does IL-15 overexpression affect the phenotype and subset composition of GPC3 iT cells or their expansion upon tumor challenge in vitro? Do the authors envision modifying mature iT cells with the IL-15/IL-15Ra complex or this modification can be made in parental iPSC?

4. The combination of IL-15 expression and DGK disruption significantly enhances the anti-tumor function of GPC3 CAR iT-cells in models of HCC and ovarian carcinoma. How generalizable are these findings? Would the same modifications enhance the function of iT cells expressing other CARs?

Minor points:

1. In the abstract, the authors state the modified iT cells will “significantly prolong survival with an accelerated cancer immunity cycle”. I suggest the authors clarify exactly what they mean as the cancer-immunity cycle is quite multifaceted and encompasses several other aspects not covered in this study.

2. In Figure 2d, CD62L is downregulated whereas CCR7 is upregulated in mature iCAR-T. Usually T-cells exhibit an opposite pattern where CD62L+ T-cells lose CCR7 expression. Is the observed phenotype stable in mature T-cells or it reflects transient shedding of CD62L and upregulation of CCR7 driven by strong TCR signaling?

3. In Figure 2g, the authors show modestly increased accumulation of iCAR-T CTL over iCAR-T ILC in tumors. To rule out that these effects are due to the variability in the number of cells injected to tumor-bearing mice, the authors should show the signal on day 0 (right after cell injection) is comparable in all these groups.

4. The discussion section is a bit unfocused and should reflect authors' thoughts on the most appropriate strategies to address current limitations of iPSC by either boosting stimulation (Signals 1-2-3) or other approaches that could bridge the gap between conventional and iPSC-derived T-cells.

5. While the write-up is clear and relatively easy to follow, the manuscript would benefit from additional editing for clarity and language.

Reviewer #2 (Report for the authors (Required)):

In this study the authors engineered iPSCs in a stepwise fashion to obtain highly functional iCAR-T cells with improved proliferative capacity and persistence at the tumor site. They modified the CAR constructs (eg including CD3z/CD28/4-1BB) to avoid tonic signaling and thereby optimized the differentiation process to achieve CD8ab+ T cells via a CD4CD8 double positive stage. The authors compare iCAR-T(ILC) with iCAR-T(CTL) and find that the adaptive type cells have superior capability to accumulate at the tumor site than innate type regenerated cells. They move on to further engineer the iCAR-T cells by deleting DAG kinases to enhance the TCR signalling. Together with further transduction of mbIL15, this led to significantly improved proliferative capacity resulting in better tumor control in vivo.

Overall, this is an ambitious study that shed further light on the role of tonic signalling in differentiation fates and the functional differences of such cells. The work also paves way for tuning TCR signalling to achieve more responsive iPSC-T cells with better persistence in the tumor. The combination approach shown in Figure 4 has merit and lifts the manuscript substantially towards the end.

Major concerns

The head-to-head comparison of innate and adaptive iCAR-T cells in Figure 2 is a bit problematic, as is often the case when comparing cells generated through different protocols. In most in vitro assays, the “innate” cells appear to outperform the “adaptive” cells (2e) and the differences in vivo are modest (non-significant) and almost at background levels compared to pCAR-T cells (Fig 2f,g). It is unclear if the different differentiation protocols per se (different stimulatory input) or the end phenotype/cell state is responsible for any differences seen in Figure 2. I realize there is no way to get around this problem since different protocols are required to generate the different cell fates. However, given the modest functional phenotype, statements such as: “..adaptive type regenerated T cells.... have superior capacity to accumulate at the tumor site in vivo than innate type....” are not supported by the data and needs to be toned down or substantiated by additional experiments.

The authors state that the comparison of the transcriptomes of these three cell populations suggest that iCAR-T(CTL) have superior memory T cell function. However, this is based on n=1 and is not definitive. Overall, this section (Fig 2 and Supp Fig 2) needs to be much substantiated or perhaps rather abandoned? The key observation seems to be that iCAR-T(CTL) underperform relative to pCAR-T and that further modifications are needed. If the authors, would like to keep the comparison between “innate” and “adaptive” iCAR T cells and their ability to accumulate at the tumor site additional experimental support for this statement is needed. For example, the data shown in Supp Fig 3a is based on how many experiments, sections? To me it makes more sense to move the comparison of innate and adaptive phenotypes to the supplement and simply state that the failure of unmodified iCAR-T (CTL) versus pCAR-T is also seen using conventional differentiation strategies yielding iCAR-T (ILC).

As a general concern, all figure legends need to include information of number of experiments. One representative experiment out of x? Key findings need to be based on multiple experiments and include proper statistics. For transcriptional comparisons, this means inclusion of multiple independent differentiation runs from the same iPSC line or ideally at assessment of two independently generated lines. As I am sure the authors are well aware of, iPSC lines have “personalities” so to compare wt (n=1) with one gene edited version (n=1) will likely generate a lot of DEG that are irrelevant and would not show up in a second wt/KO pair. It would be best if the authors identify which comparison that is best suited for deep transcriptional analysis in order to gain mechanistic insights into the behaviour of the mature iCAR-T cells. Perhaps the DAGkinase KO line to gain insight into its superior persistence?

The discussion is a bit review-like and could perhaps discuss the explored concepts in more detail in relation to previous literature. In particular it would be interesting to discuss how the addition of 41BB provide co-stimulation and yet counter the tonic signalling by z/28.

The statement (row 60) that recent reports indicated improved differentiation protocols to make CD8ab-expressing T cells that showed effector functions more closely resembling primary T cells. Which reports are the authors referring to here? Please add citations.

Reviewer #3 (Report for the authors (Required)):

This is an interesting paper addressing a number of relevant issues in T-IPSC derived CAR-T (iCAR-T cells) cells. In the first part of the study, the authors addressed the optimal CAR-design for the generation of CD4/CD8 double positive and CD8ab iCAR-T cells using the well-known CD19-targeting FMC63 CAR.

In the second part, in which the glypican3 (GPC3) CARs were used, the authors aim to improve the overall functionality, in vivo persistence and anti-tumor efficacy of CD8ab+ iCAR-T cells (designated as iCARCTLs in the paper) by i) knock out of an earlier described immune checkpoint genes DGK α and DGK δ and ii) by ectopic expression (retroviral transduction) of the membrane-bound IL-15/IL-15Ra (mbIL5) in these cells.

I appreciate the data in figure 4, which demonstrate that the final product (designated as DGK-dKO- iCAR-TCTL-mbIL15tg cells) indeed possesses an improved in vivo survival and superior anti-tumor efficacy, which is even better than the anti- tumor efficacy of a second generation conventional CART cells (pCAR-TCTL).

The results confirm that also iPSC-derived CART cells can significantly benefit from genetic modulations which are aiming at preventing exhaustion and arming the cells with (cytokine) signals for a better in vivo survival and persistence.

Nonetheless, in the light of the data presented especially in figures 1 and 2, I have the impression that the final product was not entirely developed by careful analysis and comparison of the other possibly powerful products. To my opinion this also affects the scientific merit of the study, because:

1. The data presented in figure 1 do not show a clear advantage of using a third generation CAR (CD28+BB1 co-stimulation) above a second generation CAR containing only BB1 costimulation. This is because the use of either 2nd.Gen BB1 CAR or the 3rd.gen. CAR results in the development of similar levels of DP T cells. Thus, a systematic analysis would also include the 2nd.gen.BBz CARs in the second part of the study.
2. The data presented in figure 2 (and related supplementary figures) do not show a clear advantage of using iCAR-CTL above iCAR-ILC.
Although iCAR-CTLs seem to better accumulate in the tumor tissue than iCAR-ILC, this advantage disappears within 14days (no significant difference between iCAR-CTL and iCAR-ILC at day 14 in figure 2g); In contrast, the iCAR-ILCs - thanks to their additional NK-dependent kill capacity- are significantly better killers of the antigen positive tumors as compared to iCAR-CTLs (figure 2e). Finally, there is no difference between iCAR-CTL and iCAR-ILC with respect to their in vivo anti-tumor efficacy in an intraperitoneal tumor model (supplementary figure 2f,g,l,h). Hence, a systematic analysis towards the optimal product should also include the testing of iCAR-ILCs in the further stages of the study.
3. The choice of knock down of DGK α and DGK δ genes is not based on any data specific for iPSC derived CART cells as there are a plethora of potential immune checkpoints that could be modulated. It is from the manuscript not even clear which immune checkpoints are (over) expressed on/in these cells.
4. Similarly, the decision to insert the mbIL5 gene in the cells is primarily based on earlier published successful studies but not data derived from the study.
5. Finally, the provided genetic expression profiling studies are too global to provide specific clues on the quality of the cells. To my opinion the genetic data could be further analyzed to obtain specific information about the genes related to metabolic fitness, memory phenotype, immune checkpoints and immune senescence markers on the tested cells. Preferably, such data should also be substantiated by flow-cytometry based phenotyping of not only the tested cells.

Minor comments:

6. The data presented in figure 1 show a significant negative effect of the presence of the FMC63 scFv on the development of DP cells (compare 1928x vs w/oED28z). Do the authors have any explanation for this?
7. I notice that from the inducible CAR construct on (figure 1) all CARs have "all of a sudden" a CD28 transmembrane domain instead of the CD8 transmembrane domain. Is there a specific reason for this? Please elaborate on this at least in the methods section.

Other relevant comments

8. Line 60: Recent reports indicated improved differentiation protocols to make CD8 $\alpha\beta$ -expressing T cells that showed effector functions more closely resembling primary T cells".

Starting from this sentence, the rest of the paragraph misses citations, which are elementary for the introduction.

9. Line 70: ...CD3 δ -mediated signal pathway was enhanced by inhibiting the intracellular immunological checkpoint by CRISPR/Cas9 to make iCAR-T cells proliferative in the tumor.

This sentence in the abstract should preferably contain the name of the modulated intracellular immune checkpoint, DGK α and DGK δ to be more specific .

10. line 147: "cloned" seems incorrect here. Should be "sorted"?

11. In figure 2e lower panel. From which effector to target ratio are these results derived? (20:1?) Please indicate in the legend.

12. In supplementary figure 6 the color and the shape of one of the tested CARs in the line-graph does not match with the labels given next to the graph. Please correct.

Wed 10 Aug 2022

Decision on Article nBME-21-2928A

Dear Prof Kaneko,

Thank you for your revised manuscript, "Engineering differentiation and signaling pathways of CAR-T cells from human iPSCs enhances anti-solid tumor activity". Having consulted with the original reviewers, I am pleased to write that we shall be happy to publish the manuscript in *Nature Biomedical Engineering*.

We will be performing detailed checks on your paper and will send you a checklist detailing our editorial and formatting requirements in due course.

Best wishes,

Pep

Pep Pàmies
Chief Editor, Nature Biomedical Engineering

Reviewer #1 (Report for the authors (Required)):

The authors did a stellar job addressing all comments. I have no further queries.

Reviewer #2 (Report for the authors (Required)):

The revised version is substantially improved.

Reviewer #3 (Report for the authors (Required)):

1. I appreciate the newly added data in Supplementary figure 3 and supplementary figure 16, which justifies the use of the third generation construct. I also find the discussion about this point appropriate.
2. I also appreciate the newly added data about the differences between CARTILC vs CARTCTL. The authors fairly discuss the data by stating that there are not much differences, except a slight better tumor infiltration of CARCTL. I find the discussion on this important point also open and fair and I can understand the (rather intuitive) choice of the authors to go further with iCARTCTL. At this moment I have no further questions on this issue. Further (pre)clinical studies could indeed be more conclusive on the application areas of iCARTILC vs iCARTCTL.
3. I am content with the answer and additional data provided to my question about immune checkpoints and their expression on iCART cells.
4. I am also satisfied with the answer and additional data provided to my question about the use of mbIL15 gene in the iCART cells.
5. I like the revisions on the gene expression profiles, which makes the data more easy interpret and makes more sense to the reader.

6. I am also satisfied with the answers, added data and the revision of the manuscript on my minor comments.

I have no further questions to the authors or comments.

Rebuttal 1

Point-by-point responses to reviewer's comments:

First, we would like to thank all reviewers for their helpful comments, which were very constructive and useful for improving our manuscript even though it took long time. We write our point-by-point responses to each of the reviewers' comments below. Each comment as shown in black and our response as shown in red are listed below:

Reviewer #1 (Report for the authors (Required)):

Ueda, Kaneko, and colleagues report their recent results where they optimized manufacturing of CAR-expressing T-cells from iPSC progenitors. They focus on the generation of GPC3 CAR-armed iT cells that undergo full T-cell differentiation via the DP stage producing functional CD8ab-expressing single-positive T-cells following positive selection with anti-CD3 and dexamethasone. The activity of GPC3 CAR iT cells in vivo is further augmented by disrupting the expression of DGK, a key enzyme limiting the levels of DAG downstream of CD3zeta signaling, and by ectopic expression of the IL-15/IL-15Ra complex on the surface of iT-cells. The resulting GPC3 CAR T-cells armed with the said modification produce robust anti-tumor activity in mouse xenograft models of human hepatoma and ovarian carcinoma. Overall, the study is performed at a high technical level with appropriate controls and models, addresses a key limitation of current cell therapies of cancer, and would be of interest to the audience interested in engineered T-cells. Addressing the following points and questions would further augment the impact of this study.

1. Disruption of DGK isoforms has been previously reported to augment anti-tumor function of T-cells, including CAR T-cells (Riese et al., 2013 and Jung et al., 2018 – both of these studies should be mentioned in the paper). Similarly, the benefit of IL-15/IL-15Ra expression on CAR T-cells is well established in the field (Hurton et al., 2016 and other studies). Why did the authors select these particular modifications over similar strategies that boost Signal 1 (modifying CAR expression level, amplifying zeta chains, disrupting checkpoint inhibitors) and Signal 3 (expression of IL-7 or IL-21 etc)? Disruption of DGK isoforms has been previously reported to augment anti-tumor function of T-cells, including CAR T-cells (Riese et al., 2013 and Jung et al., 2018 – both of these studies should be mentioned in the paper

We would like to thank to reviewer#1 for her/his valuable comments. The reviewer had asked us why we decided to use genetic modifications in this study. Although we had not mention this in the manuscript during the first submission, we had checked certain methods to enhance signal 1 or signal 3. In those attempts, the results of CAR overexpression and disruption or inhibition of checkpoint molecule PD-1 (Supplementary Figs. 6, 7), and cytokine signal enhancement by addition of IL-7 or IL-21 (Supplementary Fig. 9) were added to the manuscript. However, as described in the newly added sentence, we did not find significant benefit for the overall potential of these modifications. Therefore, we decided to use only particular modifications to enhance signals 1 and 3, disrupt DGK, and transduce mbIL-15.

We have now added the following sentences at line 264 –270 and cited the corresponding studies in the manuscript.

To enhance antigen receptor-mediated first signal, we modified PD-1 signaling. PD-1-deleted iPSC was established and differentiated to iCAR-T_{CTL} to assess if PD-1 deletion was effective in keeping the differentiated iCAR-T_{CTL} activated (Supplementary Fig.6a). PD-1 deletion significantly but slightly improved cytotoxicity and proliferation and did not improve ERK phosphorylation and cytokine production of iCAR-T_{CTL} (Supplementary Fig.6b–g). The tumor-suppressive capability was not enhanced by blocking the combination of iCAR-T_{CTL} and PD-1 by antibody (Supplementary Fig.6h).

For signal 3 enhancement, we checked the impact of additional cytokines such as IL-7, IL-21, and IL-15. Among these cytokines, mbIL15 showed the most effective enhancement of proliferation in response to target cells.

We have added the following sentences (line 333–336)

Because IL-15 increased the proliferation of iCAR-T_{CTL} in response to anti-CD3 antibody and target antigen-expressing cell line, the most effective among three kinds of signal 3 cytokines; IL15, IL7 and IL21 (Supplementary Fig.9a), we focused on enhancing the IL-15 signal pathway to improve the persistency *in vivo* and maintain the memory phenotype.

Does the DGK disruption and the IL-15 boost especially benefit iPSC-derived iT-cells, more so than conventional CAR T-cells that have similar modifications?

We would like to thank the reviewer for raising this point. Although we traced the methods reported by Jung et al., 2018, we only achieved on an average 30% of DGK ζ knockout and 5% of DGK α knockout in primary T cells. We believe that such low efficiencies of gene modification can make the comparison difficult at the time of first submission. However, after receiving the feedback from the reviewer, we have added *in vitro* data (Supplementary Figs. 10, 11) and *in vivo* data (modified Fig. 6, Supplementary Fig. 11). We found that enhancement of the tumor suppressive function of IL15 boost in primary CAR-T cell is compatible with that reported in a previous study (Hurton et al., 2016), whereas additional DGKKO in primary CAR-T did not improve their functions, unlike iCAR-T_{CTL} in the low-dose injection model (Fig. 6). Again, we suppose this can be due to the lower efficiency of gene modification on primary T cells (Supplementary Fig. 11) compared with the previous report (Jung et al., 2018) although we traced their methods. We believe that gene modification of iPSC cells will be an affordable strategy to produce precisely gene-modified clonal T cell products.

We added the following sentences in the Results section (411~415).

Similar to the results obtained from the peritoneal injection model, DGK-dKO-iCAR-T_{CTL}-mbIL15tg showed significantly better tumor control and survival than other cohorts (Fig. 6b-d), whereas the same combinational signal enhancement of pCAR-T_{CTL} could not be proved to be effective, which could be attributed to the limited efficiency of genome editing with a current protocol²⁶ we applied (Supplementary Fig.11e).

2. DGK limits the duration and magnitude of TCR signaling in T-cells. Does DGK disruption in iPSC affect the subsequent positive and negative selection of iT cells by modulating their TCR signaling intensity? Are DGK-deficient T-cells more prone to negative selection and does the DGK removal affect the naïve/memory/activation/exhaustion phenotype of the resulting iT cells?

As the reviewer mentioned, DGK dKO has been reported to affect thymic differentiation of murine T cells (Guo et al., PNAS, 2018). We found that DGK disruption in human iPSCs affected the differentiation of HSPCs into CD5CD8DP cells (Supplementary Fig.17a). We speculate that modulating the TCR signaling intensity by DGK disruption affects the subsequent positive and negative selections of iT cells. Therefore, we believe that enhanced TCR signaling and/or CAR tonic signaling on DP cells by DGK disruption will be sufficient to induce “negative selection” to CAR expressing cells. To avoid the unwanted “negative selection” during differentiation, we proved the concept about how to increase the efficiency by additional genetic manipulation. Because we found that the single deletion of DGK α did not affect the differentiation efficiency as is compatible with mice study, we established DGKz-floxed iPSC and generated DGKdKO T cells successfully by inducing CRE after differentiation (Supplementary Fig. 16b-d). This approach would indicate a solution for having DGKdKO iCAR-T cells with high cell production efficacy, possibly for future clinical cell processing.

Considering the phenotype of iPS-T cells after DGK dKO, we checked the impact on the phenotype by FCM and gene expression (Supplementary Fig.8 and Supplementary Fig. 11). According to the results, DGK disruption did not significantly affect the naïve/memory phenotype, except for the upregulation of *CCR7*. It increased the expression of metabolic fitness genes, activation genes, and regulatory genes, decreased NK cell-related activating receptor genes, and exhaustion genes such as *HAVCR2* and *PDCD1*.

We added the following sentences at lines 283–287 in the Results section.

Next, we evaluated their phenotype by FCM and performed gene expression analysis to compare with iCAR-T_{CTL} (Supplementary Fig.8 and Supplementary Fig.11b). DGK disruption did not considerably affect naïve/memory phenotype except slight upregulation of *CCR7*. It increased the expression of metabolic fitness genes, activation genes, and immune regulatory genes, and decreased NK cell-related activating receptor genes and exhaustion genes such as *HAVCR2* (TIM3) and *PDCD1* (PD-1).

We added the following sentences at line 469–487 in the Discussion section.

Certain genetic modifications on iPS cells for functional enhancement may inhibit differentiation. We observed decreased efficiency of T cell differentiation from DGK dKO iPSC, possibly by enhanced duration and magnitude of TCR signaling, resulting in negative selection to a part of differentiating DP cells (Supplementary Fig.17a), which is compatible with the previous observation in DGK-dKO mice²⁴. (*snip*). Good temporal controls of these engineering steps could contribute to the large-scale production of therapeutic T cells as we indicated the concept of temporal control by using of DGK α -/-DGK ζ f/f iPSC cells, which partially rescued the efficiency of T cell differentiation (Supplementary Fig.17a-d). In addition, this strategy could be applied to the temporal control of mbLL15tg, whereas other kinds of temporal control such as endogenous promoters (*CD4*, *TRAC*, *etc.*) and synNotch would be available for further studies^{30,31}. iPS cells can serve as the cell source for an infinite number of genetically engineered cells. This can be an advantage over peripheral blood T cells in terms of certainty, safety, and applicability for the mass production of cell therapies.

Finally – does DGK disruption augment cytotoxicity and survival of CAR iT cells or only modulates their proliferation?

As shown in Figure 3 and Supplementary Figure 11, DGK disruption increased the proliferation and cytokine production but did not increase the killing function of iCAR-T cells.

We added the following sentences at line 293-294 in Result.

In addition, improved IFN γ and TNF production from DGK-dKO-iCAR-T_{CTL} was observed in response to SK-Hep-GPC3 (Supplementary Fig.11d).

3. Does IL-15 overexpression affect the phenotype and subset composition of GPC3 iT cells or their expansion upon tumor challenge in vitro? Do the authors envision modifying mature iT cells with the IL-15/IL-15Ra complex or this modification can be made in parental iPSC?

IL-15-mediated signal is known to increase the proliferation, cell survival, and cytotoxicity of NK cells and cytotoxic T cells. We checked the impact of mbIL15 on their phenotype by FCM (Supplementary Fig.9) and gene expression profile (Supplementary Fig.11). The results indicated that mbIL15 overexpression increased the expression of certain early memory-related marker genes such as *LEF1* and *SELL*, increased the expression of AKT/mTOR signal-related genes, and decreased the expression of exhaustion-related markers in iCAR-T cells, whereas it enhanced their accumulation and expansion to tumors in the animal model (Fig.4, Supplementary Fig. 10).

IL-15-mediated signal is known to drive common lymphoid progenitor cells to natural killer cell progenitors in combination with AKT/mTOR signal activation (Ali et al., Front. Immunol. 6:355, 2015). In the presence of IL-15 from the beginning of iPSC differentiation, we also observed NK-lineage-biased differentiation to CD4/CD8 double-negative NK cell progenitor cells (Kitayama et al., Stem Cell Rep. 10, 2018). According to the results, we envision modifying mature iPS-T cells with mbIL15 or using iPS cells, which has the mechanism to conditionally express the *mbIL-15* gene after differentiation to CD8 T cells.

We added the following sentences at line 338–346 in the Results section.

We transduced the mbIL15 gene (mbIL15tg) with a retroviral vector (pMX-mbIL15; kindly provided by Dr. Nakayama; Fig. 4a) into iCAR-TCTL and pCAR-TCTL (Supplementary Fig.9b) and checked the impact of mbIL15 on their phenotype by FCM (Supplementary Fig.9c) and gene expression profile (Supplementary Fig.11b). The mbIL15 gene slightly transduced iCAR-TCTL but significantly increased the expression of memory-related markers such as CCR7 and CD62L showing no elevation of exhaustion-related marker expression (Supplementary Fig.9c,d). With respect to the gene profile, mbIL15 overexpression increased the expression of certain early memory-related marker genes such as *LEF1* and *SELL*, increased the expression of AKT/mTOR signal-related genes, and decreased the expression of exhaustion-related markers in iCAR-T_{CTL}.

We added the following sentences at line 475–487 in the Discussion section.

The IL-15-mediated signal is known to drive common lymphoid progenitor cells to natural killer cell progenitor in combination with AKT/mTOR signal activation²⁸. In the presence of IL-15 during T-lineage cell differentiation from iPSCs, we observed NK-lineage-biased differentiation to CD4/CD8 β double-negative NK progenitor cells¹⁷. It is compatible with previous observations in IL15tg mice²⁹. Therefore, in this study, we modified matured iCAR-T cells by mbIL15 during TCR stimulation-mediated expansion after T cell differentiation. Good temporal controls of these engineering steps could contribute to the large-scale production of therapeutic T cells as we indicated the concept of temporal control by using of DGK α -/-DGK ζ /f iPS cells, which partially rescued the efficiency of T cell differentiation (Supplementary Fig.17a-d). In addition, this strategy could be applied to the temporal control of mbIL15tg, whereas other kinds of temporal control such as endogenous promoters (*CD4*, *TRAC*, etc.) and synNotch would be available for further studies^{30,31}. iPS cells can serve as the cell source for an infinite number of genetically

engineered cells. This can be an advantage over peripheral blood T cells in terms of certainty, safety, and applicability for the mass production of cell therapies.

4. The combination of IL-15 expression and DGK disruption significantly enhances the anti-tumor function of GPC3 CAR iT-cells in models of HCC and ovarian carcinoma. How generalizable are these findings? Would the same modifications enhance the function of iT cells expressing other CARs?

We would like to thank the reviewer to mention this highly important point. To know if this combinatory modification strategy can be applied to other CARs, we transduced CD19 CARs into GPC3 iCAR-T_{CTL}, and DGK dKO GPC3 iCAR-T_{CTL}-mbIL15 and evaluated their effector functions against CD19-expressing human B cell leukemia cell line Nalm-6. Under this setting, we found increased ERK phosphorylation, improved persistency *in vivo*, and significant suppression of leukemia burden in DGK dKO CD19 iCAR-T_{CTL}-mbIL15.

We added the following sentences at lines 441–447 in the Results section.

To know if this combinatory modification strategy could be applied to other CARs, we transduced second-generation 19bbzCAR into above-characterized GPC3 iCAR-T_{CTL} and DGK dKO GPC3 iCAR-T_{CTL}-mbIL15, and found signal enhancement and proliferation advantage of the combination of IL-15 expression and DGK disruption *in vitro* (Supplementary Fig. 16a–e) as well as enhanced T cell survival and tumor suppressive capability *in vivo* (Supplementary Fig. 16f–i). These results suggest that enhancing the combinational signals of iCAR-T_{CTL} could form the basis for different CAR-modified regenerative T cell immunotherapies

Minor points:

1. In the abstract, the authors state the modified iT cells will “significantly prolong survival with an accelerated cancer immunity cycle”. I suggest the authors clarify exactly what they mean as the cancer-immunity cycle is quite multifaceted and encompasses several other aspects not covered in this study.

We deleted “with an accelerated cancer immunity cycle” from the sentence.

2. In Figure 2d, CD62L is downregulated whereas CCR7 is upregulated in mature iCAR-T. Usually T-cells exhibit an opposite pattern where CD62L+ T-cells lose CCR7 expression. Is the observed phenotype stable in mature T-cells or it reflects transient shedding of CD62L and upregulation of CCR7 driven by strong TCR signaling?

As shown in Supplementary Fig6b, we did not find the recovery of *CD62L* expression after proliferation *in vitro*; in contrast, mbIL-15 modification resulted in iCAR-T_{CTL} recovered *SELL* expression, and DGK knockout resulted in iCAR-T_{CTL}-recovered *CCR7* expression even after proliferation (Supplementary Fig.11). Presently, we could not yet elucidate the specific regulatory mechanism of *CCR7* and *CD62L* especially in iCAR-T_{CTL}, which could be due to *in vitro* differentiation process of T cells from iPSCs. We are now trying to identify the mechanism through gene expression analysis during differentiation and perturbation analysis for further development of iPS-T cells; however, it would be a topic for the next manuscript.

3. In Figure 2g, the authors show modestly increased accumulation of iCAR-T CTL over iCAR-T ILC in tumors. To rule out that these effects are due to the variability in the number of cells injected to tumor-

bearing mice, the authors should show the signal on day 0 (right after cell injection) is comparable in all these groups.

We added *in vivo* imaging data of iCAR-T_{CTL} and iCAR-T_{ILC} on day 0 acquired in an experiment as Supplementary Fig.5a. We did not detect accumulated luminescence signal in tumor on day 0.

4. The discussion section is a bit unfocused and should reflect authors' thoughts on the most appropriate strategies to address current limitations of iPSC by either boosting stimulation (Signals 1-2-3) or other approaches that could bridge the gap between conventional and iPSC-derived T-cells.

We would like to thank the reviewer for the comment to improve the Discussion section. We added the following sentences at lines 489–516 in the Discussion section.

iCAR-T_{CTL} retains the basic properties of CAR-T cells such as homing, proliferation, and cytotoxicity at the tumor site. However, as exemplified by the results of ERK phosphorylation, signaling in the differentiated cell is generally attenuated than iCAR-T cells although the underlying causes remain unclear. We consider this could be a result of the overall quality in each step of *ex vivo* manipulation to induce hematopoietic progenitor cells, progenitor T cells, CD4CD8DP T cells, and matured CD8ab T cells, etc. Therefore, each differentiation step needs to be improved physiologically, similar to the differentiating cells in our body. A recently reported method for iPSC-T cell differentiation using organoid culture could improve the quality of iCAR-T_{CTL}³². In contrast, we here report methods to improve the deficient functions by genetic manipulations. Thus, the optimization of signals 1 to 3 has an independent impact on iPSC-derived T cells in terms of their induction and function. Specifically, signal 1 impacts the proliferation and effector functions, signal 2 impacts the activation and tonic signaling, and signal 3 impacts cell survival and persistency. A combination of DGK deletion and mbIL15 transduction is one of the examples of such modifications. Because of great advances achieved in CRISPR-Cas9-based human genome editing, several new clinical applications are under development, especially in the field of T cell immunotherapy. For example, certain clinical trials using primary T cells with *TCR*- and/or *HLA*-knockout are known to diminish the alloreactivity of T cells^{33,34}, whereas *PD-1*- or *CTLA-4* knockout can overcome immunosuppression of the tumor microenvironment^{35,36}. Because iPSCs can be manipulated as a single cell clone, it is possible to achieve 100% genome editing accuracy of the target without off-target effects. By targeting DGK, we demonstrated that gene editing can enhance its function by regulating the expression of intracellular molecules, which cannot be manipulated by antibody administration such as anti-PD1 and anti-CTLA4 antibodies. PD1 inhibition, the same checkpoint molecule, unexpectedly exerted no impact. However, it is known that PD1 expression does not increase in iPSC-T cells without frequent stimulation¹⁰. Therefore, it is inferred that the response was not as strong as that of pCAR-T. Conversely, a combination of DGK deletion and mbIL15 transduction exerted a limited effect on T cells (Figs.5,6); however, it was effective in boosting the function of less-responsive iCAR-T cells. In addition to the approach used in this experiment, several target genes exist that can improve each signal; it is important to optimize the combination of manipulations of these genes, especially for iPSC-derived T cells because they tend to have less responsiveness as mentioned above. It is possible that a more intense approach could be more useful than that for primary T cells.

5. While the write-up is clear and relatively easy to follow, the manuscript would benefit from additional editing for clarity and language.

The revised manuscript has been additionally edited by commercial editing service.

Reviewer #2 (Report for the authors (Required)):

In this study the authors engineered iPSCs in a stepwise fashion to obtain highly functional iCAR-T cells with improved proliferative capacity and persistence at the tumor site. They modified the CAR constructs (eg including CD3z/CD28/4-1BB) to avoid tonic signaling and thereby optimized the differentiation process to achieve CD8ab⁺ T cells via a CD4CD8 double positive stage. The authors compare iCAR-T(ILC) with iCAR-T(CTL) and find that the adaptive type cells have superior capability to accumulate at the tumor site than innate type regenerated cells. They move on to further engineer the iCAR-T cells by deleting DAG kinases to enhance the TCR signalling. Together with further transduction of mbIL15, this led to significantly improved proliferative capacity resulting in better tumor control *in vivo*.

Overall, this is an ambitious study that shed further light on the role of tonic signalling in differentiation fates and the functional differences of such cells. The work also paves way for tuning TCR signalling to achieve more responsive iPSC-T cells with better persistence in the tumor. The combination approach shown in Figure 4 has merit and lifts the manuscript substantially towards the end.

Major concerns

The head-to-head comparison of innate and adaptive iCAR-T cells in Figure 2 is a bit problematic, as is often the case when comparing cells generated through different protocols. In most *in vitro* assays, the “innate” cells appear to outperform the “adaptive” cells (2e) and the differences *in vivo* are modest (non-significant) and almost at background levels compared to pCAR-T cells (Fig 2f,g). It is unclear if the different differentiation protocols per se (different stimulatory input) or the end phenotype/cell state is responsible for any differences seen in Figure 2. I realize there is no way to get around this problem since different protocols are required to generate the different cell fates. However, given the modest functional phenotype, statements such as: “...adaptive type regenerated T cells... have superior capacity to accumulate at the tumor site *in vivo* than innate type...” are not supported by the data and needs to be toned down or substantiated by additional experiments.

We would like to thank the reviewer for her/his important comment about potential difficulty to compare different types of cells induced from stem cells using different protocols. To avoid the bias caused by different materials and protocols for having iCAR-T_{ILC} and iCAR-T_{CTL}, we used T/NK progenitor cells from a single iPSC clone in the same culture as starting cells to obtain iCAR-T_{ILC} and iCAR-T_{CTL}. In addition, we used the same protocol to expand both cells in same frequencies. These efforts were made to reduce indicated biases. Agreeing to the reviewer’s comment, we toned down a level of our claims in the manuscript.

The aim of experiments shown in Fig. 2 was to evaluate which iPS-derived immune cells were better for further modification as CAR-based cell therapy platform to obtain therapeutic effect as close as primary CAR-T cells using the solid tumor animal model. Therefore, we thought the level of accumulation did not matter in Figure 2 but thought it to be important to know the tendency between iCAR-T_{CTL} and iCAR-T_{ILC} although the level was a kind of background in comparison to pCAR-T cells. (Please be aware that polyclonal TCR-expressing pCAR-T cells were continuously stimulated and activated by murine xenoantigens in the animal model, which elevated the background noise of *in vivo* imaging worse than iPSC-derived monoclonal TCR expressing cells). Actually, iCAR-T_{CTL} with *mbIL-15*

transgene become closer about therapeutic potential (accumulation and cell persistency) of primary CAR-T cells and further be confirmed better than iCAR-T_{ILC} with mbIL-15 (Fig 4, Fig S 10, S15).

The results obtained from additional substantiated experiments suggested by the reviewer show that that our initial aim and meaning of the experiments in Figure 2 became clearer. We would like to thank the reviewer for the same.

We added the following sentences at lines 430–440 in the Results section.

Next, we evaluated the therapeutic impact of these genetic modifications on iCAR-T_{ILC} in comparison with iCAR-T_{CTL} to understand if these modifications generated better iCAR-T_{ILC} than iCAR-T_{CTL} (Supplementary Figs.12,13,14,15). In comparison with DGK-dKO iCAR-T_{CTL}, we did not find any advantage of DGK-dKO iCAR-T_{ILC} about ERK phosphorylation and proliferation in co-culture with SK-Hep-GPC3 (Supplementary Fig. 12). In addition, we did not observe any significant results with respect to peritoneally disseminated tumor control when test cells were directly injected into the peritoneal cavity of immunodeficient mice (Supplementary Fig. 13). iCAR-T_{ILC} exhibited stronger but non-specific cytotoxicity to tumor cell lines than iCAR-T_{CTL} (Supplementary Fig.14) after transduction of the *mbIL-15* gene, however, we still found inferiority of tumor accumulation of iCAR-T_{ILC}-mbIL15tg in comparison to iCAR-T_{CTL}-mbIL15tg when they were intravenously injected into JHH7-bearing mice (Supplementary Fig.15). Based on these results, we conclude that iCAR-T_{CTL}-based modified cells would be useful for solid tumor immunotherapy.

We added the following sentences at lines 518–530 in the Discussion section.

Although cytotoxic ILCs (NK cells) are similar to cytotoxic T lymphocytes (CTLs) in their ability to eliminate target cells, they differ significantly in the manner they sense target cells and in their kinetics³⁷. It is reported that iPSC-derived T cells with properties similar to ILC/NK appear during T cell differentiation from iPSCs. A direct comparison of iCAR-T_{ILC} and iCAR-T_{CTL} differentiated from identical CAR-iPSCs revealed their different properties that were compatible with those of their physiological counterparts, namely, NK cells and CTLs. mbIL-15, which was found to be useful in iCAR-T_{CTL} in this study, has been reported to be useful in cord blood-derived NK cells, and in improving the persistency of CD19CAR iPST cells. In this study, the DGK deletion for iCAR-T_{ILC} resulted in improved effector function comparable to DGK-dKO-iCAR-T_{CTL}, and mbIL-15 improved the persistency *in vivo*. However, the lack of improvement in the subcutaneous tumor accumulation suggested that these two modifications are insufficient for iCAR-T_{ILC} and may indicate differences in properties from iCAR-T_{CTL}. However, iCAR-T_{ILC} has an advantage in cytotoxicity, and the expression of mbIL15 from the iPS cell stage may be advantageous for NK cell differentiation, which would be reflected in future clinical development¹⁵.

The authors state that the comparison of the transcriptomes of these three cell populations suggest that iCAR-T(CTL) have superior memory T cell function. However, this is based on n=1 and is not definitive. Overall, this section (Fig 2 and Supp Fig 2) needs to be much substantiated or perhaps rather abandoned? The key observation seems to be that iCAR-T(CTL) underperform relative to pCAR-T and that further modifications are needed.If the authors, would like to keep the comparison between “innate” and “adaptive” iCAR T cells and their ability to accumulate at the tumor site additional experimental support for this statement is needed. For example, the data shown in Supp Fig 3a is based on how many experiments, sections? To me it makes more sense to move the comparison of innate and adaptive phenotypes to the supplement and simply state that the failure of unmodified iCAR-T (CTL) versus pCAR-T is also seen using conventional differentiation strategies yielding iCAR-T (ILC).

We would like to thank the reviewer for the important comment related to the accuracy of results and readability of the manuscript. Considering that the result was from $n = 1$ experiment, we have modified the sentence mentioning the impact of the phenotypic difference between iCAR-T_{CTL} and iCAR-T_{ILC} obtained by the scRNA assay ($N = 1$). Instead, we described the underperformance of iCAR-T_{CTL} relative to pCAR-T in the Results section. The number of experiments in Supp Fig 3b was three ($n = 3$). We believe that additional substantiated experiments suggested by the reviewers (Fig.S10 and 15) made it clearer that iCAR-T_{CTL} showed longer persistency in tumors than iCAR-T_{ILC} under the same genetic modification by *mbIL-15*. It made the initial aim and meaning of the experiments in Figure 2 clearer.

We have modified the following sentences to tone down our claims at lines 177–187 in the Results section. If the reviewer still feel better to delete the description related to the scRNA data, we could delete the claims.

Among differentially expressed genes (DEGs) between iCAR-T_{CTL} and iCAR-T_{ILC}, the expression of naïve/memory-related genes, such as *SELL*, *CCR7*, *TCF7*, *IL7R*, and *CD27*, was high in iCAR-T_{CTL}, suggesting that CD5CD8 β DP iCAR-T_{CTL} maintained a suitable phenotype for therapeutic efficacy *in vivo* compared with iCAR-T_{ILC} even after proliferation (Supplementary Fig.2c). A gene ontology (GO) analysis of those genes revealed a high enrichment of genes related to T cell differentiation (fold enrichment = 47.34, FDR = 2.24E-11) and T cell activation (fold enrichment = 34.04, FDR = 1.06E-11). On the other hand, the top 30 DEGs for pCAR-T_{CTL} compared with iCAR-T_{CTL} included *IL2*, *IFNG*, *TNF*, and *IL7R*, indicating possible enrichment of terms for cytokine-mediated signaling pathways (fold enrichment = 21.12, FDR = 8.18E-23) and lymphocyte activation (fold enrichment = 23.71, FDR = 3.75E-13). These results suggested that iCAR-T_{CTL} could be functionally closer to pCAR-T_{CTL} than iCAR-T_{ILC} (Supplementary Fig.2d). However, it could be insufficient in multiple aspects of cancer immunity in comparison to pCAR-T_{CTL}.

As a general concern, all figure legends need to include information of number of experiments. One representative experiment out of x ? Key findings need to be based on multiple experiments and include proper statistics. For transcriptional comparisons, this means inclusion of multiple independent differentiation runs from the same iPSC line or ideally at assessment of two independently generated lines. As I am sure the authors are well aware of, iPSC lines have “personalities” so to compare wt ($n=1$) with one gene edited version ($n=1$) will likely generate a lot of DEG that are irrelevant and would not show up in a second wt/KO pair. It would be best if the authors identify which comparison that is best suited for deep transcriptional analysis in order to gain mechanistic insights into the behaviour of the mature iCAR-T cells. Perhaps the DAGkinase KO line to gain insight into its superior persistence?

We would like to thank the reviewer for the important comment related to the accuracy of results. As the reviewer mentioned, we have included the information on the number of experiments in all figure legends. For transcriptional comparisons, we used multiple independent differentiation runs ($n = 3$) from the same iPSC line (Fig S11). We agree to the reviewer’s point about the risk of having irrelevant DEGs from the comparison within the same clone; therefore, we toned down the description about the result part and showed the actual expression data of representative genes related to T/NK cell functions such as activation, co-stimulation, metabolic fitness, memory phenotype, immune checkpoints, and immune senescence to gain insight into functional change on mbIL-15 and/or DGK knockout (Fig S11). We observed the individual gene expression change related to T cell functions such as activation, co-stimulation, metabolic fitness, memory phenotype, immune checkpoints, and immune senescence.

We have added the following description about the comparison of T/NK cell functions related to the gene expression between DGK-dKO vs. DGK-wildtype (n=3) at lines 279–287 in the Results section.

Although we observed decreased efficiency of T cell differentiation along with disruption of both DGKs (Supplementary Fig.17a), which is compatible with the previous observation in DGK-dKO mice²⁴, we successfully obtained DGK-dKO-iCAR-T_{CTL} that were confirmed to have no DGK α and DGK ζ proteins (Supplementary Fig.8b). Next, we evaluated their phenotype by FCM and performed gene expression analysis to compare with iCAR-T_{CTL} (Supplementary Fig.8 and Supplementary Fig.11b). DGK disruption did not considerably affect naïve/memory phenotype except slight upregulation of *CCR7*. It increased the expression of metabolic fitness genes, activation genes, and immune regulatory genes, and decreased NK cell-related activating receptor genes and exhaustion genes such as *HAVCR2* (TIM3) and *PDCD1* (PD-1).

We have added the following description about the comparison of T/NK cell functions related to the gene expression between mbIL-15tg vs. no modification (n=3) at lines 341–346 in the Results section.

The *mbIL15* gene slightly transduced iCAR-T_{CTL} but significantly increased the expression of memory-related markers such as *CCR7* and *CD62L* showing no elevation of exhaustion-related marker expression (Supplementary Fig.9c,d). With respect to the gene profile, *mbIL15* overexpression increased the expression of certain early memory-related marker genes such as *LEF1* and *SELL*, increased the expression of AKT/mTOR signal-related genes, and decreased the expression of exhaustion-related markers in iCAR-T_{CTL}.

We have added the following description about GO terms extracted from the above two sets of comparison (DGK-dKO vs. DGK wild-type, mbIL-15 transduced vs. untransduced) and about comparison of T/NK cell function-related gene expression by additional mbIL-15 to DGK-dKO at lines 376-387 in the Results section.

We examined the impact of each signal enhancement on the cell phenotype by comparing the gene expression profiles of iCAR-T_{CTL}, iCAR-T_{CTL}-mbIL15tg, DGK-dKO-iCAR-T_{CTL}, and DGK-dKO-iCAR-T_{CTL}-mbIL15tg. A PCA and hierarchical clustering analysis revealed that iCAR-T_{CTL}-mbIL15tg and DGK-dKO-iCAR-T_{CTL} formed a distinct population from iCAR-T_{CTL} (Supplementary Fig.11a). The transduction of the *mbIL15* gene resulted in an enrichment of DNA conformation change, DNA replication, chromosome organization, DNA metabolic process, DNA-dependent replication, and cell cycle, whereas DGK-dKO induced enrichment of inflammatory response, regulation of response to external stimulus, locomotion, cell migration, regulation of cell proliferation, and cell motility (Supplementary Table 2). DGK-dKO increased the activation and co-stimulation-related genes, whereas mbIL15tg increased the expression of naiveness-related genes and decreased that of exhaustion-related genes. In combination with both manipulations, DGK-dKO iCAR-T_{CTL}-mbIL15tg showed additional expression of genes related to cell survival and persistence such as *TP53*, *MYC*, and *ICOS* to DGK-dKO iCAR-T_{CTL} (Supplementary Fig.11b).

The discussion is a bit review-like and could perhaps discuss the explored concepts in more detail in relation to previous literature. In particular it would be interesting to discuss how the addition of 41BB provide co-stimulation and yet counter the tonic signalling by z/28.

We would like to thank the reviewer for her/his comment to improve the Discussion section. We have added the following sentences at line 458–468 in Discussion.

A tonic signal is attributed to the aggregation of CAR molecules that cause CAR-T exhaustion¹² and are reported to inhibit the expression of master transcription factor BCL11L by inhibiting the Notch signaling that affects the lymphopoiesis of CAR-modified hematopoietic stem and progenitor cells^{14,22}. We

investigated how the engineering of iPSCs impacts their differentiation propensity and found that the 28z construct decreased the differentiation efficiency of iPSCs to CD4CD8 DP cells through tonic signaling. In addition, we found that replacement with 4-1BBz or additional 4-1BB signaling to 28z attenuated the tonic signal during differentiation. This finding is consistent with that reported in the literature on primary CAR-Ts that an additional 4-1BB signal to 28z-based CAR restricted the downstream Zap70 phosphorylation at a basal level as well as after antigenic stimulation, thus preserving the therapeutic function by different affinity CARs²⁷. Although the detailed mechanism of how additional 4-1BB signaling rescues T cell differentiation from iPSC needs to be elucidated, we believe a compatible mechanism to that reported previously should be present.

The statement (row 60) that recent reports indicated improved differentiation protocols to make CD8ab-expressing T cells that showed effector functions more closely resembling primary T cells. Which reports are the authors referring to here? Please add citations.

We would like to thank the reviewer for the comment on readability. We have added three citations for the sentence “Recent reports indicated improved differentiation protocols to make CD8αβ-expressing T cells showing effector functions more closely resembling those of primary T cells”.

Reviewer #3 (Report for the authors (Required)):

This is an interesting paper addressing a number of relevant issues in T-IPSC derived CAR-T (iCAR-T cells) cells. In the first part of the study, the authors addressed the optimal CAR-design for the generation of CD4/CD8 double positive and CD8ab iCAR-T cells using the well-known CD19-targeting FMC63 CAR.

In the second part, in which the glypican3 (GPC3) CARs were used, the authors aim to improve the overall functionality, in vivo persistence and anti-tumor efficacy of CD8ab+ iCAR-T cells (designated as iCARCTLs in the paper) by i) knock out of an earlier described immune checkpoint genes DGK α and DGK δ and ii) by ectopic expression (retroviral transduction) of the membrane-bound IL-15/IL-15Ra (mbIL15) in these cells.

I appreciate the data in figure 4, which demonstrate that the final product (designated as DGK-dKO-iCAR-TCTL-mbIL15tg cells) indeed possesses an improved in vivo survival and superior anti-tumor efficacy, which is even better than the anti-tumor efficacy of a second generation conventional CART cells (pCAR-TCTL). The results confirm that also iPSC-derived CART cells can significantly benefit from genetic modulations which are aiming at preventing exhaustion and armoring the cells with (cytokine) signals for a better in vivo survival and persistence.

Nonetheless, in the light of the data presented especially in figures 1 and 2, I have the impression that the final product was not entirely developed by careful analysis and comparison of the other possibly powerful products. To my opinion this also affects the scientific merit of the study, because:

1. The data presented in figure 1 do not show a clear advantage of using a third generation CAR (CD28+BB1 co-stimulation) above a second generation CAR containing only BB1 costimulation. This is because the use of either 2nd.Gen BB1 CAR or the 3rd.gen. CAR results in the development of similar levels of DP T cells. Thus, a systematic analysis would also include the 2nd.gen.BBz CARs in the second part of the study.

I would like to thank the reviewer for the constructive comment. We have added comparison data between third-generation and second-generation 4-1BB iCAR-T_{CTL} about effector functions (Supplementary Fig 3). In the comparison, we found that third-generation iCAR-T_{CTL} could produce IFN- γ and TNF slightly but significantly better than second-generation 4-1BB iCAR-T_{CTL} in coculture with tumor cell line expressing target molecule of CAR, although it did not cause significant difference in tumor growth control in the animal model. Based on the results, we selected third-generation anti-GPC3 CAR for the study but we could understand the reviewer's point. Thus, in addition to Fig S3, we added an experiment using iCAR-T cells modified by CD19 targeted second-generation BBz CAR in combination with DGK dKO and mbIL-15 modifications, which strengthen the usefulness of the genetic modification strategy in the study to different types of CARs (Supplementary Fig. 16).

We have added the following sentences at line 195–200 in the Results section.

We did not observe any difference between the impact of 4-1BBz-based second-generation and third-generation CAR on T cell differentiation. Thus, we compared cytokine production between second-generation BBz iCAR-T_{CTL} and third-generation 28BBz iCAR-T_{CTL}, and found that 28BBz iCAR-T_{CTL} produced IFN- γ and TNF significantly better than BBz iCAR-T_{CTL} following SK-Hep-GPC3 stimulation (Supplementary Fig.3). Thus, we selected third-generation 28BBz iCAR-T_{CTL} for further experiments.

We have added the following sentences at lines 441–447 in the Results section.

To know if this combinatory modification strategy could be applied to other CARs, we transduced second-generation 19bbzCAR into above-characterized GPC3 iCAR-T_{CTL} and DGK dKO GPC3 iCAR-T_{CTL}-mbIL15, and found signal enhancement and proliferation advantage of the combination of IL-15 expression and DGK disruption *in vitro* (Supplementary Fig. 16a–e) as well as enhanced T cell survival and tumor suppressive capability *in vivo* (Supplementary Fig. 16f–i). These results suggest that enhancing the combinational signals of iCAR-T_{CTL} could form the basis for different CAR-modified regenerative T cell immunotherapies.

2. The data presented in figure 2 (and related supplementary figures) do not show a clear advantage of using iCAR-CTL above iCAR-ILC.

Although iCAR-CTLs seem to better accumulate in the tumor tissue than iCAR-ILC, this advantage disappears within 14 days (no significant difference between iCAR-CTL and iCAR-ILC at day 14 in figure 2g); In contrast, the iCAR-ILCs - thanks to their additional NK-dependent kill capacity- are significantly better killers of the antigen positive tumors as compared to iCAR-CTLs (figure 2e). Finally, there is no difference between iCAR-CTL and iCAR-ILC with respect to their *in vivo* anti-tumor efficacy in an intraperitoneal tumor model (supplementary figure 2f,g,i,h). Hence, a systematic analysis towards the optimal product should also include the testing of iCAR-ILCs in the further stages of the study.

We would like to thank the reviewer for the comment about genetic modifications in iCAR-T_{ILC}. The aim of experiments shown in Figure 2 was to evaluate which iPSC-derived immune cells were better for further modification as CAR-based cell therapy platform to obtain therapeutic effect as close as that of primary CAR-T cells using a solid tumor animal model. Therefore, we thought the level of accumulation did not matter in Figure 2 but had thought it showed a tendency between iCAR-T_{CTL} and iCAR-T_{ILC} although the level was a kind of background in comparison to pCAR-T cells. (Please be aware that polyclonal TCR expressing pCAR-T cells were continuously stimulated and activated by murine xenoantigens in the animal model, which elevated the background noise of *in vivo* imaging worse than iPSC-derived monoclonal TCR-expressing cells). Following the reviewer's comment, we added the systematic analysis about how the difference between the two types of cells along with our selected two kinds of modifications; DGK-dKO and mbIL15tg. In the experiments, we found that DGK-dKO-iCAR-T_{ILC} showed high cytokine production and proliferation capability as well as DGK-dKO-iCAR-T_{CTL}; however, accumulation to the tumor and persistency of iCAR-T_{ILC}-mbIL15tg were found to be significantly lower than those of iCAR-T_{CTL}-mbIL15tg after 14 days. Particularly, iCAR-T_{CTL} with *mbIL-15* transgene has higher therapeutic potential in terms of accumulation and cell persistency for primary CAR-T cells and was further confirmed to be better than iCAR-T_{ILC} with mbIL-15 (Fig 4, Fig S 10, S15).

The results obtained from additional experiments suggested by the reviewers substantiated our initial aim and meaning of the experiments in Figure 2 became clearer. We would like to thank the reviewer also for the point.

We have added the following sentences at lines 430–440 in the Results section.

Next, we evaluated the therapeutic impact of these genetic modifications on iCAR-T_{ILC} in comparison with iCAR-T_{CTL} to understand if these modifications generated better iCAR-T_{ILC} than iCAR-T_{CTL} (Supplementary Figs. 12, 13, 14, 15). In comparison with DGK-dKO iCAR-T_{CTL}, we did not find any advantage of DGK-dKO iCAR-T_{ILC} about ERK phosphorylation and proliferation in co-culture with SK-Hep-GPC3 (Supplementary Fig. 12). In addition, we did not observe any significant results with respect to peritoneally disseminated tumor control when test cells were directly injected into the peritoneal cavity of immunodeficient mice (Supplementary Fig. 13). iCAR-T_{ILC} exhibited stronger but non-specific cytotoxicity

to tumor cell lines than iCAR-T_{CTL} (Supplementary Fig.14) after transduction of the *mbIL-15* gene, however, we still found inferiority of tumor accumulation of iCAR-T_{ILC}-mbIL15tg in comparison to iCAR-T_{CTL}-mbIL15tg when they were intravenously injected into JHH7-bearing mice (Supplementary Fig.15). Based on these results, we conclude that iCAR-T_{CTL}-based modified cells would be useful for solid tumor immunotherapy.

We have added the following sentences at lines 518–530 in Discussion.

Although cytotoxic ILCs (NK cells) are similar to cytotoxic T lymphocytes (CTLs) in their ability to eliminate target cells, they differ significantly in the manner they sense target cells and in their kinetics³⁷. It is reported that iPSC-derived T cells with properties similar to ILC/NK appear during T cell differentiation from iPSCs. A direct comparison of iCAR-T_{ILC} and iCAR-T_{CTL} differentiated from identical CAR-iPSCs revealed their different properties that were compatible with those of their physiological counterparts, namely, NK cells and CTLs. mbIL-15, which was found to be useful in iCAR-T_{CTL} in this study, has been reported to be useful in cord blood-derived NK cells, and in improving the persistency of CD19CAR iPST cells. In this study, the DGK deletion for iCAR-T_{ILC} resulted in improved effector function comparable to DGK-dKO-iCAR-T_{CTL}, and mbIL-15 improved the persistency *in vivo*. However, the lack of improvement in the subcutaneous tumor accumulation suggested that these two modifications are insufficient for iCAR-T_{ILC} and may indicate differences in properties from iCAR-T_{CTL}. However, iCAR-T_{ILC} has an advantage in cytotoxicity, and the expression of mbIL15 from the iPS cell stage may be advantageous for NK cell differentiation, which would be reflected in future clinical development¹⁵.

3. The choice of knock down of DGK α and DGK δ genes is not based on any data specific for iPSC derived CART cells as there are a plethora of potential immune checkpoints that could be modulated. It is from the manuscript not even clear which immune checkpoints are (over) expressed on/in these cells.

About checkpoint molecules, no exhaustion-related molecules such as PD-1, CTLA4, LAG3, and TIGIT were expressed on the surface of iCAR-T_{CTL} except for TIM3 where a portion of the cells expressed (Supplementary Fig.8 and Supplementary Fig11). These results led us to hypothesize that the blockade of such representative exhaustion molecules without enough expression can not improve the therapeutic efficacy of iCAR-T CTL. Consistent to the hypothesis, disruption or inhibition of checkpoint molecule PD-1 as shown in Supplementary Fig. 6 did not induce substantial enhancement of ERK phosphorylation in iCAR-T cell. Because we obtained no positive results of cell surface checkpoint molecule disruption such as PD-1, TIM3, and TIGIT (Fig S6, and Data not shown), we next attempted to disrupt intracellular checkpoint molecules by CRISPR/Cas9, and DGK was listed as the most potent target by its function in human T cells and previously reported results about DGK knockout in murine T cells, although the expression of DGKs was not significantly elevated in iCAR-T cells. DGK is known to inhibit CD3z-mediated TCR/CAR-signaling by degrading the signaling molecule DAG that could induce phosphorylation of RAS and ERK.

We have added the gene expression heatmaps of iCAR-T_{CTL} including exhaustion-related genes (Supplementary Fig. 11) and added the following sentences at lines 264–272 in the Results section.

To enhance antigen receptor-mediated first signal, we modified PD-1 signaling. PD-1-deleted iPSC was established and differentiated to iCAR-T_{CTL} to assess if PD-1 deletion was effective in keeping the differentiated iCAR-T_{CTL} activated (Supplementary Fig.6a). PD-1 deletion significantly but slightly improved cytotoxicity and proliferation and did not improve ERK phosphorylation and cytokine production of iCAR-T_{CTL} (Supplementary Fig.6b–g). The tumor-suppressive capability was not enhanced by blocking the combination of iCAR-T_{CTL} and PD-1 by antibody (Supplementary Fig.6h). As a different approach,

CAR overexpression slightly improved cytotoxicity; however, it showed less proliferation and cytokine producibility with increasing expression of exhaustion markers (Supplementary Fig. 7a–g).

4. Similarly, the decision to insert the mbIL5 gene in the cells is primarily based on earlier published successful studies but not data derived from the study.

For cytokine receptor-mediated signal 3 enhancement, we evaluated the impact of additional cytokines such as IL-7, IL-21, IL-15 in the culture. Among these conditions, IL-15 showed the most effective enhancement of proliferation in response to target cells (Fig S9a). Thus, we decided to enhance IL-15-mediated signaling in iCAR-T by transduction of mbIL-15 following an earlier published successful study.

We have added the following sentences at lines 333–336 in the Results section.

Because IL-15 increased the proliferation of iCAR-T_{CTL} in response to anti-CD3 antibody and target antigen-expressing cell line, the most effective among three kinds of signal 3 cytokines; IL15, IL7 and IL21 (Supplementary Fig.9a), we focused on enhancing the IL-15 signal pathway to improve the persistency *in vivo* and maintain the memory phenotype.

5. Finally, the provided genetic expression profiling studies are too global to provide specific clues on the quality of the cells. To my opinion the genetic data could be further analyzed to obtain specific information about the genes related to metabolic fitness, memory phenotype, immune checkpoints and immune senescence markers on the tested cells. Preferably, such data should also be substantiated by flow-cytometry based phenotyping of not only the tested cells.

We would like to thank the reviewer for her/his constructive comment about genetic data. As suggested by the reviewer, we reanalyzed the data to obtain specific information about the genes related to metabolic fitness, memory phenotype, immune checkpoints, and immune senescence markers in tested cells (FigS11). We observed the certain gene expression changes related to T-cell functions including metabolic fitness, memory phenotype, immune checkpoints, and immune senescence. The expression of some of the surface molecules was also confirmed by FCM.

We have added the following description about comparison of T/NK cell function-related gene expression between DGK-dKO vs. DGK-wild-type (n=3) at lines 279-287 in the Results section.

Although we observed decreased efficiency of T cell differentiation along with disruption of both DGKs (Supplementary Fig.17a), which is compatible with the previous observation in DGK-dKO mice²⁴, we successfully obtained DGK-dKO-iCAR-T_{CTL} that were confirmed to have no DGK α and DGK ζ proteins (Supplementary Fig.8b). Next, we evaluated their phenotype by FCM and performed gene expression analysis to compare with iCAR-T_{CTL} (Supplementary Fig.8 and Supplementary Fig.11b). DGK disruption did not considerably affect naïve/memory phenotype except slight upregulation of *CCR7*. It increased the expression of metabolic fitness genes, activation genes, and immune regulatory genes, and decreased NK cell-related activating receptor genes and exhaustion genes such as *HAVCR2* (TIM3) and *PDCD1* (PD-1).

We have added the following description about comparison of T/NK cell function-related gene expression between mbIL-15tg vs. no modification (n=3) at lines 341-346 in the Results section.

The *mbIL15* gene slightly transduced iCAR-T_{CTL} but significantly increased the expression of memory-related markers such as CCR7 and CD62L showing no elevation of exhaustion-related marker expression (Supplementary Fig.9c,d). With respect to the gene profile, *mbIL15* overexpression increased the expression of certain early memory-related marker genes such as *LEF1* and *SELL*, increased the expression of AKT/mTOR signal-related genes, and decreased the expression of exhaustion-related markers in iCAR-T_{CTL}.

We have added the following description about GO terms extracted from the above two sets of comparison (DGK-dKO vs. DGK wild-type, mbIL-15 transduced vs. untransduced) and about comparison of T/NK cell function-related gene expression by additional mbIL-15 to DGK-dKO at lines 376-387 in the Results section.

We examined the impact of each signal enhancement on the cell phenotype by comparing the gene expression profiles of iCAR-T_{CTL}, iCAR-T_{CTL}-mbIL15tg, DGK-dKO-iCAR-T_{CTL}, and DGK-dKO-iCAR-T_{CTL}-mbIL15tg. A PCA and hierarchical clustering analysis revealed that iCAR-T_{CTL}-mbIL15tg and DGK-dKO-iCAR-T_{CTL} formed a distinct population from iCAR-T_{CTL} (Supplementary Fig.11a). The transduction of the *mbIL15* gene resulted in an enrichment of DNA conformation change, DNA replication, chromosome organization, DNA metabolic process, DNA-dependent replication, and cell cycle, whereas DGK-dKO induced enrichment of inflammatory response, regulation of response to external stimulus, locomotion, cell migration, regulation of cell proliferation, and cell motility (Supplementary Table 2). DGK-dKO increased the activation and co-stimulation-related genes, whereas mbIL15tg increased the expression of naivness-related genes and decreased that of exhaustion-related genes. In combination with both manipulations, DGK-dKO iCAR-T_{CTL}-mbIL15tg showed additional expression of genes related to cell survival and persistence such as *TP53*, *MYC*, and *ICOS* to DGK-dKO iCAR-T_{CTL} (Supplementary Fig.11b).

Minor comments:

6. The data presented in figure 1 show a significant negative effect of the presence of the FMC63 scFv on the development of DP cells (compare 1928x vs w/oED28z). Do the authors have any explanation for this?

We would like to thank the reviewer' for the important suggestion. It is previously reported that spontaneous aggregation of CARs due to scFV causes tonic signal even without the target antigens (Long et al, *Nat Med*, 2015) and the signal disturbs the normal differentiation process from human HPC to T cells (Maluski et al., *J Clin Invest*, 2019). In this study, we speculate that uncontrollable tonic signal on DP cell enhances TCR signaling, which is sufficient for inducing apoptosis of CAR-expressing cells as "negative selection."

We have added the following explanation at lines 103-107 in the Results section.

Recent investigations reported that 1928z transduction into HSCs impaired T cell differentiation capability^{14,13} and promoted NK-like cell development by suppressing the transcription factor BCL11B, which is indispensable for T lineage development of lymphoid progenitors during early phases of *ex vivo* T cell generation. It could be a possible reason also for T cell differentiation from 1928z CAR-transduced iPSC⁵.

7. I notice that from the inducible CAR construct on (figure 1) all CARs have "all of a sudden" a CD28 transmembrane domain instead of the CD8 transmembrane domain. Is there a specific reason for this? Please elaborate on this at least in the methods section.

There was no specific reasons for using CD28TM for inducible CAR construct in the study, which came along with the CAR in a previous report (Kowolik CM et al., CD28 costimulation provided through a CD19-specific chimeric antigen receptor enhances *in vivo* persistence and antitumor efficacy of adoptively transferred T cells. *Cancer Res.* 2006 Nov 15;66(22):10995-1004).

We have added the following description in lines 578–579 at the Methods section.

For the inducible CAR construct, we used the CD28 transmembrane domain⁴⁷.

Other relevant comments

We would like to thank to the reviewer for careful reviewing.

8. Line 60: Recent reports indicated improved differentiation protocols to make CD8 $\alpha\beta$ -expressing T cells that showed effector functions more closely resembling primary T cells”. Starting from this sentence, the rest of the paragraph misses citations, which are elementary for the introduction.

We have added three citations for the sentence “Recent reports indicated improved differentiation protocols to make CD8 $\alpha\beta$ -expressing T cells that showed effector functions more closely resembling primary T cells” and added one citation for the sentence “CAR transduction to such iPS-T cells was confirmed to work as effectively as primary T cells on a B-cell malignancy animal model when the iPS-T cells were supported with an IL-15-mediated third signal.”

9. Line 70: ...CD3 δ -mediated signal pathway was enhanced by inhibiting the intracellular immunological checkpoint by CRISPR/Cas9 to make iCAR-T cells proliferative in the tumor. This sentence in the abstract should preferably contain the name of the modulated intracellular immune checkpoint, DGK α and DGK δ to be more specific .

We have added the molecules’ name in the sentence as follows:

Next, the CD3 ζ -mediated signal pathway was enhanced by inhibiting the intracellular immunological checkpoint molecules, namely, DGK α and DGK ζ by CRISPR/Cas9, to allow the proliferation of iCAR-T cells in the tumor.

10. line 147: "cloned" seems incorrect here. Should be “sorted”?

We have modified the sentence to explain the details of methods in lines 152–154 in the Results section as follows:

A CAR-targeting GPC3 (Fig. 2a) was transduced into iPSCs (CAR-iPSCs) using a lentiviral vector, and the cells were cloned by limiting the dilution after selection by stably expressing humanized Kusabira Orange 1 (hKO1).

11. In figure 2e lower panel. From which effector to target ratio are these results derived? (20:1?) Please indicate in the legend.

We have added the following sentence to explain the detail of methods in the legend of Figure 2d:

Lower panel shows the cytotoxicity at E:T ratio = 20 :1.

12. In supplementary figure 6 the color and the shape of one of the tested CARs in the line-graph does not match with the labels given next to the graph. Please correct.

We matched the color and shape in all figures related to the combination study.